# DAPE V2: Process Attention Score as Feature Map for Length Extrapolation

## Abstract

The attention mechanism is a fundamental component of the Transformer model, contributing to interactions among distinct tokens. In general, the attention scores are determined simply by the key-query products. However, this work's occasional trial (combining DAPE and NoPE) of including additional MLPs on attention scores without position encoding indicates that the classical key-query multiplication may limit the performance of Transformers. In this work, we conceptualize attention as a feature map and apply the convolution operator (for neighboring attention scores across different heads) to mimic the processing methods in computer vision. Specifically, **the main contribution of this paper is identifying and interpreting the Transformer length extrapolation problem as a result of the limited expressiveness of the naive query and key dot product, and we successfully translate the length extrapolation issue into a well-understood feature map processing problem.** The novel insight, which can be adapted to various attention-related models, reveals that the current Transformer architecture has the potential for further evolution. Extensive experiments demonstrate that treating attention as a feature map and applying convolution as a processing method significantly enhances Transformer performance.

## 1 Introduction

Transformer-based models (Vaswani et al., 2017) have delivered exceptional performances across widespread applications, including language processing (Zhang et al., 2020; Guo et al., 2022; Ainslie et al., 2023), computer vision (Alexey, 2020; Touvron et al., 2021; Liu et al., 2021a; Chen et al., 2024; Peebles & Xie, 2023), quantitative research (Zhou et al., 2024b; Liu et al., 2021b; Wu et al., 2023), and scientific machine learning (Taylor et al., 2022; Geneva & Zabaras, 2022). However, the quadratic cost of the key-query multiplication for processing a sequence raised much concern about the modern architecture of Transformers especially for long context inputs. To address the issue of storage and computation efficiency, recent research delves into developing more efficient architectures, such as sparse structural attention (Xiao et al., 2024d; Zhu et al., 2024), adaptive key selection (Xiao et al., 2024a; Fountas et al., 2024), and hybrid models (Lieber et al., 2024). While these adaptations enhance efficiency, they often involve tradeoffs with model effectiveness.

At the same time, there is another voice advocating for refining the model design for tackling complex tasks, rather than prioritizing efficiency. Positional encoding is one of the key components of the attention mechanism. Although the widely recognized decoder-based Transformer can implicitly incorporate the positional information of tokens, growing evidence both theoretically and empirically shows that the well-designed explicit positional encoding significantly enhances the model performances, especially in long-context tasks (Su et al., 2024b; Press et al., 2021; Zhao et al., 2023). In practice, Transformers depend on positional encoding to explicitly incorporate positional information, enabling the model to make meaningful token predictions. Without these encodings, token generation would lack the necessary contextual order. The well-recognized RoPE (Su et al., 2024b), which is adopted in LLaMA (Touvron et al., 2023), distinguishes the token order by rotating with different angles depending on the token position. However, it demonstrated a notable performance degradation, failing entirely when the input length is double that of the training length (Peng et al., 2023b; Chen et al., 2023a; Ding et al., 2024b). The undesirable performance degradation is also observed for other positional encoding methods, e.g., ALiBi (Press et al., 2021) and Kerple (Chi et al., 2022) . FIRE (Li et al., 2023c) alleviates the long-context extrapolation by learnable posi-

tional encodings, trying to capture the suitable positional representation by MLPs. Recently, the data-adaptive positional encoding method, namely DAPE (Zheng et al., 2024), which adjusts dynamically with context, enhances the length generalization by incorporating the attention scores and positional information with a more complex mechanism.

In this paper, we propose that precise attention scores are crucial for improving Transformer length extrapolation, and we introduce a new perspective on the attention mechanisms. Traditionally, attention scores are computed through the dot product of the query and key vectors. As illustrated in Figure 1, further processing these attention scores using a neural network—a general case of DAPE (Zheng et al., 2024)—can significantly enhance the length generalization of Transformers, even in the absence of positional encoding (NoPE). Therefore, we suggest treating attention scores as feature maps. By conceptualizing attention as an image feature map (with dimensions $[B, C, W, H]$ for batch size, channel size, width, and height), we can achieve more accurate attention scores by applying techniques used in image processing. In this work, we employ different kernel sizes (such as 1×3) to process attention, finding that the perplexity (ppl) of attention decreases significantly—from over 600 to just above 100—when trained on a sequence length of 128 and evaluated on a length of 8192.

In summary, our contributions are as follows:

1. We highlight that the coarse attention mechanism, which is the direct result of the query and key dot product, limits the Transformer's ability to extrapolate to longer sequences. However, Transformers can achieve good length extrapolation performance with careful processing of attention scores.

2. Besides developing better position encoding (Vaswani et al., 2017) or position interpolation (Chen et al., 2023b) for length extrapolation, we propose the thid direction: by treating attention scores as feature maps and refining them using image processing techniques like convolution, we can enhance the Transformer's extrapolation capabilities.

3. We conducted extensive experiments on language tasks to support our claims and believe that these insights can significantly improve the Transformer's performance in length extrapolation.

## 2 RELATED WORKS

**Absolute Positional Encoding**  Absolute positional encoding (APE), introduced by Vaswani et al. (2017), enables Transformers to incorporate positional information. Specifically, at the first layer, each position $i$ is assigned a real-valued encoding $\boldsymbol{e}_i \in \mathbb{R}^d$, which can be either learnable or a fixed sinusoidal encoding (Vaswani et al., 2017; Kiyono et al., 2021; Likhomanenko et al., 2021; Wang et al., 2020; Liu et al., 2020), and this encoding is then added to the input sequence. Although this approach is straightforward, Transformers relying on APE tend to struggle with generalizing to longer sequences (Press et al., 2021).

**Relative Positional Encoding**  Relative positional encoding (RPE) offers an alternative for embedding positional information (Shaw et al., 2018; Raffel et al., 2020; Press et al., 2021). A widely used RPE method in large language models is rotary positional encoding (RoPE)(Su et al., 2024b; Chowdhery et al., 2023; Touvron et al., 2023). To address length extrapolation challenges(Press et al., 2021; Kazemnejad et al., 2024), positional interpolation (PI) has been introduced (Chen et al., 2023b) to extend the context window. Building on this approach, models like LongLora (Chen et al., 2023c), LongRope (Ding et al., 2024b), YaRN (Peng et al., 2023b), and CLEX (Chen et al., 2023a) have emerged. Another notable direction involves additive positional encoding. For most additive RPE techniques, the computation of pre-softmax attention logits can be expressed using the formula: $\boldsymbol{A}_{\mathrm{RPE}}(\boldsymbol{X}) = \boldsymbol{X}\boldsymbol{W}_Q(\boldsymbol{X}\boldsymbol{W}_K)^\top + \boldsymbol{B}$, where the bias matrix $\boldsymbol{B} \in \mathbb{R}^{n \times n}$ is derived from the positional encoding function $b : \mathbb{N}^2 \to \mathbb{R}$, with the $(i, j)$-th entry of $\boldsymbol{B}$ defined as $b(i, j)$. Different parameterizations of $b$ give rise to various RPE variants. Methods supporting arbitrary sequence lengths include T5's RPE (Raffel et al., 2020), ALiBi (Press et al., 2021), Kerple (Chi et al., 2022), Sandwich (Chi et al., 2023a), and FIRE (Li et al., 2023c). Recently, DAPE (Zheng et al., 2024) has been introduced, employing MLPs to dynamically adjust bias values based on the input data.

**Data-Adaptive Related Positional Encoding.** Transformer-XL (Dai et al., 2019) introduced the use of learnable query and key biases for adaptive positional encodings. Data-Adaptive Positional Encoding (DAPE)(Zheng et al., 2024) extends this idea by leveraging MLPs to adjust positional encodings based on attention over the head dimension for length extrapolation, ensuring different input data receive unique positional encodings. Contextual Positional Encoding(Golovneva et al., 2024) further refines this by conditioning position increments on specific tokens, as determined by the model, allowing positions to adapt based on context."

## 3 METHOD

In this section, we first review the previously developed Data-Adaptive Positional Encoding method (DAPE), which incorporates attention scores and positional information through MLPs. As a proof-of-concept, our occasional trial on DAPE without the positional information (as shown in Figure 1) suggests that regarding attention as a feature map and processing it with classical operators (e.g., convolution) can enhance the Transformers' behavior. As discussed in some previous works the perplexity scores come mostly from the associative recall (i.e., copy) tasks. In addition, we theoretically show by construction that the proposed method can explicitly realize the associative recall task, in contrast to the implicit conduct through positional encoding in standard Transformers. **The two key differences between DAPE (Zheng et al., 2024) and this work are: 1) Insight:** DAPE attributes length extrapolation performance gains to adaptive position encoding, while this work finds DAPE could still improve performance without position encoding so that we take a broader view, explaining that the Transformer's length extrapolation ability is limited by the expressiveness of the naive query-key dot product, which can be enhanced using image processing techniques; **2) Performance:** As shown in Figure 1, DAPE is designed for additive RPE and may underperform with non-additive RPE (e.g., RoPE), whereas this work suggests that increasing kernel size (e.g., with $\text{DAPE}_{1\times3}$) may improve RoPE's performance. The $\text{DAPE}_{1\times3}$ implementation is shown in Appendix L.

### 3.1 ADDITIVE RELATIVE POSITIONAL ENCODING

For most additive relative positional encoding (ARPE) methods, the computation of pre-softmax attention logits can be unified under the following formula:

$$\boldsymbol{A}_{\text{ARPE}}(\boldsymbol{X}) = \boldsymbol{X}\boldsymbol{W}_Q(\boldsymbol{X}\boldsymbol{W}_K)^\top + \boldsymbol{B}, \tag{1}$$

where the bias matrix $\boldsymbol{B} \in \mathbb{R}^{n \times n}$ is induced by the position encoding function $b : \mathbb{N}^2 \to \mathbb{R}$ and the $(i, j)$-th entry of $\boldsymbol{B}$ is defined as $b(i, j)$. Various formulations and parameterizations of $b$ give rise to different variants of RPE. Examples of additive RPE include: (1) ALiBi: $b(i, j) = -r|i - j|$, with the scaler $r > 0$ as a hyper-parameter; (2) Kerple: $b(i, j) = -r_1 log(1 + r_2|i - j|)$ with $r_1$ and $r_2$ are two learnable parameters; (3) FIRE: $b(i, j) = f_\theta \left( \frac{\psi(i-j)}{\psi(\max\{L,i\})} \right)$, where the positional encoding function $f_\theta$ parameterized by $\theta$ is learned from data and $\psi$ is a transformation function aimed at assigning more model capacity to local positions.

**Data-Adaptive Position Encoding (DAPE)** The DAPE rewrite the Equation 1 as the following:

$$\boldsymbol{A}_{\text{DAPE}}(\boldsymbol{X}) = \boldsymbol{X}\boldsymbol{W}_Q(\boldsymbol{X}\boldsymbol{W}_K)^\top + f(\boldsymbol{X}\boldsymbol{W}_Q(\boldsymbol{X}\boldsymbol{W}_K)^\top, \boldsymbol{B}). \tag{2}$$

Here, $f : \mathbb{R}^{T \times T} \times \mathbb{R}^{T \times T} \to \mathbb{R}^{T \times T}$ is an element-wise function and $T$ is the sequence length. Another variant of DAPE is with residual, which is the following:

$$\boldsymbol{A}_{\text{DAPE}}(\boldsymbol{X}) = \boldsymbol{X}\boldsymbol{W}_Q(\boldsymbol{X}\boldsymbol{W}_K)^\top + \boldsymbol{B} + f(\boldsymbol{X}\boldsymbol{W}_Q(\boldsymbol{X}\boldsymbol{W}_K)^\top, \boldsymbol{B}). \tag{3}$$

In practice, DAPE (Zheng et al., 2024) utilizes a two-layer *LeakyReLU* MLP with hidden dimension $D_{\text{DAPE}}$ (default value is 32) to parameterize $f(\cdot)$ due to its universal approximability (Leshno et al., 1993). All parameters are learned directly from the data during the training process. This architecture allows $f(\cdot)$ to dynamically adjust positional embeddings based on the input sequence data, ensuring that the encoding method is both adaptive and dependent on the input data.

### 3.2 SPECIAL CASE OF DAPE: BIAS IS ZERO

DAPE was originally designed to dynamically adjust the positional encoding by incorporating input data information. Generally, any additive positional encoding method that includes positional information can be represented as the matrix $\boldsymbol{B}$ in the DAPE model, as outlined in Equation 2. Notably,

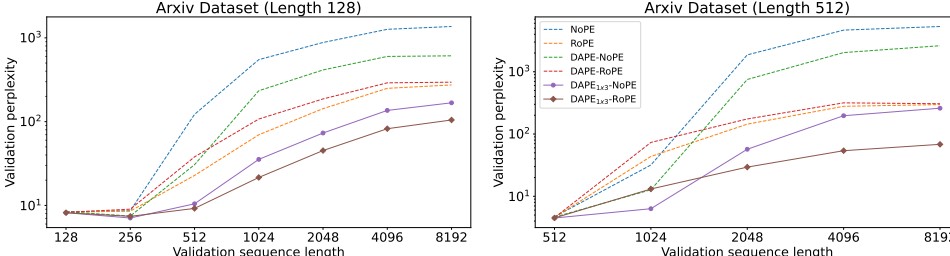

Figure 1: **The result of DAPE (Zheng et al., 2024) (equivalent to kernel $1 \times 1$ in our explanation) and DAPE$_{1 \times 3}$ (kernel $1 \times 3$ by this work), with baseline NoPE and RoPE.** The model is trained with length 128 and length 512 respectively. The DAPE$_{1 \times 3}$ denotes that we use $H \times 1 \times 3$ convolutions kernel size on the attention score with shape $[B, H, T, T]$. **We find that DAPE can even improve the performance of NoPE (without biased position encoding), suggesting that the explanation in Zheng et al. (2024), which attributes the improvement to adaptive position encoding, may have a more general underlying cause.**

No Positional Encoding (NoPE) (Kazemnejad et al., 2024) is a special case of additive RPE that assigns zero value to the matrix $\boldsymbol{B}$. The mathematical formulation of DAPE equipped with NoPE is given by:

$$\boldsymbol{A}_{\text{DAPE}}(\boldsymbol{X}) = \boldsymbol{X}\boldsymbol{W}_Q(\boldsymbol{X}\boldsymbol{W}_K)^\top + f(\boldsymbol{X}\boldsymbol{W}_Q(\boldsymbol{X}\boldsymbol{W}_K)^\top). \tag{4}$$

The DAPE Zheng et al. (2024) is designed for additive RPE but not trying NoPE or RoPE, and we present the results of DAPE-NoPE and DAPE-RoPE in the following.

**The result of DAPE-NoPE**  Compared with the standard Transformer architecture, DAPE-NoPE introduces additional MLPs post the key-query multiplication and prior to the softmax operator. As shown in Figure 1, experimental evidence suggests that DAPE with NoPE significantly outperforms the basic NoPE, prompting a reconsideration of the behaviors of standard Transformers. The additional MLPs (i.e., denoted as $f(\cdot)$ in Equation 4) facilitate information sharing across attention heads and complicate the attention calculation with nonlinear transformation beyond the simple key-query multiplication. This leads to a critical question: *Is the current Transformer architecture, particularly the attention mechanism, sufficiently expressive for real-world language tasks?* Although numerous studies aim to enhance efficiency by reducing computation and storage in standard Transformers, these often come at the cost of effectiveness, potentially hindering the evolution of next-generation Transformer models. Motivated by these insights and observations, we enhance the Transformer's expressiveness and behavior by regarding attention as a feature map and applying convolutional operations, akin to those used in computer vision.

**The result of DAPE-RoPE.**  Building on the hypothesis that DAPE enhances Transformer performance by processing pre-softmax scores with MLPs, we explore its applicability to non-additive positional encoding methods, specifically RoPE (Su et al., 2024b). In the DAPE-RoPE configuration, DAPE-RoPE first computes the classic attention scores of key-query multiplication with RoPE, which are then refined using the MLPs described in Equation 4. The visualized results of the validation perplexity for DAPE-RoPE and other positional encoding methods are presented in Figure 1. The results indicate that DAPE-RoPE may degrade the performance, while DAPE$_{1 \times 3}$-RoPE (with kernel size $1 \times 3$, propsoed by this work) not only improves overall performance but also excels in length extrapolation tasks, particularly at larger sequence lengths. This finding substantiates the effectiveness of DAPE$_{1 \times 3}$-RoPE, confirming its superior performance compared to standard RoPE, attributing to the additionally introduced convolution operations to the attention scores.

### 3.3 DAPE V2: Process Attention Scores as Feature Maps

As discussed above, improving Transformer performance necessitates refining the processing of attention score computation beyond the conventional key-query multiplication. We propose regarding the pre-softmax attention scores as feature maps (4-dimensional tensors) and applying convolutional operators, which may could additionally involve position information with zero padding and higher expressiveness (Kayhan & Gemert, 2020) but MLP does not involve additional position information because there is no zero padding. This approach facilitates enhanced communication across neighboring tokens and heads, drawing parallels to popular techniques used in computer vision. This

novel method aims to leverage the spatial relationships within tokens, potentially unlocking new aspects of model capabilities.

**Rethink the DAPE formulation.** In DAPE (Zheng et al., 2024), MLPs are utilized to process and integrate attention and biases. Notably, these MLP operations can be equated to convolution operations with $1 \times 1$ kernel (Krizhevsky et al., 2012; Simonyan & Zisserman, 2014; He et al., 2016), a stride of one, and no padding. Consequently, we can reformulate the DAPE in Equation 3 as the following:

$$A_{\text{DAPE}}(\boldsymbol{X}) = \boldsymbol{X}\boldsymbol{W}_Q(\boldsymbol{X}\boldsymbol{W}_K)^\top + \boldsymbol{B} + Conv(tril((\boldsymbol{X}\boldsymbol{W}_Q(\boldsymbol{X}\boldsymbol{W}_K)^\top, \boldsymbol{B})). \tag{5}$$

where $\boldsymbol{X}$ is the input embedding, $\boldsymbol{X}\boldsymbol{W}_Q$ gives the query embedding and the $\boldsymbol{X}\boldsymbol{W}_K$ gives the kery embedding. Under such formulation, DAPE employs convolution operation to process the pre-softmax attention scores of key-query multiplication. The $\texttt{tril}(\cdot)$ returns the lower triangular part of the matrix and the other elements of the result tensor out are set to 0. The resulting attention tensor has a shape of $[B, H, T, T]$, where the four dimensions correspond to the batch size, number of heads, and the context length for both the query and key. This mirrors the structure of an image feature tensor with shape $[B, C, H, W]$, where the dimensions represent the batch size, number of channels, image height, and image width, respectively. This structural similarity underscores the feasibility of considering attention scores as a tensor of feature mappings, where popular and effective convolution operations can be leveraged for refined processing.

**Process attention with more powerful convolution operation.** In computer vision, the limitations of $1 \times 1$ kernels for processing image features are well-recognized. To improve upon the attention scores processed by these kernels (e.g., DAPE), we introduce $1 \times k$ kernels with a stride of 1 and padding of $k-1$. This approach allows for wider and deeper convolution across key dimensions and heads without information leakage, as we ensure the attention scores remain lower-triangular. This mechanism is visualized in Appendix K. The use of $1 \times k$ kernels suggests a targeted convolution along the key dimensions across heads. In general, while extending this to include the query dimensions as a standard kernel is theoretically possible, it would significantly increase computational demands. Our forthcoming analysis demonstrates that Transformers modified with $1 \times k$ convolution are adept at associative recall tasks (i.e., the copy task), validating the benefits of integrating convolution in attention calculation. We left as a future work investigating the performances and the soundness of general convolution kernels, such as square sizes. **The key contribution of this work is providing a novel insight that suggests applying convolution operations and processing attention as feature maps to improve Transformers' performances.**

**Realizing associate recall tasks through convolution.** As pointed out in some previous works (Arora et al., 2024), the perplexity scores of Transformers mostly result from the performances on associate recall tasks (i.e., the copy tasks). Numerous studies have explored the mechanism of associative recall within Transformers, both from theoretical perspectives and experimental validations (Arora et al., 2024; Bietti et al., 2024; Golovneva et al., 2024). Here, we theoretically prove that the proposed model can realize the associative recall tasks. Notably, this capability is achieved independently of positional encodings, marking a significant advancement in the flexibility and applicability of the proposed architecture. By integrating convolutional operations, we enable the model to handle associative tasks more effectively, leveraging spatial relationships inherent in the data, similar to methods used in image processing. To explain the associative recall mechanism, (Bietti et al., 2024) proved that the first layer of the Transformer is responsible for the previous token mechanism through the positional encoding. More specifically, given a sequence of input tokens $\boldsymbol{X} = [\boldsymbol{x}_1, \boldsymbol{x}_2, \cdots, \boldsymbol{x}_N]$ with corresponding orthogonal positional encoding vectors $[\boldsymbol{p}_1, \boldsymbol{p}_2, \cdots, \boldsymbol{p}_N]$, the first layer primarily facilitates the copying of the previous token to the current token (e.g., $\boldsymbol{x}_i + \boldsymbol{W}_V^1 \boldsymbol{x}_{i-1}$, where $\boldsymbol{W}_V^1$ is the value matrix at the first layer of the Transformer). The input tokens are combined with positional encodings $\boldsymbol{x}_i + \boldsymbol{p}_i$ and the key-query weight matrix is defined as $\boldsymbol{W}_K^{1\top}\boldsymbol{W}_Q^1 = \sum_{i=1}^N \boldsymbol{p}_{i-1}\boldsymbol{p}_i^\top$. The orthogonality of positional encoding vectors and the special choices of the key-query matrix ensure that attention scores predominantly focus on the previous token. In contrast to this implicit mechanism in standard Transformers, our proposed method leverages a convolution operation to explicitly realize associative recall. This approach not only simplifies the process but also enhances its effectiveness by directly manipulating the spatial relationships within tokens and attention scores. Consider a scenario where the word "Hakuna" is

consistently followed by "Matata" within a lengthy paragraph. Without the loss of generality, we assume that $x_1$ and $x_2$ represent the tokens of "Hakuna" and "Matata" respectively, and $x_N = x_1$ implies that the N-th token in the sequence is "Hakuna". Then we expect that the Transformer can predict and output the next token $x_{N+1}$ as "Matata". For simplicity, we consider a one-head Transformer without positional encoding. We employ a convolution operation with a kernel size of $1 \times 2$ and weights $[-1, 1]$. Note that the convolution is linear and processing the attention scores along the key dimensions is effectively equivalent to applying convolutions directly to the key vectors themselves. Consequently, the key vector of $x_2$ can be expressed as $W_K^1 (x_2 - x_1)$ and the query vector for $x_N$ admits $W_Q^1 x_N$. By configuring the matrix $W_K^{1\top} W_Q^1$ to be $-I$, the attention mechanism after the convolution predominantly allocates the attention values of $x_N$ to the token $x_2$. This ensures that the token values of $x_2$ are effectively copied to $x_N$, resulting in the model outputting "Matata" following "Hakuna".

**Proposition 1.** *Transformers incorporating convolution operations can perform associative recall tasks without the need for positional encoding.*

**Comparisons with hybrid models of convolution and Transformers.**    Recent developments in hybrid architectures have seen the integration of convolutional and Transformer models to capitalize on the strengths of both. For instance, Fu et al. (2022) introduced the FlashConv layer, which combines the efficiency of State Space Models (SSMs) with the capabilities of attention-based models. Similarly, Arora et al. (2024) developed a gated convolution layer, noted for its effectiveness in addressing associative recall tasks. These models typically stack convolution layers directly with standard Transformer layers, resulting in modifications to the token values through convolution. In contrast, our model adopts a distinctive approach by applying convolution along the key dimension during the computation of attention scores. This method preserves the original token values while still leveraging the convolution's benefits for processing attention.

## 4 EXPERIMENT

**Baselines.**    We evaluate the proposed $\text{DAPE}_{1\times3}$ against several well-established baselines, including NoPE (Kazemnejad et al., 2024), RoPE (Su et al., 2024b), T5's Bias (Raffel et al., 2020), ALiBi (Press et al., 2021), Kerple (Chi et al., 2022), FIRE (Li et al., 2023c), CoPE (Golovneva et al., 2024), and DAPE (Zheng et al., 2024). As our kernels are applied across all heads, we simplify by omitting the kernel size description at the head dimension. For example, $\text{DAPE}_{1\times3}$ indicates the use of a $H \times 1 \times 3$ convolution kernel size on the attention scores, with a shape of $[B, H, T, T]$.

**Datasets.**    Our analysis is based on training language models using the Arxiv and Books3 datasets, commonly employed benchmarks for assessing model performance (Press et al., 2021; Chi et al., 2022; Li et al., 2023c; Ding et al., 2024b). We begin our evaluation by processing entire sequences and comparing the zero-shot perplexity of the last 256 tokens across various input lengths. In addition to perplexity, we also leverage downstream datasets with randomized positional encoding (Ruoss et al., 2023) to further assess $\text{DAPE}_{1\times3}$.

**Experiment settings.**    Initially, we compare $\text{DAPE}_{1\times3}$ with other baselines at training lengths of 128, 512, and 1024, using 125M decoder-only Transformers (Brown et al., 2020), with model configurations detailed in Appendix I. Subsequently, we evaluate the performance of different training lengths using the same number of training tokens but with larger model sizes (350M and 2.7B). We also explore the impact of the convolutional hidden dimension $D_{\text{DAPE}}$, the effect of information leakage, and the influence of varying kernel sizes. Additionally, we examine the computational efficiency of $\text{DAPE}_{1\times3}$, focusing on processing times. Lastly, we evaluate $\text{DAPE}_{1\times3}$ on algorithmic reasoning datasets using accuracy metrics. Compared to DAPE (Zheng et al., 2024), $\text{DAPE}_{1\times3}$ demonstrates a more pronounced attention sink (Xiao et al., 2024d), as visualized in Appendix K.

### 4.1 COMPARE WITH BASELINES

**$\text{DAPE}_{1\times3}$-Kerple improves performance within training length, proving its ability to process the entire sequence.**    According to Figure 2, the proposed $\text{DAPE}_{1\times3}$-Kerple demonstrates superior performance across various training and evaluation lengths. Specifically, $\text{DAPE}_{1\times3}$-Kerple achieves

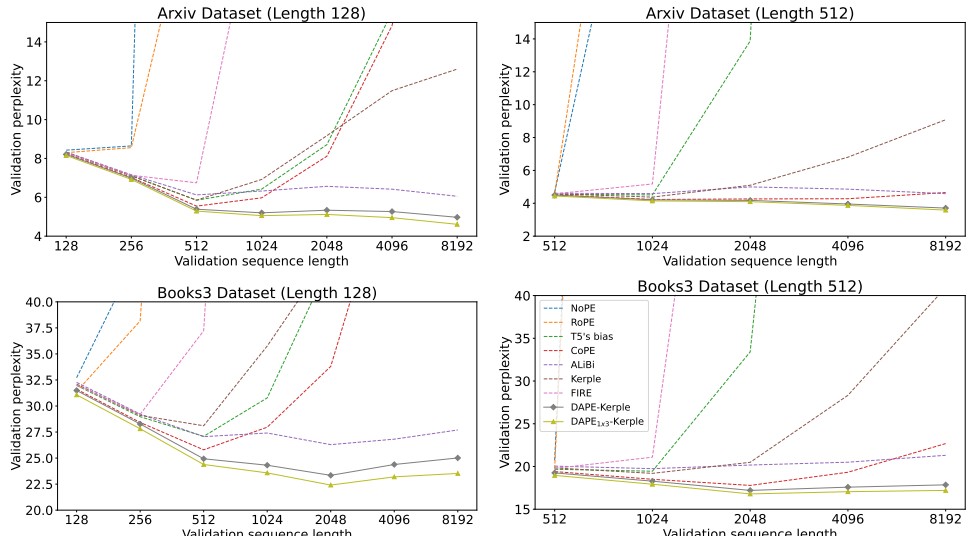

Figure 2: **Comparisons with baselines:** performance with training lengths 128 and 512 on Arxiv and Books3 datasets.

the best performance where the training length is 128 or 512 and the evaluation length ranges from 128 to 8192. This performance consistency is observed across both the arXiv and Books datasets. For instance, on the arXiv dataset with a training length of 512, DAPE$_{1\times3}$-Kerple achieves a perplexity score of 4.44. This score surpasses those of other methods, such as DAPE-Kerple with a perplexity of 4.49, CoPE with 4.51, Kerple with 4.57, and RoPE with 4.57. These results indicate that DAPE$_{1\times3}$-Kerple has a more robust modeling capability within the training length compared to the other methods evaluated. The Appendix B also presents the performance of different methods with training length 1024. The improvements are not only significant but also consistent, reinforcing the efficacy of the DAPE$_{1\times3}$-Kerple approach in handling various training lengths effectively.

**DAPE$_{1\times3}$-Kerple improves performance beyond training length.** The advantages of DAPE$_{1\times3}$-Kerple extend beyond the training length. When the training length is set to 128 and the evaluation length is extended to 8192, DAPE$_{1\times3}$-Kerple achieves a perplexity score of 4.60 on the arXiv dataset and 23.52 on the Books3 dataset. These scores are significantly better than those achieved by DAPE-Kerple, which records perplexity scores of 4.97 and 25.01 on the arXiv and Books3 datasets, respectively. Similarly, CoPE performs poorly with perplexity scores of 29.86 on the arXiv dataset and 90.66 on the Books3 dataset under the same conditions. Furthermore, when the training duration is increased to 512, DAPE$_{1\times3}$-Kerple continues to deliver the best performance, further validating its superior generalization capabilities. These findings highlight the scalability and robustness of DAPE$_{1\times3}$-Kerple, which is attributed to the introduced convolution operator, making it a promising approach for diverse data scenarios and lengths.

## 4.2 PERFORMANCE WITH SAME TRAINING TOKENS AND DIFFERENT TRAINING LENGTH

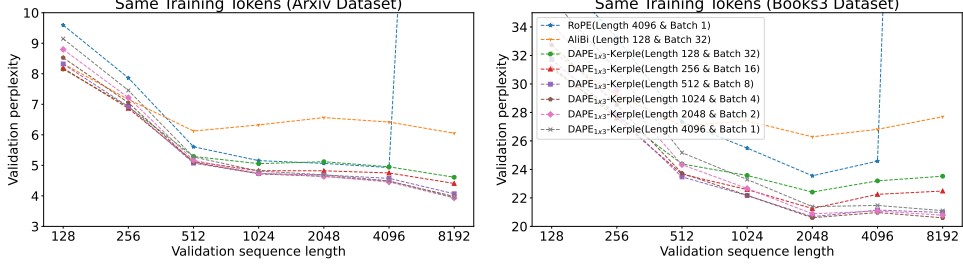

Figure 3: **The performance with same training tokens and different training length**. With the same training tokens, DAPE$_{1\times3}$ with training length 512 could even achieve better performance than RoPE with training length 4096.

**Compared to RoPE, with the same training tokens, DAPE$_{1\times3}$-Kerple with a training length of 128 achieves performance comparable to RoPE with a training length of 4096, for varying evaluation length.** As shown in Figure 3, for DAPE$_{1\times3}$-Kerple trained with a length of 128, it achieves a perplexity (ppl) of 8.15 at an evaluation length of 128 and 4.95 at an evaluation length of 4096 on the arXiv dataset. In comparison, RoPE trained with a length of 4096 achieves a ppl of 9.59 at an evaluation length of 128 and 4.92 at an evaluation length of 4096. Similarly, on the Books3 dataset, DAPE$_{1\times3}$-Kerple trained with a length of 128 achieves a ppl of 31.07 at an evaluation length of 128 and 23.19 at an evaluation length of 4096, while RoPE trained with a length of 4096 achieves 38.36 and 24.58, respectively. This suggests the superiority of the proposed DAPE$_{1\times3}$ with the introduced convolution operators among heads and neighboring tokens.

**With the same training tokens, compared to DAPE$_{1\times3}$ with longer training lengths, DAPE$_{1\times3}$ with shorter training lengths can achieve comparable performance, indicating that DAPE$_{1\times3}$ enhances the model's understanding of text structure.** On the arXiv dataset, DAPE$_{1\times3}$-Kerple with training lengths of 512 demonstrates performance close to that of training with a length of 4096 when the evaluation length is 4096. Moreover, the performance curves for training lengths of 1024, and 2048 are almost identical. This trend is also observed with the Books3 dataset. These results indicate that DAPE$_{1\times3}$-Kerple effectively helps the model comprehend text structure, enabling it to extend to longer lengths.

**Transformers may overfit their training length: training on longer sequences may decrease performance when testing on shorter sequences.** On the arXiv dataset, DAPE$_{1\times3}$-Kerple with a training length of 128 achieves the best performance when the evaluation length is 128. Similarly, DAPE$_{1\times3}$-Kerple with training lengths of 256, 512, 1024, and 2048 achieves the best performance at evaluation lengths of 256, 512, 1024, and 2048, respectively. Also, on evaluation 128, the RoPE with training length 4096 and batch size 1 also achieves worse performance than the RoPE with training length 128 and batch size 32. This suggests that training on longer sequences may worsen a Transformer's performance at shorter sequence lengths.

**DAPE$_{1\times3}$ can reduce the training time cost via larger batch size and shorter training length, achieving comparable performance compared to trained on longer length.** The cost of DAPE$_{1\times3}$ is $\mathcal{O}\big(B \cdot (h \cdot d \cdot T^2 + h \cdot D_{\text{DAPE}} \cdot T^2)\big)$, where $B$, $h$, $d$, $T$ and $D_{\text{DAPE}}$ are the batch size, attention hidden dimension, attention head number, sequence length and DAPE hidden dimension. By reducing the training length from $T$ to $\frac{T}{K}$ and increasing the batch size from $B$ to $B \cdot K$ with the same training tokens, the cost becomes $\mathcal{O}\big(B \cdot K \cdot (h \cdot d \cdot (\frac{T}{K})^2 + h \cdot D_{\text{DAPE}} \cdot (\frac{T}{K})^2)\big)$, which simplifies to $\mathcal{O}\big(\frac{B \cdot (h \cdot d \cdot T^2 + h \cdot D_{\text{DAPE}} \cdot T^2)}{K}\big)$. For example, when the training length is 128 and the batch size is 32, the time cost of one step is 40.30ms. The time cost of length 256 (batch 16), length 512 (batch 8), length 1024 (batch 4), and length 2048 (batch 2) are 42.61ms, 50.38ms, 79.36ms, and 120.14ms. This reduction demonstrates the potential for significant training time savings.

## 4.3 The Effect of Larger Model Size

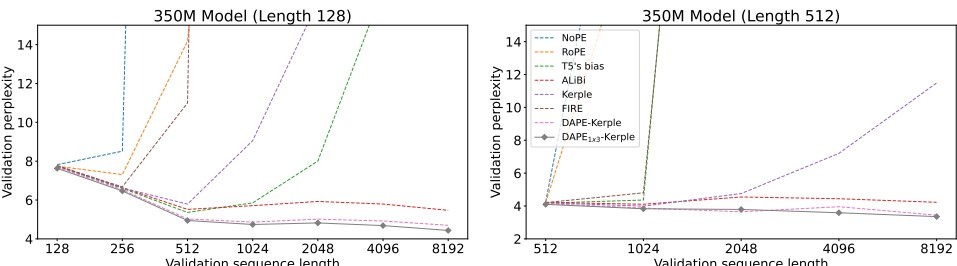

Figure 4: **The Effect of Larger Model Size 350M**. We show the results with training length 128 and training length 512 on Arxiv dataset.

**DAPE$_{1\times3}$ performs well with larger model sizes, such as 350M and 2.7B.** As illustrated in Figure 4, the proposed DAPE$_{1\times3}$ shows superior performance at varying evaluation lengths with a model size of 350M. For a training length of 128, DAPE$_{1\times3}$-Kerple achieves a perplexity (ppl) of 7.63 at an evaluation length of 128 and 4.43 at an evaluation length of 8192, compared to DAPE's

7.69 and 4.69, respectively. Similarly, for a training length of 512, DAPE$_{1\times3}$-Kerple achieves a ppl of 4.10 at an evaluation length of 128 and 3.35 at an evaluation length of 8192, whereas DAPE achieves 4.14 and 3.44, respectively. We also present the 2.7B model size result in Appendix C. Therefore, the proposed DAPE$_{1\times3}$ demonstrates excellent performance with larger model sizes, showing the potential of including the proposed processing techniques in existing large language models.

## 4.4 The Effect of DAPE$_{1\times3}$

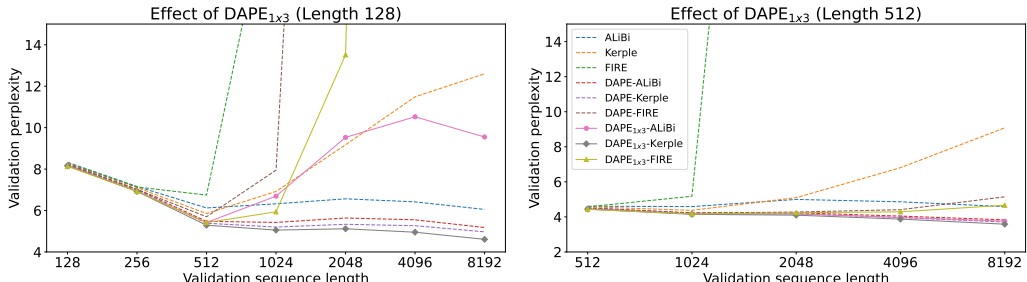

Figure 5: **The effect of DAPE$_{1\times3}$.** Whatever the baseline is ALiBi, Kerple or FIRE, the proposed DAPE$_{1\times3}$ can all improve their performance. The Figure 1 also proves that the proposed DAPE$_{1\times3}$ is effective for NoPE and RoPE.

**For Additive Positional Encoding, DAPE$_{1\times3}$ enhances performance within and beyond the training length.** As demonstrated in Figure 5, for varying additive positional encoding such as ALiBi, Kerple, and FIRE, their incorporations with DAPE$_{1\times3}$ (i.e., DAPE$_{1\times3}$-ALiBi, DAPE$_{1\times3}$-Kerple, and DAPE$_{1\times3}$-FIRE) consistently improve performance, while DAPE$_{1\times3}$-ALiBi may needs longer training length to achieve better performance than DAPE-ALiBi. Furthermore, regardless of the specific additive positional encoding used, the proposed DAPE$_{1\times3}$ (configured with a kernel size of $1 \times 3$) outperforms the standard DAPE method (which employs a kernel size of $1 \times 1$). Also, as shown in Figure 1, DAPE$_{1\times3}$ imrpoves the performance of NoPE, both within and beyond the training length These results highlight the robustness and scalability of DAPE$_{1\times3}$, suggesting its broad applicability in enhancing additive positional encoding frameworks.

**For Non-Additive Positional Encoding, DAPE$_{1\times3}$ also improves performance within and beyond the training length.** As illustrated in Figure 1, DAPE$_{1\times3}$ enhances the performance of RoPE, both within and beyond the training length. In contrast, naive DAPE reduces the performance of RoPE, with training lengths of 128 and 512. This indicates that the proposed DAPE$_{1\times3}$ is a versatile and widely applicable method with the potential to be applied to various position encoding techniques on the language modeling task.

## 4.5 The Performance of DAPE$_{1\times3}$ with Information Leakage

**The DAPE$_{1\times3}$ can utilize attention data, which is supported by almost zero loss (perplexity is 1) under information leakage.** To prevent the information leakage, we use the $torch.tril$ before DAPE$_{1\times3}$ to make the attention score lower-triangular matrix. For the cheating version, we do not use the `torch.tril`. As shown in Figure 6, whatever DAPE$_{1\times3}$-ALiBi, DAPE$_{1\times3}$-Kerple or DAPE$_{1\times3}$-FIRE, their cheating version can all achieve about zero loss within evaluation length 1024. Furthermore, the DAPE$_{1\times3}$-Kerple can even aachievezero loss when the evaluation length is extended to 8096. This suggest that the proposed DAPE$_{1\times3}$ can really realize and utilize the information of attention score.

## 4.6 Compare DAPE and DAPE$_{1\times3}$ with Approximate Computational Cost

**DAPE$_{1\times3}$ achieves even better performance at a lower computational cost.** As shown in Appendix E, when the training length is set to 128, DAPE$_{1\times3}$-Kerple with $D_{DAPE}$ as 10 achieves a perplexity (ppl) of 8.16 at an evaluation length of 128 and 4.74 at an evaluation length of 8192. This performance is notably better than that of DAPE-Kerple with $D_{DAPE}$ as 64, which achieves

perplexities of 8.21 and 4.87, respectively. Moreover, when the training length is extended to 512 and the evaluation length is smaller or equal to 4096, $DAPE_{1\times3}$-Kerple with $D_{DAPE}$ as 10 continues to surpass the performance of DAPE-Kerple with $D_{DAPE}$ as 64. Also, $DAPE_{1\times3}$-Kerple with $D_{DAPE}$ as 21 always achieves better performance than DAPE-Kerple with $D_{DAPE}$ as 64. This demonstrates that $DAPE_{1\times3}$ not only maintains its performance advantage across different training lengths but also requires a lower computational cost.

### 4.7 THE PERFORMANCE WITH DIFFERENT KERNEL SIZES

**Different experiment settings may have different optimal kernel sizes.** Appendix F shows the performance of DAPE with various kernel sizes, including DAPE (equivalent to a $1 \times 1$ kernel size), $DAPE_{1\times3}$, $DAPE_{1\times5}$, and $DAPE_{1\times7}$. For the Arxiv dataset, larger kernel sizes consistently achieve better performance, evaluating with training lengths of 128 or 512. However, for the Books3 dataset, $DAPE_{1\times3}$ performs best when the training length is 128 and evaluated at 8192, whereas $DAPE_{1\times5}$ performs best at the same evaluation level when the training length is 512. These results suggest that the optimal kernel size may vary depending on the experimental setting, ranging from $1 \times 1$ to larger kernel sizes. Although larger kernel sizes contribute to stronger expressiveness from intuition, we conjecture that the performance degradation for overly large kernel sizes results from optimization challenges.

### 4.8 THE PERFORMANCE ON CHE BENCHMARK WITH ACCURACY EVALUATION METRICS

**Different tasks have different optimal kernel sizes, as shown in Appendix G and Appendix F.** For example, on MISSING DUPLICATE task, the $DAPE_{1\times3}$-Kerple improves the 87.57 of DAPE-Kerple to 99.65. However, on the STACK MANIPULATIONtask, the $DAPE_{1\times3}$-Kerple decreases the 72.04 of DAPE-Kerple to 68.18. Also, as shown in Appendix F, the larger kernel size does not always lead to better performance. Overall, larger kernel size provides a potential way to improve the Transformer length extrapolation performance, and we usually could find a suitable kernel size (ranging from 1×1 to larger kernel sizes) to achieve better performance than without further processing attention score.

**The large kernel size performance improvement is related to the baseline bias matrix.** As shown in Appendix G, the best performance is usually achieved by further processing attention scores via kernel size 1 or 3. Moreover, on 11 permutation-variant tasks, the $DAPE_{1\times3}$-Kerple achieves better performance on 8 of 11 tasks compared to Kerple. And the $DAPE_{1\times3}$-FIRE achieves better performance on 6 of 11 tasks compared to FIRE. This suggests that the large kernel size performance improvement is related to the baseline bias matrix.

### 4.9 THE TIME COST

**As the model size increases, the additional computational cost ratio gradually decreases.** As shown in Appendix H, when the model size is 350M, the time cost for Kerple is 189.91 ms, while DAPE-Kerple takes 224.22 ms, and $DAPE_{1\times3}$-Kerple requires 252.84 ms. Compared to $DAPE_{1\times3}$-Kerple, the time cost ratios for Kerple and DAPE-Kerple are 0.7511 and 0.8868, respectively. As the model size increases from 350M to 2.7B and 6.7B, the time cost ratio for Kerple rises from 0.7511 to 0.8205 and 0.8918, respectively. Similarly, the time cost ratio for DAPE-Kerple increases from 0.8868 to 0.9361 and 0.9677. Therefore, as the model size increases, the time cost ratio also increases, indicating that the additional computational cost decreases progressively.

## 5 CONCLUSION

In this paper, we point out that the key of Transformer length extrapolation is the better and more accurate attention score. Therefore, we develop and analyze $DAPE_{1\times3}$ by processing the attention score as feature maps via convolution operation. Theoretically, we show that the associative recall tasks, which account for the most perplexity scores, can be realized by the proposed Transformer with convolution, in contrast to the vanilla Transformer. We conducted comprehensive experiments on Arxiv, Books3, and CHE to validate the effectiveness of the proposed method, where the proposed method exhibits significant superiority.

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

# A ∆ PERPLEXITY FOR LENGTH EXTRAPOLATION EVALUATION

Table 1: The $\Delta P$ on Book dataset with training length 512, compared to baselines.

| Method | RoPE | ALiBi | Kerple | DAPE-Kerple | DAPE$_{1 \times 3}$-Kerple |
|---|---|---|---|---|---|
| $P(M(x_{512}), T_{train})$ | 19.74 | 20.04 | 19.83 | 19.25 | 18.95 |
| $P(M(x_{1024}), T_{train})$ | 261.39 | 19.74 | 19.19 | 18.28 | 17.92 |
| $P(M(x_{1024}[-T_{train}]:), T_{train})$ | 19.51 | 19.79 | 19.58 | 19.03 | 18.74 |
| $\Delta P_{1024}$ | -241.88 | 0.05 | 0.39 | 0.75 | 0.82 |
| $P(M(x_{2048}), T_{train})$ | 411.23 | 20.17 | 20.48 | 17.20 | 16.79 |
| $P(M(x_{2048}[-T_{train}]:), T_{train})$ | 18.74 | 19.03 | 19.84 | 18.28 | 18.01 |
| $\Delta P_{2048}$ | -392.49 | -1.14 | -0.64 | 1.08 | 1.22 |
| $P(M(x_{4096}), T_{train})$ | 635.80 | 20.50 | 28.33 | 17.58 | 17.05 |
| $P(M(x_{4096}[-T_{train}]:), T_{train})$ | 19.11 | 19.35 | 19.07 | 18.59 | 18.19 |
| $\Delta P_{4096}$ | -616.69 | -1.15 | -9.26 | 1.01 | 1.14 |
| $P(M(x_{8192}), T_{train})$ | 762.86 | 21.30 | 40.94 | 17.85 | 17.20 |
| $P(M(x_{8192}[-T_{train}]:), T_{train})$ | 19.78 | 20.02 | 19.85 | 19.38 | 18.98 |
| $\Delta P_{8192}$ | -743.08 | -1.28 | -21.09 | 1.53 | 1.78 |

In this discussion, we explore how to effectively use perplexity as a metric, incorporating concepts of information gain and entropy. Let $L(\cdot)$ represent the process for calculating loss, and $M(x)$ denote the logit output generated by the model after processing an input sequence $x$. For evaluating model performance, we define $P(M(x), K)$ as follows:

1. Process the entire sequence $x$ using $M(x)$.

2. Compute the perplexity on the last $K$ tokens of the sequence.

To interpret information gain, we consider the training sequence length $T_{\text{train}}$. Given an input $x$, we calculate the change in loss/perplexity, $\Delta P$, as:

$$\Delta P = P(M(x[-T_{\text{train}}:]), T_{\text{train}}) - P(M(x), T_{\text{train}}) \tag{6}$$

The term $\Delta P$ provides insights into the model's information gain relative to local and global context, allowing us to quantify entropy in terms of model uncertainty reduction. We interpret $\Delta L$ as follows:

- When $\Delta P = 0$: The model's information gain from the full sequence is negligible, indicating an entropy level comparable to local attention (e.g., models like ALiBi when the evaluation length is 1024). This suggests the model does not leverage context beyond a limited range.

- When $\Delta P < 0$: Processing the entire sequence increases entropy, resulting in worse performance than focusing only on the last $T_{\text{train}}$ tokens. This implies negative information gain and limited extrapolation capability (e.g. such as RoPE), as the model may overfit to recent tokens without capturing broader context effectively.

- When $\Delta P > 0$: The model benefits from the information within $x[: T_{\text{train}}]$, achieving a reduction in entropy that reflects positive information gain. This suggests the model leverages contextual information beyond the training sequence, indicating extrapolation capability.

- Our suggestion of bias matrix. The Kerple is a good choice for almost all settings, the FIRE may need longer training length/tokens to present its ability, and do not use ALiBi unless necessary . It is easy to train Kerple, as Kerple usually has few trainable parameters compared to FIRE. If you do not know which one to use, directly use Kerple. FIRE may have better performance, but may need longer training length (diverges at 128 but works well at 512, with DAPE). FIRE $b(i, j) = f_\theta \left( \frac{\psi(i-j)}{\psi(\max\{L, i\})} \right)$ so that we may need longer training length or more training tokens to well-train the neural network $f_\theta$. Do not use ALiBi unless necessary. The ALiBi will quickly become local attention as the sequence length increases.

By examining $\Delta P$, we can evaluate the model's ability to reduce entropy and gain information from extended sequences, providing a measure of its extrapolative power.

## B   COMPARE WITH BASELINE ON ARXIV DATASET WITH TRAINING LENGTH 1024

Table 2: The performance (ppl) on Arxiv dataset with training length 1024, compared to baselines.

| Method | 1024 | 2048 | 4096 | 8192 |
|---|---|---|---|---|
| NoPE (Kazemnejad et al., 2024) | 4.16 | 42.27 | 1854.73 | 17167.32 |
| RoPE (Su et al., 2024b) | 4.07 | 86.20 | 237.67 | 256.12 |
| T5's bias (Raffel et al., 2020) | 4.03 | 4.28 | 13.07 | 79.55 |
| ALiBi (Press et al., 2021) | 4.09 | 4.53 | 4.45 | 4.22 |
| Kerple (Chi et al., 2022) | 4.06 | 4.09 | 4.68 | 6.951 |
| FIRE (Li et al., 2023c) | 4.06 | 9.21 | 236.18 | 440.60 |
| DAPE-Kerple (Zheng et al., 2024) | 3.98 | 3.91 | 3.68 | 3.41 |
| $\text{DAPE}_{1\times3}$-Kerple | 3.93 | 3.86 | 3.61 | 3.37 |

## C   LARGE MODEL SIZE

Table 3: The performance (ppl) under large model size 2.7B on Books3 dataset.

| Method | 512 | 1024 | 2048 | 4096 |
|---|---|---|---|---|
| RoPE | 21.01 | 25.00 | 48.13 | 160.59 |
| T5's bias | 21.10 | 21.88 | 23.59 | 33.23 |
| Kerple | 21.14 | 22.08 | 23.38 | 27.21 |
| DAPE-Kerple | 20.52 | 21.01 | 20.23 | 19.67 |
| $\text{DAPE}_{1\times3}$-Kerple (kernel size 1x3) | 20.16 | 20.54 | 19.80 | 19.02 |

# D THE PERFORMANCE OF DAPE$_{1 \times 3}$ WITH INFORMATION LEAKAGE

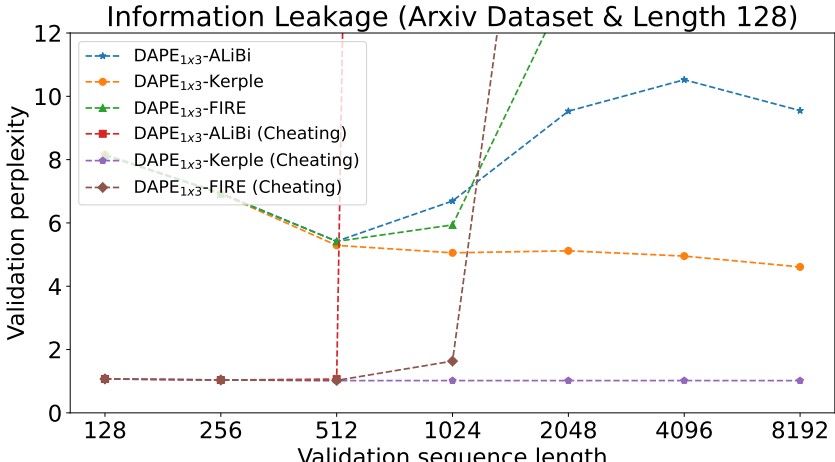

Figure 6: **Result with information leakage.**

# E COMPARE DAPE AND DAPE$_{1 \times 3}$ WITH APPROXIMATE COMPUTATIONAL COST

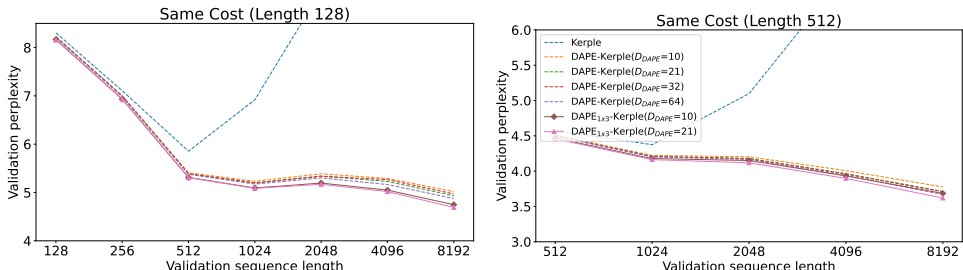

Figure 7: **Compare DAPE$_{1 \times 3}$ and DAPE with the approximately same cost on Arxiv Dataset.** We compare the DAPE$_{1 \times 3}$ and DAPE with approximate cost and different $D_{\text{DAPE}}$. As the kernel size of is DAPE$_{1 \times 3}$ $1 \times 3$, the proposed DAPE$_{1 \times 3}$ is the triple computation cost of DAPE, with the same $D_{\text{DAPE}}$.

# F  THE PERFORMANCE WITH DIFFERENT KERNEL SIZE

Table 4: The performance with different kernel sizes, with training length 128 and evaluation from length 128 to 8192. For different datasets and training length, the optimal kernel size may not always be the largest one, especially when the evaluation length is larger.

| Dataset | Method | 128 | 256 | 512 | 1024 | 2048 | 4096 | 8192 |
|---------|--------|-----|-----|-----|------|------|------|------|
| Arxiv | Kerple | 8.30 | 7.10 | 5.85 | 6.91 | 9.17 | 11.48 | 12.59 |
| | DAPE-Kerple (Kernel Size 1x1) | 8.21 | 6.98 | 5.38 | 5.20 | 5.33 | 5.26 | 4.97 |
| | DAPE$_{1\times3}$-Kerple (Kernel Size 1x3) | 8.15 | 6.92 | 5.29 | 5.05 | 5.11 | 4.95 | 4.60 |
| | DAPE$_{1\times5}$-Kerple (Kernel Size 1x5) | 8.13 | 6.91 | 5.27 | 5.04 | 5.10 | 4.91 | 4.57 |
| | DAPE$_{1\times7}$-Kerple (Kernel Size 1x7) | **8.12** | **6.89** | **5.26** | **5.02** | **5.09** | **4.91** | **4.57** |
| Books3 | Kerple | 32.10 | 29.09 | 28.10 | 35.75 | 44.68 | 56.39 | 66.23 |
| | DAPE-Kerple (Kernel Size 1x1) | 31.49 | 28.27 | 24.93 | 24.31 | 23.34 | 24.38 | 25.01 |
| | DAPE$_{1\times3}$-Kerple (Kernel Size 1x3) | 31.07 | 27.81 | 24.38 | 23.57 | 22.40 | 23.19 | **23.52** |
| | DAPE$_{1\times5}$-Kerple (Kernel Size 1x5) | 31.02 | 27.79 | 24.36 | 23.57 | 22.41 | 23.32 | 23.71 |
| | DAPE$_{1\times7}$-Kerple (Kernel Size 1x7) | **30.98** | **27.76** | **24.31** | **23.47** | **22.30** | **23.00** | 23.57 |

Table 5: The performance with different kernel size, with training length 512 and evaluation from length 512 to 8192. For different datasets and training length, the optimal kernel size may not always be the largest one, especially when the evaluation length is larger.

| Dataset | Method | 512 | 1024 | 2048 | 4096 | 8192 |
|---------|--------|-----|------|------|------|------|
| Arxiv | Kerple | 4.57 | 4.37 | 5.09 | 6.80 | 9.08 |
| | DAPE-Kerple (Kernel Size 1x1) | 4.49 | 4.20 | 4.17 | 3.95 | 3.70 |
| | DAPE$_{1\times3}$-Kerple (Kernel Size 1x3) | 4.44 | 4.14 | 4.09 | 3.87 | 3.58 |
| | DAPE$_{1\times5}$-Kerple (Kernel Size 1x5) | 4.44 | 4.14 | 4.10 | 3.85 | 3.59 |
| | DAPE$_{1\times7}$-Kerple (Kernel Size 1x7) | **4.43** | **4.13** | **4.08** | **3.85** | **3.57** |
| Books3 | Kerple | 19.83 | 19.19 | 20.48 | 28.33 | 40.94 |
| | DAPE-Kerple (Kernel Size 1x1) | 19.25 | 18.28 | 17.20 | 17.58 | 17.85 |
| | DAPE$_{1\times3}$-Kerple (Kernel Size 1x3) | 18.95 | 17.92 | 16.79 | 17.05 | 17.20 |
| | DAPE$_{1\times5}$-Kerple (Kernel Size 1x5) | 18.89 | 17.87 | 16.76 | 17.09 | **17.10** |
| | DAPE$_{1\times7}$-Kerple (Kernel Size 1x7) | **18.86** | **17.82** | **16.70** | **17.01** | 17.16 |

# G    THE PERFORMANCE OF DAPE$_{1\times3}$ ON CHE BENCHMARK

Table 6: Train on length 40 with 200k steps, and test from lengths 41 to 500. The random accuracy is 50%, except for MODULAR ARITHMETIC (SIMPLE), CYCLE NAVIGATION, BUCKET SORT, SOLVE EQUATION and MODULAR ARITHMETIC, where it is 20%. ††† denotes permutation-invariant tasks, which are expected to be solved without positional information. The dataset comes from Choromanski et al. (2021), with experiment setting from Randomized PE(Ruoss et al., 2023).

| Level | Task | Baseline | | | | | DAPE (Kernel Size 1) | | | DAPE (Kernel Size 3) | | |
|---|---|---|---|---|---|---|---|---|---|---|---|---|
| | | RoPE | Relative | ALiBi | Kerple | FIRE | ALiBi | Kerple | FIRE | ALiBi | Kerple | FIRE |
| R | EVEN PAIRS | 99.98 | 96.60 | 73.52 | 57.50 | 73.86 | 99.99 | 99.58 | **100** | 99.99 | **100** | **100** |
| | MODULAR ARITHMETIC (SIMPLE) | 21.35 | 20.84 | 20.02 | 21.79 | 21.09 | 23.58 | **24.47** | 24.46 | 21.48 | 23.90 | 23.43 |
| | PARITY CHECK††† | 50.05 | 50.09 | 50.09 | 50.07 | 50.97 | 50.30 | 50.07 | 50.04 | 50.13 | **52.51** | 50.11 |
| | CYCLE NAVIGATION††† | 27.63 | 26.95 | 24.64 | 29.47 | 28.41 | 22.99 | **34.53** | 27.54 | 24.43 | 24.32 | 24.34 |
| DCF | STACK MANIPULATION | 61.49 | 64.73 | 66.42 | 66.93 | 69.33 | 68.18 | **72.04** | 70.90 | 58.90 | 68.18 | 60.90 |
| | REVERSE STRING | 65.23 | 65.59 | 71.09 | 71.54 | 65.89 | 73.37 | 70.74 | 76.40 | 56.61 | **81.84** | 70.11 |
| | MODULAR ARITHMETIC | 31.25 | 31.74 | 30.56 | 24.79 | 30.92 | 31.34 | **32.37** | 31.50 | 29.46 | 26.13 | 27.00 |
| | SOLVE EQUATION | 21.85 | 22.93 | 19.92 | 21.15 | 22.06 | 20.03 | 22.49 | 22.42 | 20.26 | **23.95** | 23.62 |
| CS | DUPLICATE STRING | 64.97 | 67.66 | 65.13 | 66.72 | 69.03 | 70.84 | **72.95** | 72.71 | 52.96 | 57.03 | 66.01 |
| | MISSING DUPLICATE | 63.37 | 72.34 | 74.21 | 79.06 | 79.27 | 83.41 | 87.57 | 89.17 | 59.33 | **99.65** | 74.83 |
| | ODDS FIRST | 61.00 | 61.57 | 59.88 | 62.59 | 63.28 | 63.78 | **67.08** | 66.34 | 57.35 | 56.87 | 56.57 |
| | BINARY ADDITION | 55.59 | 56.96 | 54.72 | 56.35 | 55.70 | 59.71 | **60.88** | 56.62 | 57.49 | 55.32 | 57.86 |
| | COMPUTE SQRT | 51.88 | 51.63 | 50.63 | 51.11 | 50.80 | 51.64 | 51.33 | **52.46** | 52.08 | 51.76 | 51.93 |
| | BUCKET SORT††† | 98.12 | 99.31 | 98.45 | 99.38 | **99.57** | 99.38 | 98.81 | 99.37 | 96.61 | 99.06 | 98.56 |

# H    DAPE$_{1\times3}$ TIME COST

Table 7: The time cost (millisecond) under different testing lengths, with $D_{\text{DAPE}}$ as 32 and default batch size 1, with training length 512.

| Method | 350M Total | Ratio | 2.7B Total | Ratio | 6.7B Total | Ratio |
|---|---|---|---|---|---|---|
| RoPE (Su et al., 2024b) | 210.01 | 0.8306 | 472.63 | 1.0472 | 635.57 | 0.8564 |
| T5's bias (Raffel et al., 2020) | 355.16 | 1.4046 | 537.62 | 1.1912 | 808.85 | 1.0899 |
| ALiBi (Press et al., 2021) | 172.60 | 0.6826 | 325.95 | 0.7222 | 596.77 | 0.8041 |
| Kerple (Chi et al., 2022) | 189.91 | 0.7511 | 370.32 | 0.8205 | 661.82 | 0.8918 |
| FIRE (Li et al., 2023c) | 248.13 | 0.9813 | 432.63 | 0.9586 | 797.68 | 1.0748 |
| DAPE-Kerple (Zheng et al., 2024) | 224.22 | 0.8868 | 422.48 | 0.9361 | 717.46 | 0.9667 |
| DAPE$_{1\times3}$-Kerple | 252.84 | 1.0000 | 451.29 | 1.0000 | 742.10 | 1.0000 |

# I    MODEL CONFIGURATION

All experiments are conducted on 8 GPUs. The 125M and 350M model configuration is the following.

Table 8: **Model Configurations.**

|                              | **125M**      | **350M**      |
|------------------------------|---------------|---------------|
| Training sequence length     | 512           | 512           |
| Batch size                   | $32 \times 8$ | $32 \times 8$ |
| Numer of iterations          | 50k           | 50k           |
| Dropout prob.                | 0.0           | 0.0           |
| Attention dropout prob.      | 0.0           | 0.0           |
| Attention head               | 12            | 16            |
| Feature dimension            | 768           | 1024          |
| Layer number                 | 12            | 24            |
| Optimizer                    | Adam          | Adam          |
| Optimizer parameter betas    | [0.9, 0.95]   | [0.9, 0.95]   |
| Learning rate                | $6e-4$        | $3e-4$        |
| Precision                    | float16       | float16       |

# J    DATA-ADAPTIVE RELATED POSITION ENCODING PERFORMANCE COMPARISON

Table 9: The performance comparison between data-related position encoding, with dataset Books3 and training length 128.

| **Method**                              | 128   | 256   | 512   | 1024  | 2048  | 4096  | 8192    |
|-----------------------------------------|-------|-------|-------|-------|-------|-------|---------|
| Transformer-XL                          | 31.57 | 28.49 | 26.07 | 26.98 | 27.90 | 32.76 | 41.12   |
| CoPE                                    | 31.61 | 28.41 | 25.79 | 27.96 | 33.80 | 54.08 | 90.66   |
| DAPE-Kerple (Kernel Size 1x1)           | 31.49 | 28.27 | 24.93 | 24.31 | 23.34 | 24.38 | 25.01   |
| DAPE$_{1 \times 3}$-Kerple (Kernel Size 1x3) | 31.07 | 27.81 | 24.38 | 23.57 | 22.40 | 23.19 | **23.52** |

# K  DAPE$_{1\times3}$ VISUALIZATION

The model is trained with DAPE$_{1\times3}$-Kerple on length 512. Compared to DAPE (Zheng et al., 2024), it seems that the DAPE$_{1\times3}$ presents a more obvious attention sink (Xiao et al., 2024d).

## K.1  VISUALIZATION ON LENGTH 512

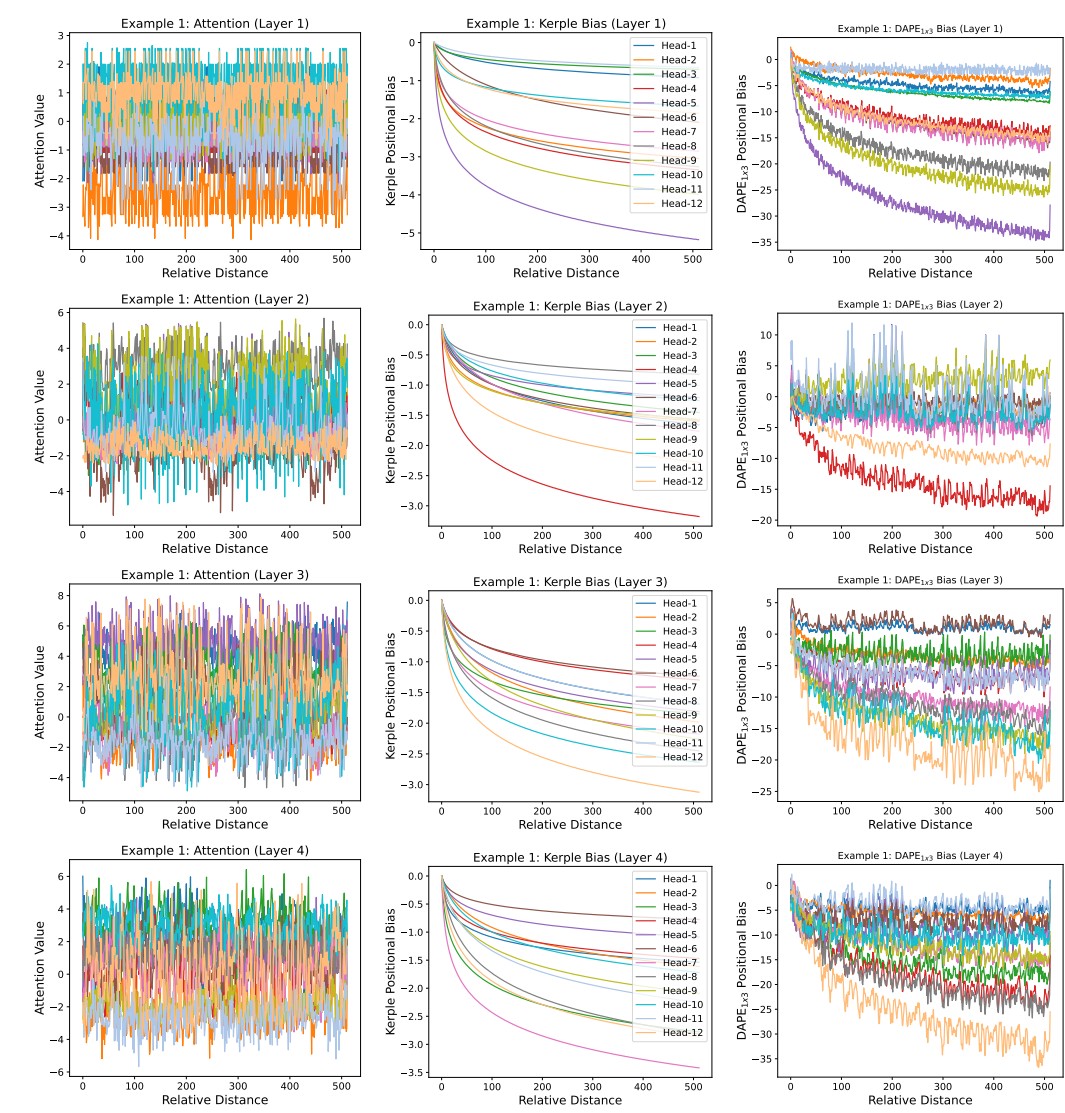

Figure 8: **Evaluation Length 512 Example 1: Part 1. From Left to Right: (1) The Attention is $XW_Q(XW_K)^\top$; (2) The Kerple bias is $B$; (3) The DAPE$_{1\times3}$ (with Kerple) bias is $f(XW_Q(XW_K)^\top, B)$.**

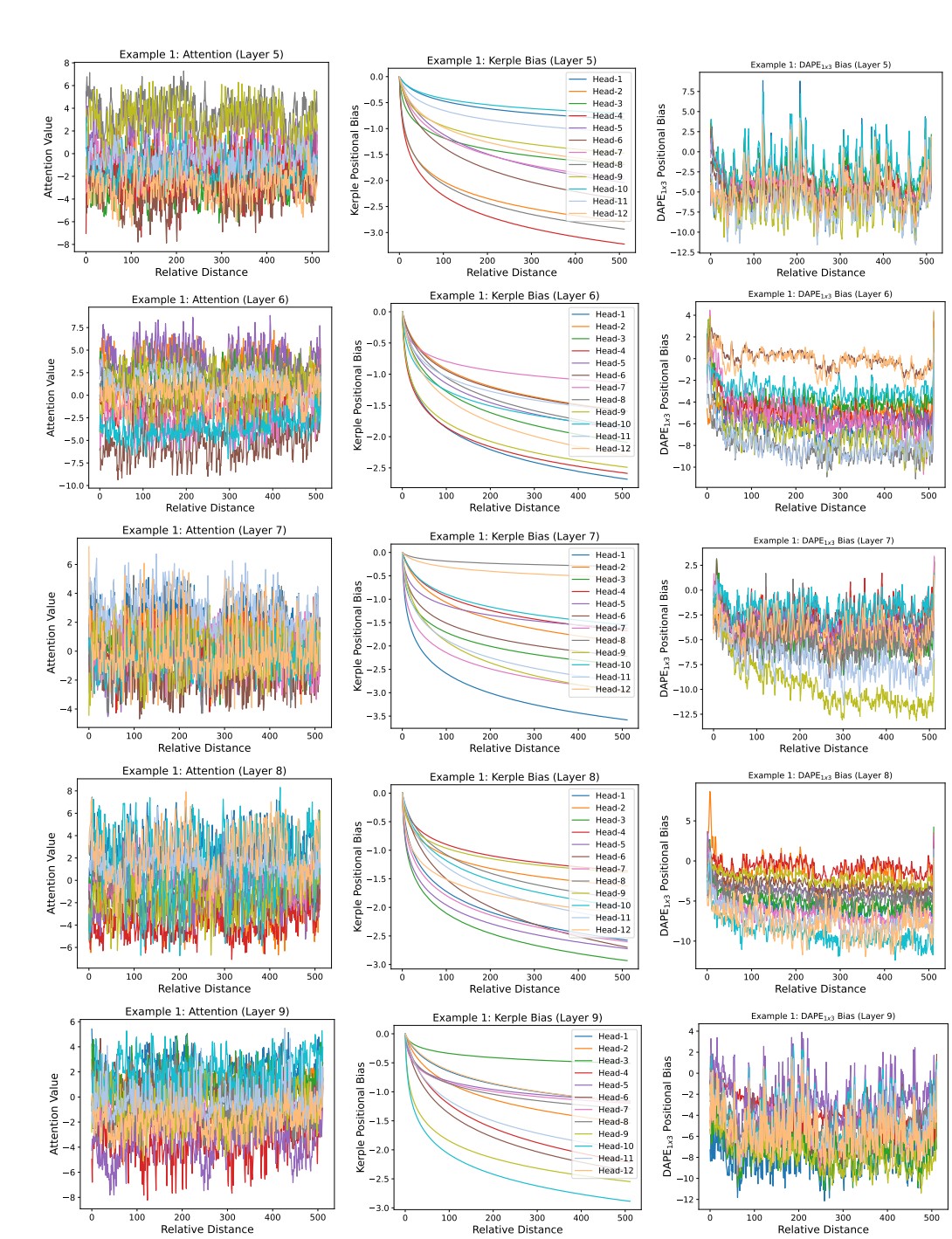

Figure 9: **Evaluation Length 512 Example 1: Part 2. From Left to Right: (1) The Attention is $\boldsymbol{X}\boldsymbol{W}_Q(\boldsymbol{X}\boldsymbol{W}_K)^\top$; (2) The Kerple bias is $\boldsymbol{B}$; (3) The DAPE$_{1\times3}$ (with Kerple) bias is $f(\boldsymbol{X}\boldsymbol{W}_Q(\boldsymbol{X}\boldsymbol{W}_K)^\top, \boldsymbol{B})$.**

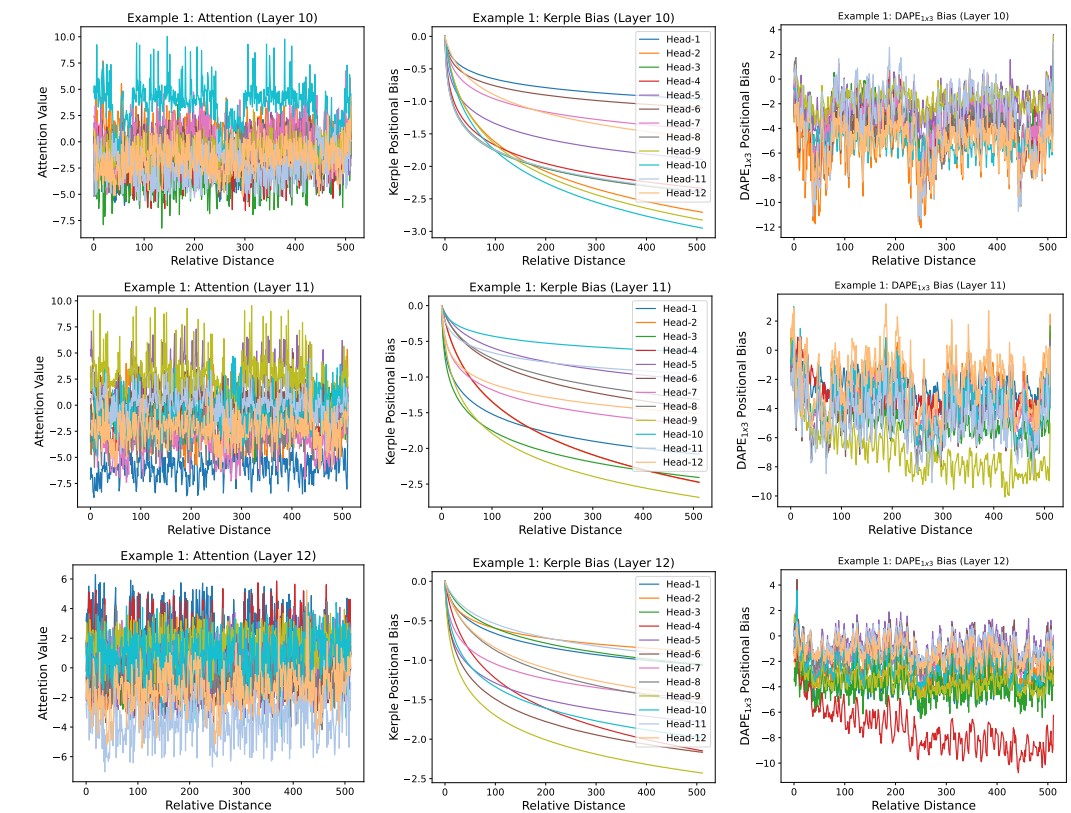

Figure 10: **Evaluation Length 512 Example 1: Part 3. From Left to Right: (1) The Attention is $XW_Q(XW_K)^\top$; (2) The Kerple bias is $B$; (3) The DAPE$_{1\times3}$ (with Kerple) bias is** $f(XW_Q(XW_K)^\top, B)$**.**

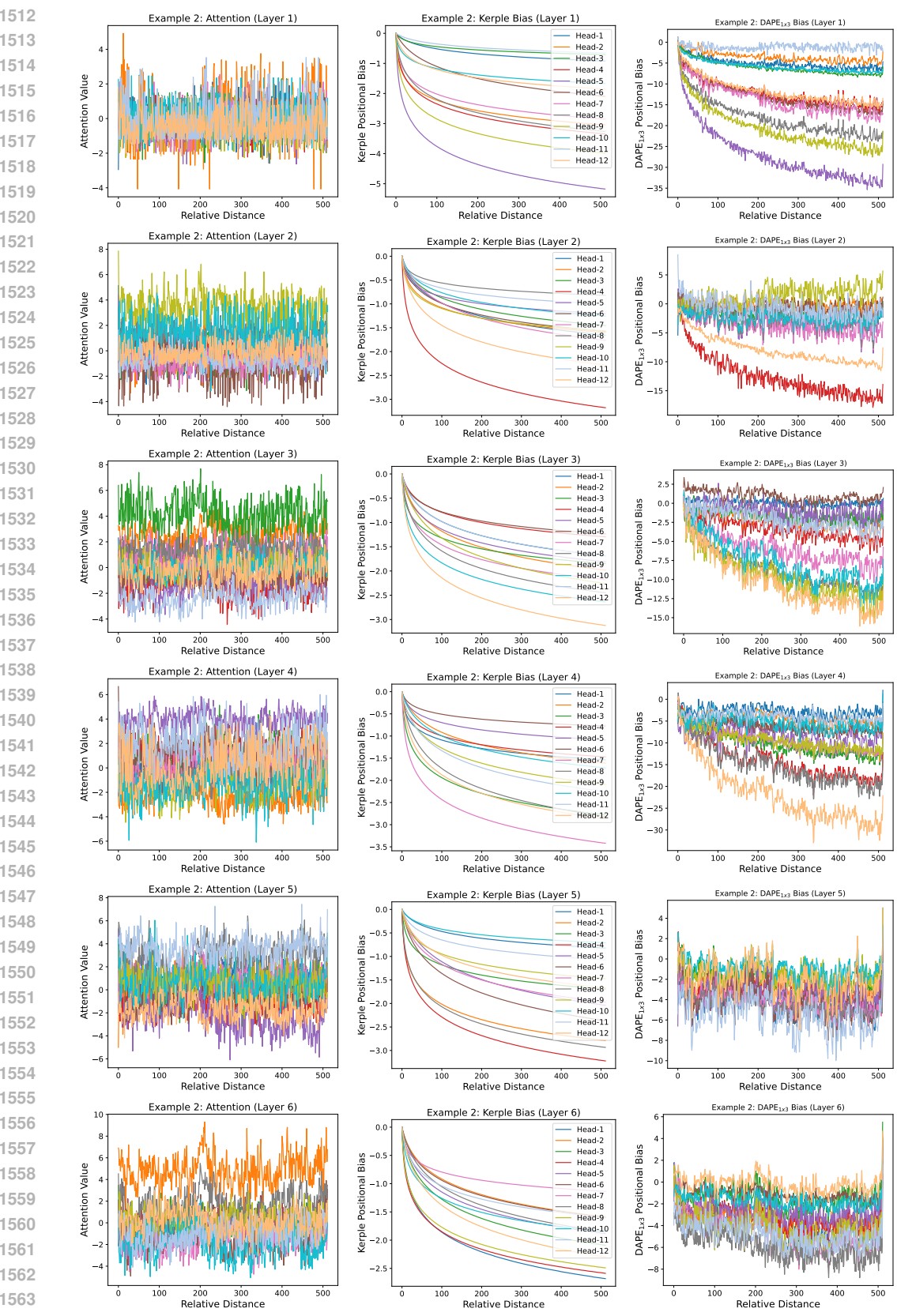

Figure 11: **Evaluation Length 512 Example 2: Part 1. From Left to Right: (1) The At-tention is $XW_Q(XW_K)^\top$; (2) The Kerple bias is $B$; (3) The DAPE$_{1\times3}$ (with Kerple) bias is $f(XW_Q(XW_K)^\top, B)$.**

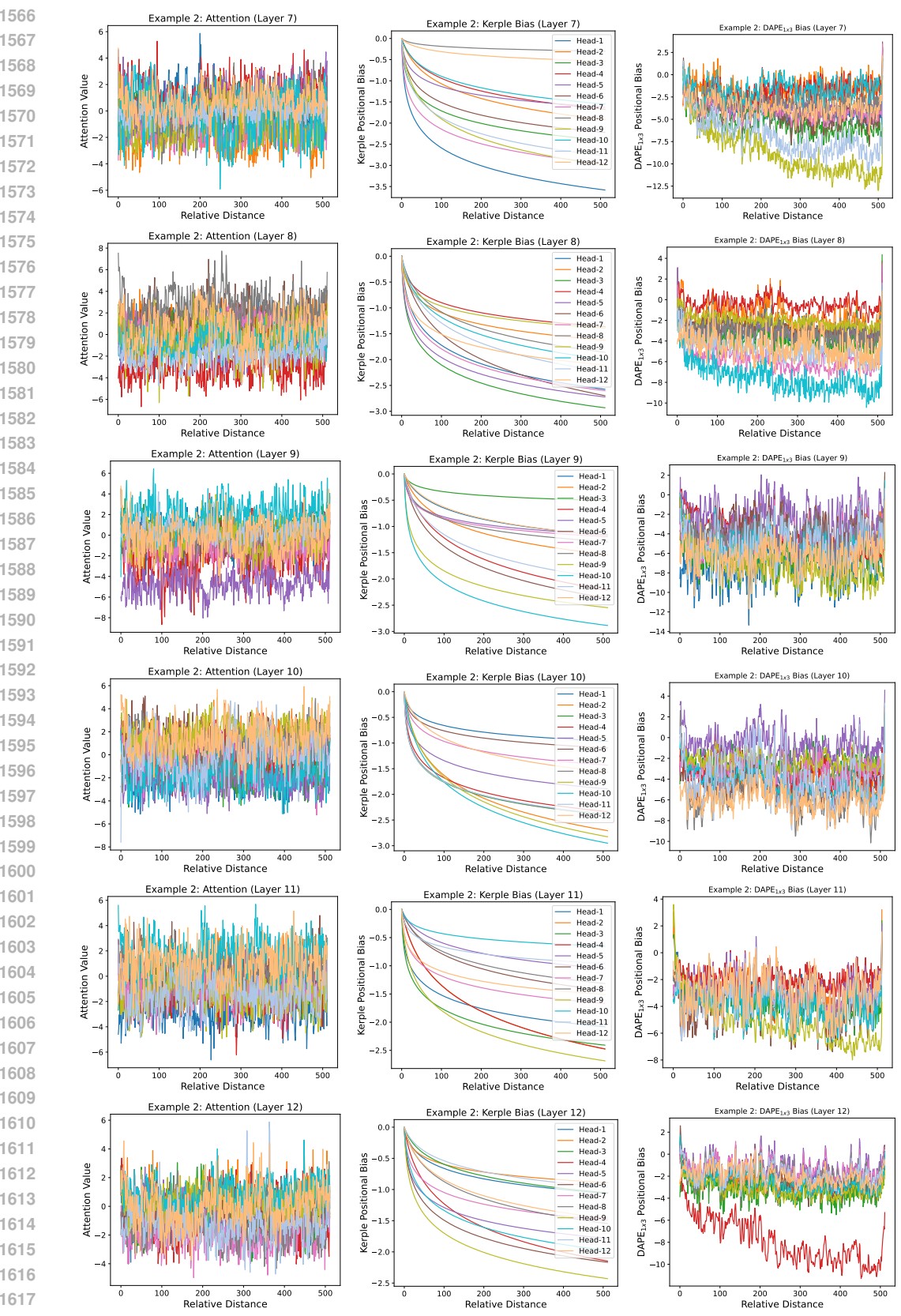

Figure 12: **Evaluation Length 512 Example 2: Part 2. From Left to Right: (1) The Attention is $\boldsymbol{XW_Q(XW_K)}^\top$; (2) The Kerple bias is $\boldsymbol{B}$; (3) The DAPE$_{1\times3}$ (with Kerple) bias is $f(\boldsymbol{XW_Q(XW_K)}^\top, \boldsymbol{B})$.**

## K.2 VISUALIZATION ON LENGTH 2048

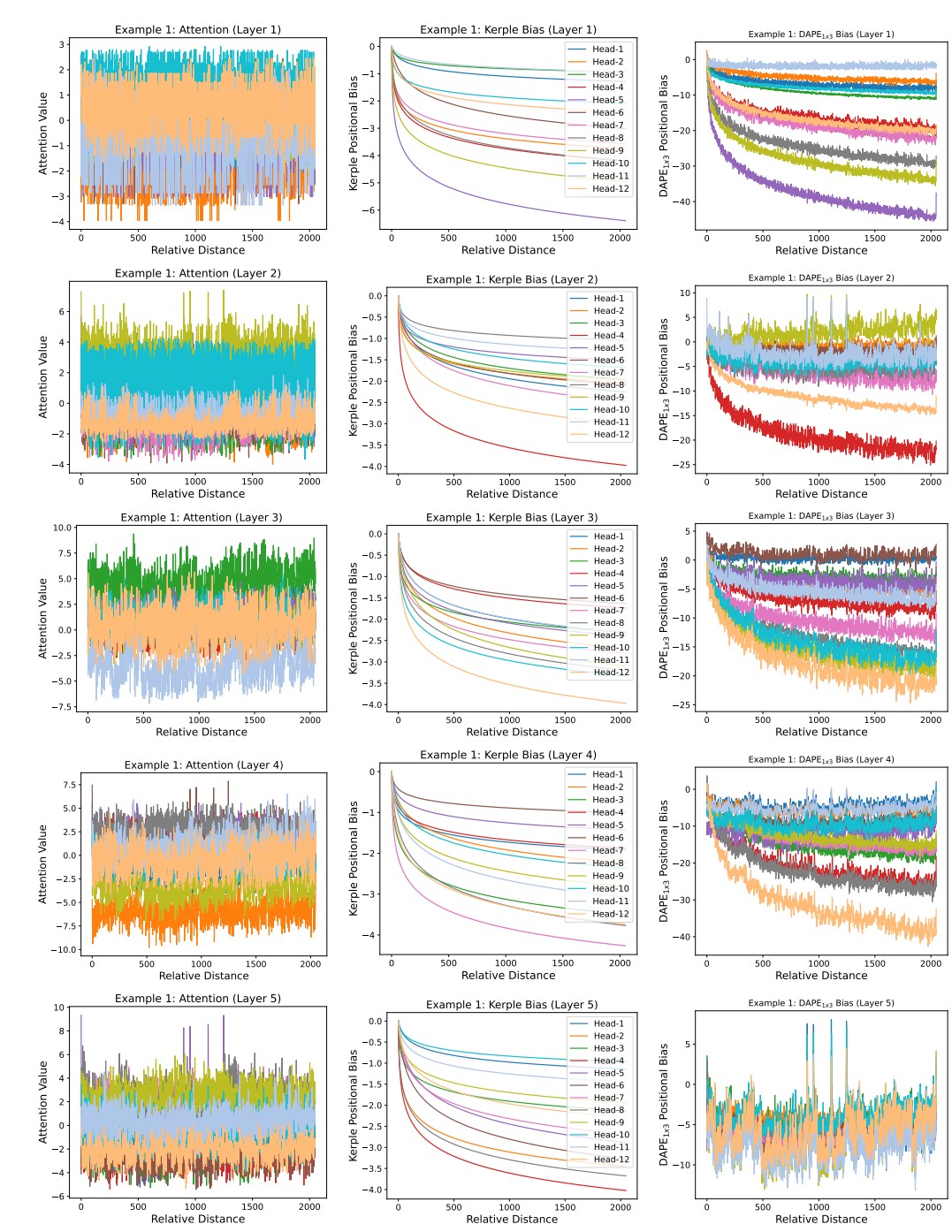

Figure 13: **Evaluation Length 2048 Example 1: Part 1. From Left to Right: (1) The Attention is $XW_Q(XW_K)^\top$; (2) The Kerple bias is $B$; (3) The DAPE$_{1\times3}$ (with Kerple) bias is $f(XW_Q(XW_K)^\top, B)$.**

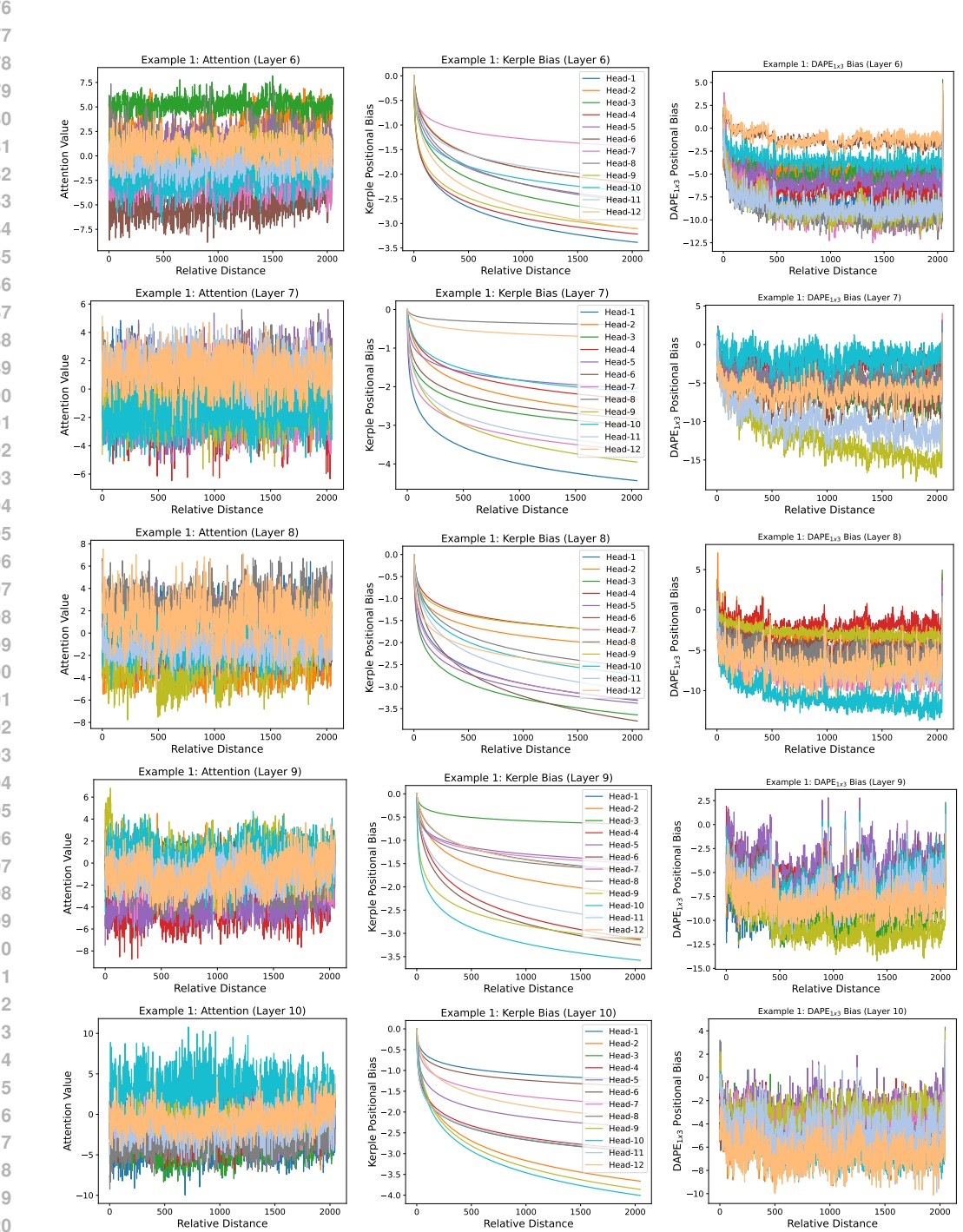

Figure 14: **Evaluation Length 2048 Example 1: Part 2. From Left to Right: (1) The Attention is $XW_Q(XW_K)^\top$; (2) The Kerple bias is $B$; (3) The DAPE$_{1\times3}$ (with Kerple) bias is $f(XW_Q(XW_K)^\top, B)$.**

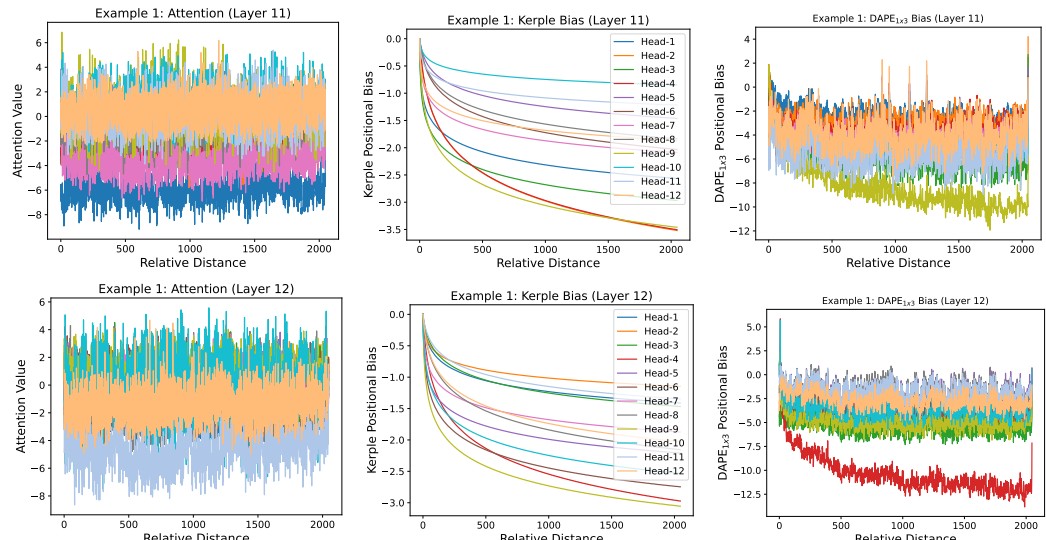

Figure 15: **Evaluation Length 2048 Example 1: Part 3. From Left to Right: (1) The Attention is $XW_Q(XW_K)^\top$; (2) The Kerple bias is $B$; (3) The DAPE$_{1\times3}$ (with Kerple) bias is $f(XW_Q(XW_K)^\top, B)$.**

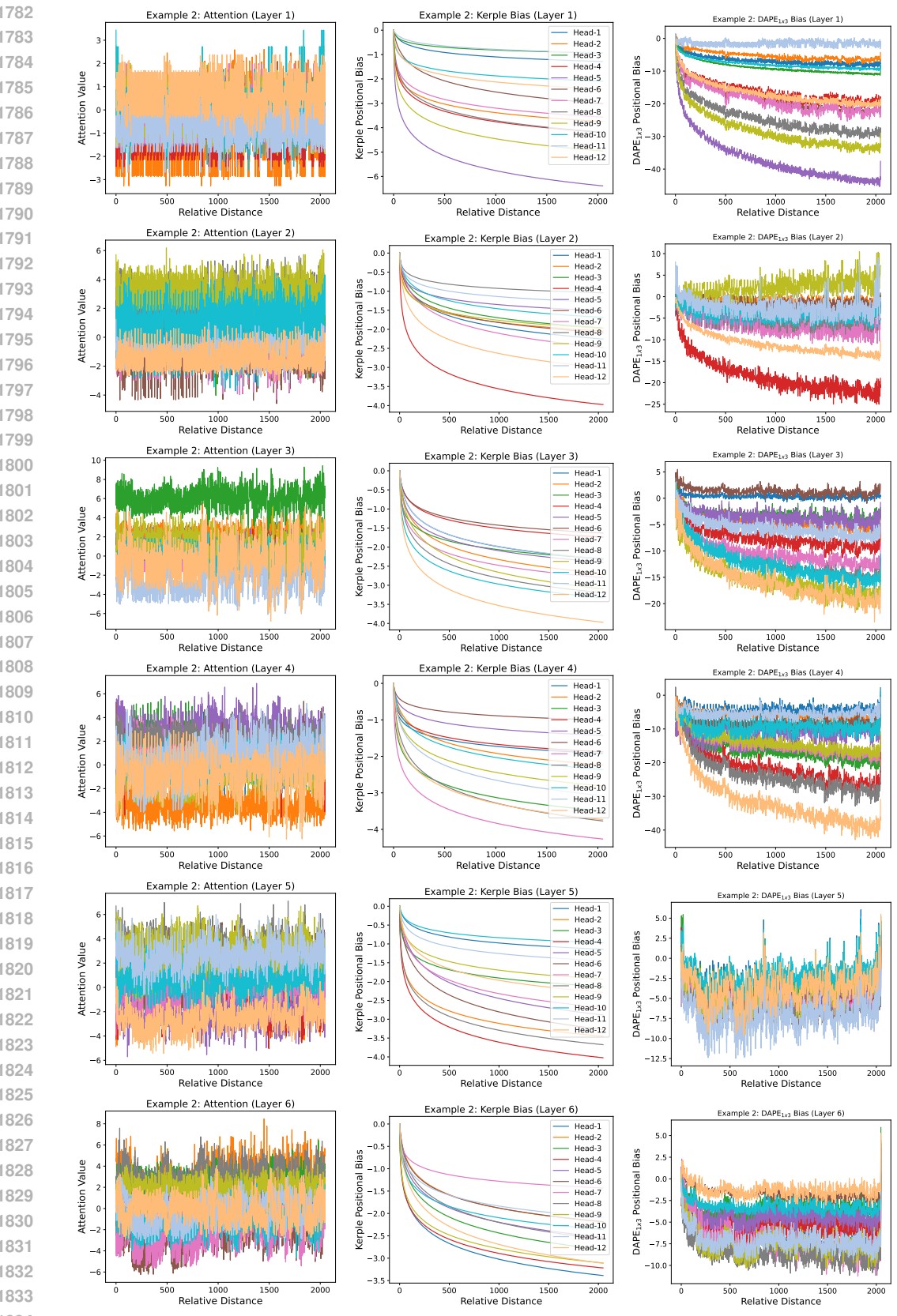

Figure 16: **Evaluation Length 2048 Example 2: Part 1. From Left to Right: (1) The Attention is $XW_Q(XW_K)^\top$; (2) The Kerple bias is $B$; (3) The DAPE$_{1\times3}$ (with Kerple) bias is $f(XW_Q(XW_K)^\top, B)$.**

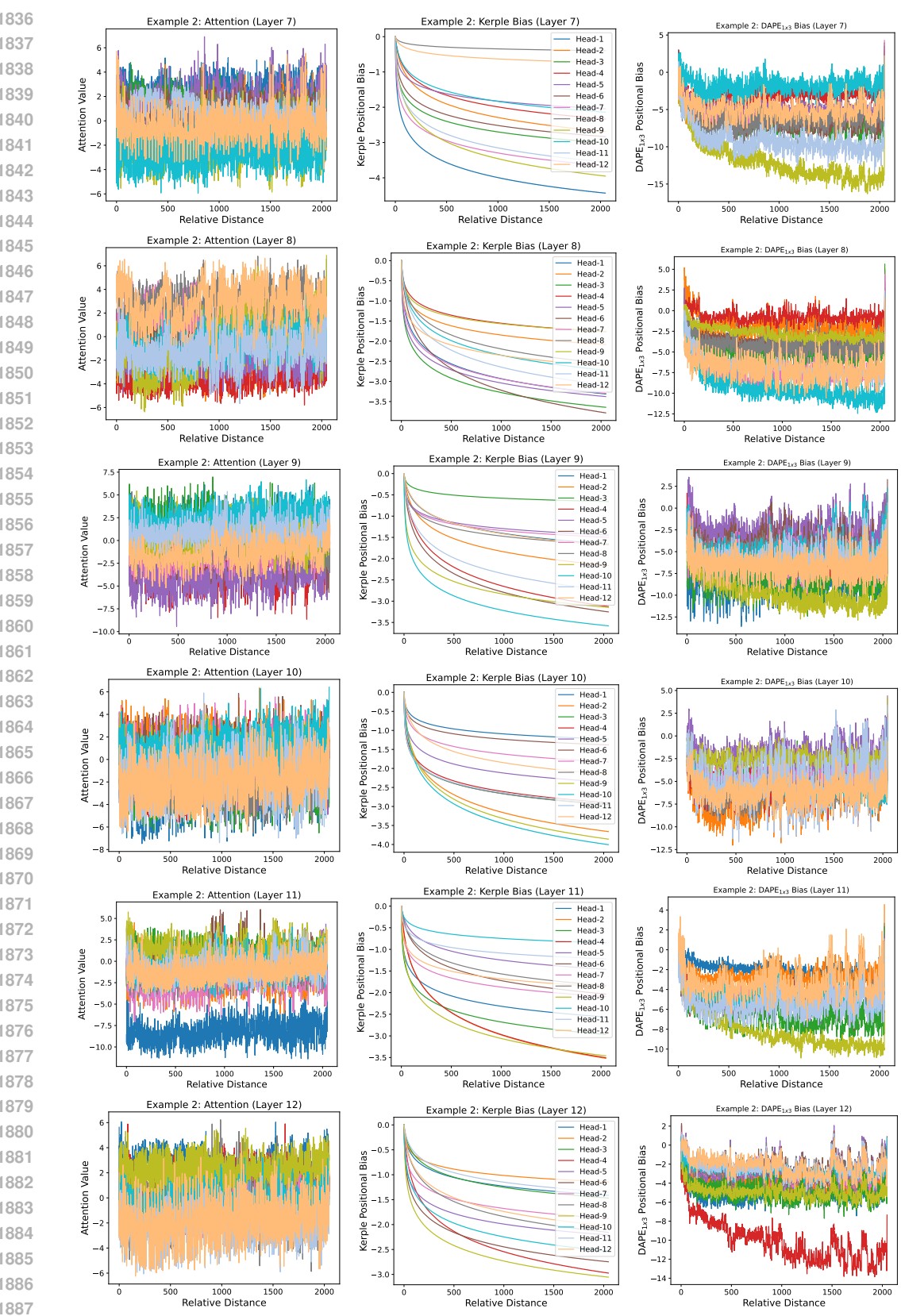

Figure 17: **Evaluation Length 2048 Example 2: Part 2. From Left to Right: (1) The Attention is $XW_Q(XW_K)^\top$; (2) The Kerple bias is $B$; (3) The DAPE$_{1\times3}$ (with Kerple) bias is $f(XW_Q(XW_K)^\top, B)$.**

### K.3 VISUALIZATION ON LENGTH 8192

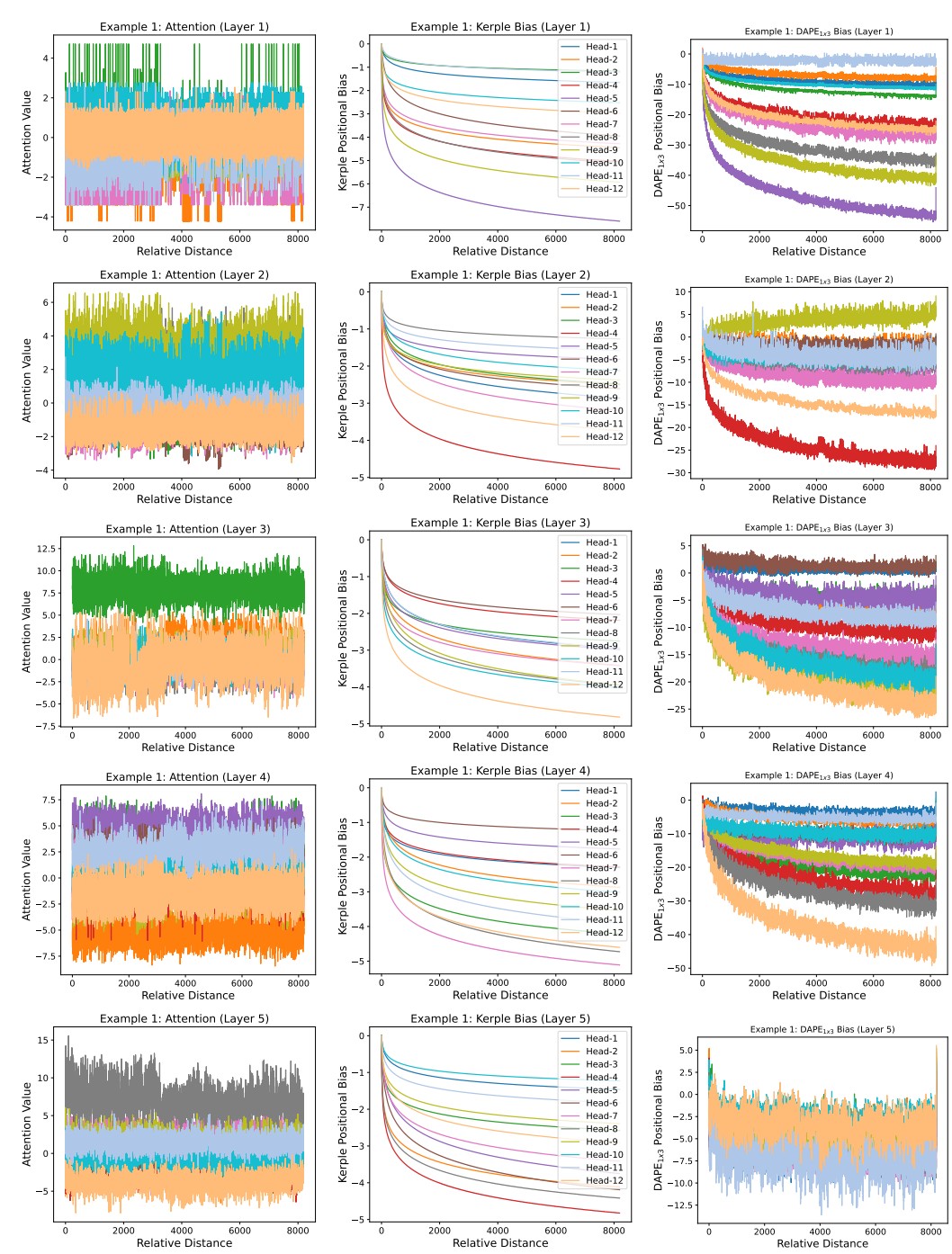

Figure 18: **Evaluation Length 8192 Example 1: Part 1. From Left to Right: (1) The Attention is $XW_Q(XW_K)^\top$; (2) The Kerple bias is $B$; (3) The DAPE$_{1\times3}$ (with Kerple) bias is $f(XW_Q(XW_K)^\top, B)$.**

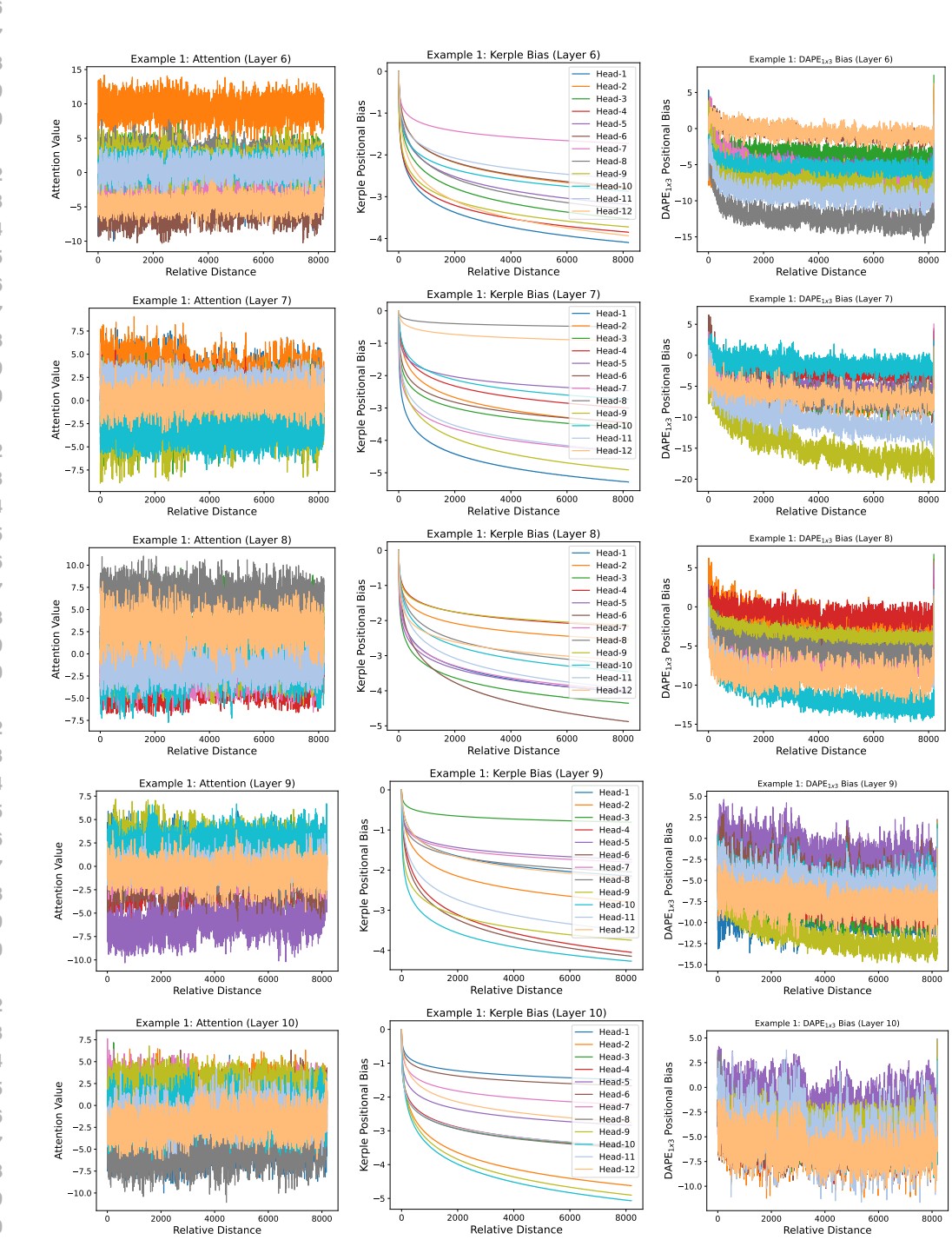

Figure 19: **Evaluation Length 8192 Example 1: Part 2. From Left to Right: (1) The Attention is $XW_Q(XW_K)^\top$; (2) The Kerple bias is $B$; (3) The DAPE$_{1\times 3}$ (with Kerple) bias is $f(XW_Q(XW_K)^\top, B)$.**

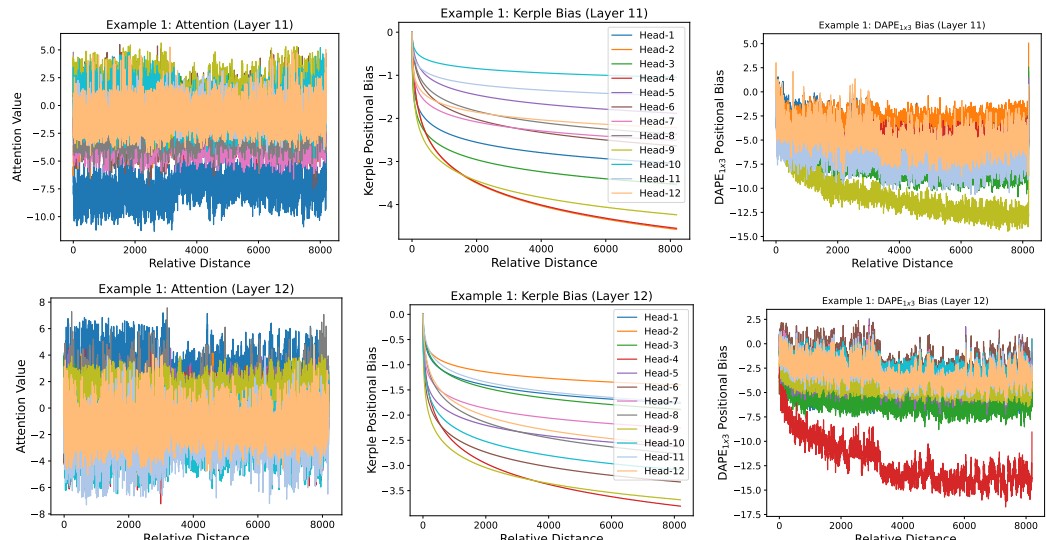

Figure 20: **Evaluation Length 8192 Example 1: Part 3. From Left to Right: (1) The Attention is $XW_Q(XW_K)^\top$; (2) The Kerple bias is $B$; (3) The DAPE$_{1\times3}$ (with Kerple) bias is** $f(XW_Q(XW_K)^\top, B)$.

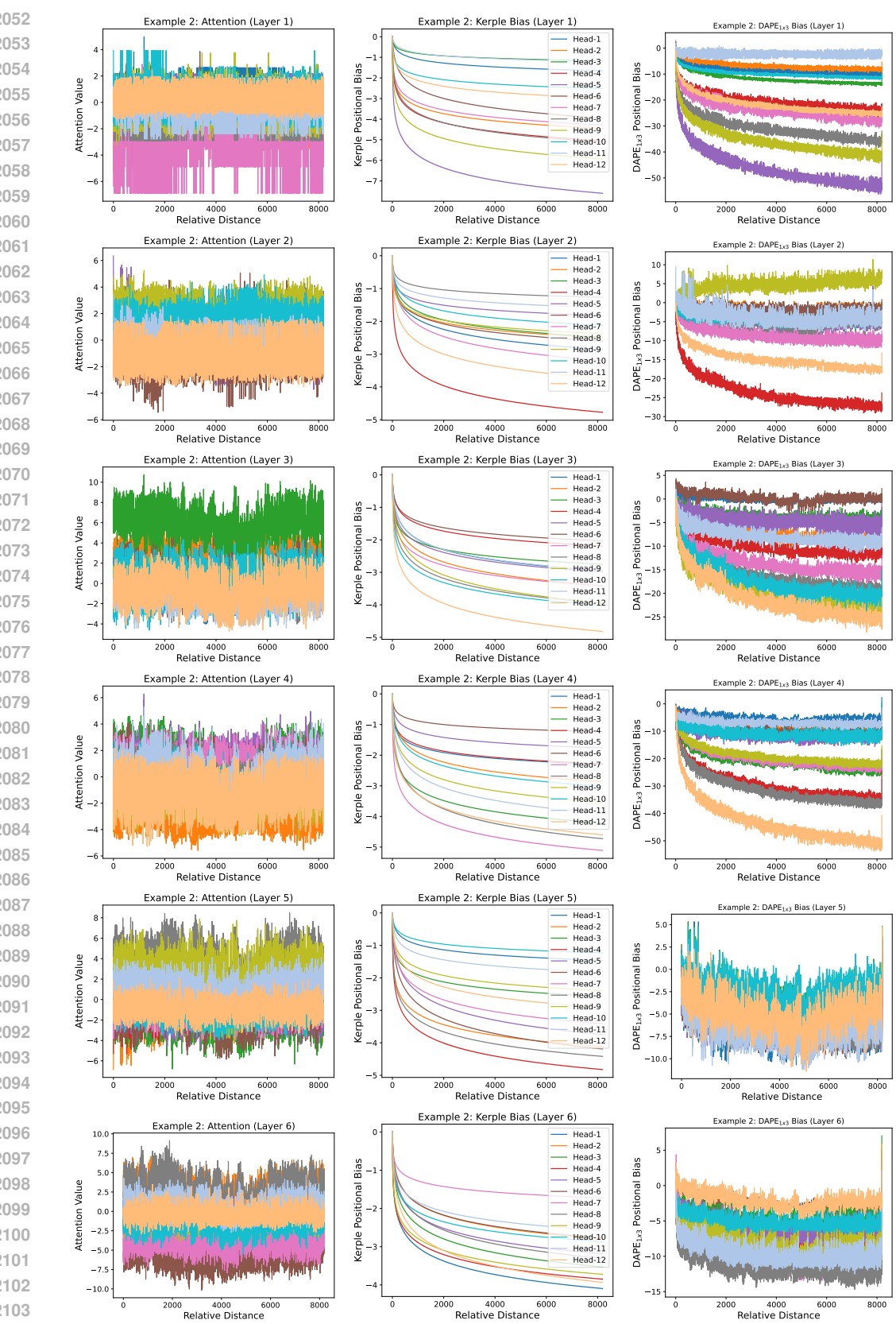

Figure 21: **Evaluation Length 8192 Example 2: Part 1. From Left to Right: (1) The Attention is $\boldsymbol{XW_Q(XW_K)}^\top$; (2) The Kerple bias is $\boldsymbol{B}$; (3) The DAPE$_{1\times3}$ (with Kerple) bias is $f(\boldsymbol{XW_Q(XW_K)}^\top, \boldsymbol{B})$.**

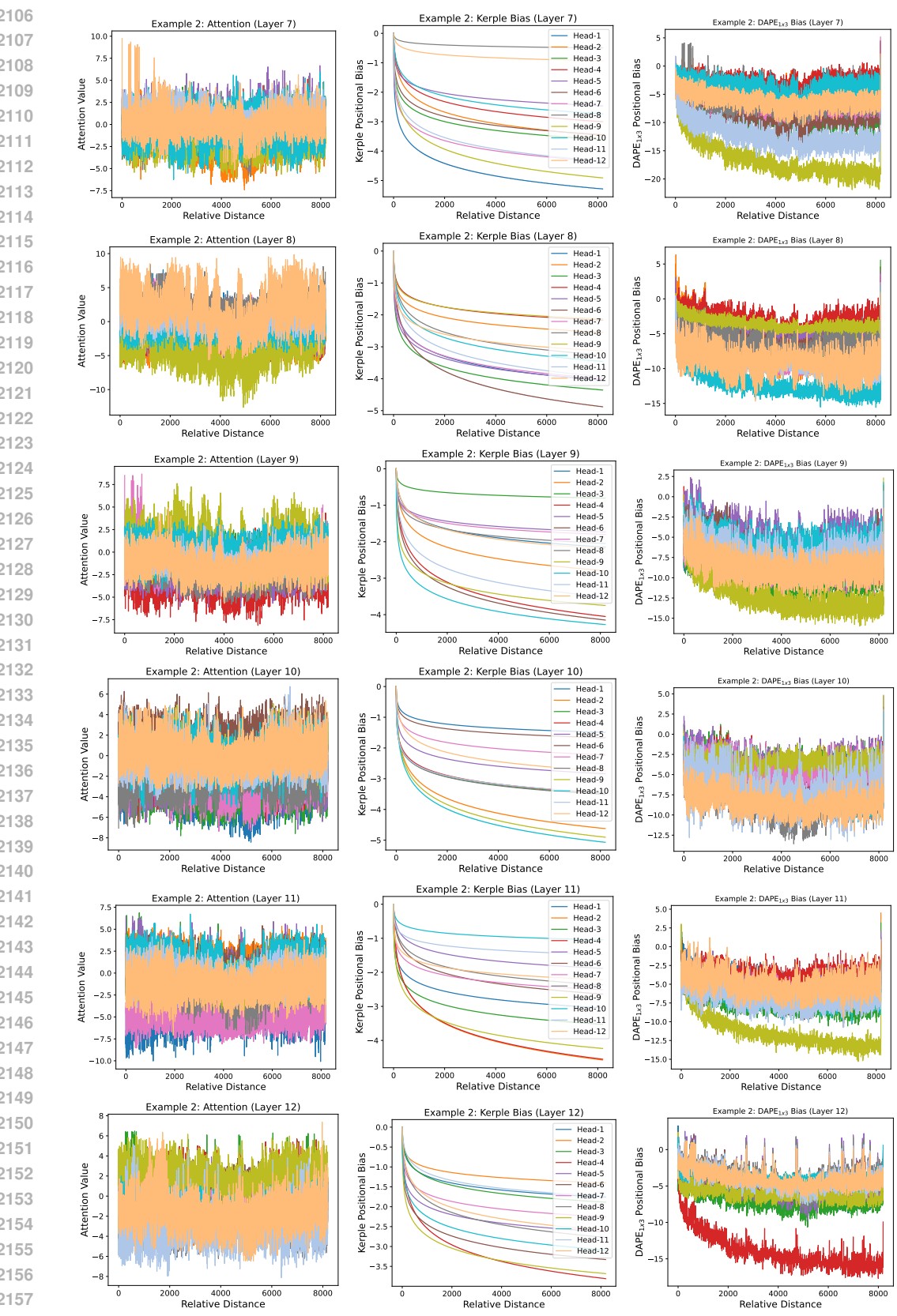

Figure 22: **Evaluation Length 8192 Example 2: Part 2. From Left to Right: (1) The Attention is $XW_Q(XW_K)^\top$; (2) The Kerple bias is $B$; (3) The DAPE$_{1\times3}$ (with Kerple) bias is $f(XW_Q(XW_K)^\top, B)$.**

## L  IMPLEMENTATION

In this section, we present the implementation of the proposed DAPE$_{1\times3}$ module in `PyTorch` (Paszke et al., 2019).

```python
import torch
import torch.nn as nn

class DAPEV2(nn.Module):
  def __init__(self, head_number=12, mlp_width=32, kernel_size=3):
    """
    DAPEV2 attention bias module.

    Args:
      num_heads: number of attention heads.
      mlp_width: Width of MLP.
      kernel_size: convolution kernel size.
    """
    super(DAPEV2, self).__init__()

    self.mlp =  nn.Sequential(
            nn.Conv2d(in_channels=head_number*2, out_channels=
                mlp_width, kernel_size=(1,kernel_size), stride=(1,1),
                padding=(0,kernel_size//2), dilation=(1,1)),
            nn.LeakyReLU(),
            nn.Conv2d(in_channels=mlp_width, out_channels=
                head_number, kernel_size=(1,kernel_size), stride=(1,1)
                , padding=(0,kernel_size//2), dilation=(1,1)))

  def forward(self, attention: torch.Tensor, bias: torch.Tensor):
    """
    Args:
      attention: input sequence, which is q^T * k,
          shape [bsz, num_heads, seq_len, seq_len]
      bias: bias matrix, which can be generated by ALiBi, Kerple
      FIRE or other additive position encodings
          shape [1,num_heads, seq_len, seq_len]

    Returns:
      attention with DAPEV2,
      shape [bsz, num_heads, seq_len, seq_len]
    """
    bias_tile=torch.tile(fire_bias, (x.shape[0],1,1,1) )
    attention_bias_concat=torch.cat( (attention, bias_tile), dim=1)
    attention_bias_concat=torch.tril(attention_bias_concat)
    attention_bias_concat=self.mlp(attention_bias_concat)

    return attention+bias+attention_bias_concat
```