# OpenReview forum: "DAPE V2: Process Attention Score as Feature Map for Length Extrapolation"
_ICLR.cc/2025/Conference — Submitted to ICLR 2025_

### Official Review · Reviewer_7aAt · 2024-11-03

**Soundness:** 3
**Presentation:** 1
**Contribution:** 2
**Rating:** 6
**Confidence:** 4

**Summary:**

This paper extends DAPE and proposes a new approach to address the Transformer long context extrapolation problem by treating attention scores as feature maps. The authors conceptualize attention mechanisms akin to image processing techniques, utilizing convolutional operations on attention scores across different heads. This methodology, inspired by methods in computer vision, enhances Transformer performance on extrapolation tasks across multiple lengths, both in theoretical underpinnings and through empirical validation. It has outperformed some popular position embedding methods such as RoPE and NoPE.

**Strengths:**

+ The application of convolution on attention maps to improve relative position encoding in large language models is both novel and inspiring. The use of convolution, a fast and efficient operation, allows for seamless integration into existing frameworks.

+ The proposed method demonstrates strong performance on the length extrapolation task, outperforming established techniques such as RoPE and NoPE, which underscores its effectiveness.

**Weaknesses:**

- The paper suffers from poor writing and organizational structure. Basic variables such as X, W_Q, and W_K​ are not adequately explained as the context, despite that Transformers are quite popular.

- Confusing Arguments:
1) Line 181-182 states: "The result of DAPE-NoPE (the Zheng et al. (2024) only combine DAPE with ALiBi, Kerple and FIRE but not with NoPE or RoPE)." This sentence is confusing and seems disconnected from the preceding context.
2) Line 191-192 mentions: "potentially hindering the evolution of next-generation Transformer models," which lacks clarity and context.
3) Line 198 states: "RoPE first computes the classic attention scores of key-query multiplication with RoPE." This description is unclear and requires further elaboration.

- The authors fail to adequately explain the rationale and motivation for applying convolution to embed position information, abruptly transitioning to technical details without sufficient context.

- The current popular solution for long-context extrapolation is to fine-tune RoPE-based LLMs on long-context data, which is not addressed in the baseline comparisons.

- There is no discussion of computational efficiency metrics such as FLOPS, which would be valuable for assessing the proposed method's practicality.

- The benchmarks employed in the study are limited, reducing the generalizability of the findings.

**Questions:**

Could the authors elaborate on "Proposition 1: Transformers incorporating convolution operations can perform associative recall tasks without the need for positional encoding"? The rationale behind this proposition is unclear and requires further explanation.

---

> ### Author Response · Authors · 2024-11-13
> **Response to Reviewer 7aAt  (Part 1/5)**
>
> Dear Reviewer 7aAt,
>
> Thank you very much for your suggestion, we will address your concerns below.
>
> **Q1: Basic variables such as X, W_Q, and W_K are not adequately explained in the context, despite the fact that Transformers are quite popular.**
>
> A1: Thank you very much for your notice. We have further explained the notations in our paper.
>
> **Q2: "The result of DAPE-NoPE (the Zheng et al. (2024) only combines DAPE with ALiBi, Kerple and FIRE but not with NoPE or RoPE)." This sentence is confusing and seems disconnected from the preceding context.**
>
> A2: The DAPE is designed for Additive RPE (such as ALiBi, Kerple or FIRE), but it does not try NoPE or RoPE. We have revised the presentation to the following.
>
> **Original**: The result of DAPE-NoPE (the Zheng et al. (2024) only combine DAPE with ALiBi, Kerple
> and FIRE but not with NoPE or RoPE).
>
> **Currently**: The DAPE~\cite{zheng2024dape} is designed for additive RPE but not trying NoPE or RoPE, and we present the results of DAPE-NoPE and DAPE-RoPE in the following.
>
> **Q3: Line 191-192 mentions: "potentially hindering the evolution of next-generation Transformer models," which lacks clarity and context.**
>
> A3: While the efficiency of transformers, particularly the quadratic cost of attention computation, is a significant concern in practical applications, we argue that the current transformer architecture may still lack sufficient expressiveness, as performance limitations are evident in certain cases. Sacrificing the expressiveness of transformers in favor of efficiency could hinder the development of architectures that meet the growing demands for large language models (LLMs). Therefore, in our work, we do not prioritize efficiency variants of transformers. Instead, we introduce additional convolution to the attention mechanism, which, though computationally more intensive, is able to enhance the model's capabilities.
>
>
> **Q4: Line 198 states: "RoPE first computes the classic attention scores of key-query multiplication with RoPE." This description is unclear and requires further elaboration.**
>
> A4: Thank you very much for your suggestion. We have revised the presentations.
>
> **Original:** RoPE first computes the classic attention scores of key-query multiplication with RoPE." This description is unclear and requires further elaboration.
>
> **Current**: In the DAPE-RoPE configuration, DAPE-RoPE first computes the classic attention scores of key-query multiplication with RoPE, which are then refined using the MLPs.
>
> **Q5: The authors fail to adequately explain the rationale and motivation for applying convolution to embed position information, abruptly transitioning to technical details without sufficient context.**
>
> A5: We have discussed the motivation in Section 3.2. The following is our thinking process.
>
> * DAPE is designed for additive RPE (such as ALiBI, Kerple, or FIRE), but not for NoPE or RoPE. **Then, what if there is no position encoding (NoPE)?**
> * Then, we conduct experiments to combine DAPE with NoPE and RoPE.
>    * For NoPE, we find that **DAPE-NoPE could directly improve the performance but DAPE-RoPE cannot**.
>    * For DAPE-NoPE, there is no bias matrix and DAPE could still improve the performance, suggesting that there have a more general underlying cause. The formulation of DAPE-NoPE is $QK^T + f(QK^T)$, while Q is query embedding and K is key embedding.
>    * We then realize that $QK^T + f(QK^T)$ is very similar to the ResNet's skip connection, and the MLP could be regarded as a 1x1 convolution operation. Then, we try 1x3 convolution and find that $DAPE_{1x3}$-NoPE achieves better performance than $DAPE$-NoPE.
>    * Also, though DAPE-RoPE cannot improve performance (compared to RoPE), the $DAPE_{1x3}$-RoPE could improve the performance compared to RoPE
> * Therefore, **until here, we finally realize that the we could extend the DAPE to ALL Transformer Attention (whatever the position encoding is) as long as we regards the Attention Score as Feature Maps**.

---

> ### Author Response · Authors · 2024-11-13
> **Response to Reviewer 7aAt (Part 2/5)**
>
> **A6: The current popular solution for long-context extrapolation is to fine-tune RoPE-based LLMs on long-context data, which is not addressed in the baseline comparisons.**
>
> A6: We answer the question in two questions:  1) the definition of length extrapolation; 2) RoPE Extrapolation Ability; 3) Our long-context evaluation.
>
> **The Definition of Length Extrapolation**
>
> According to ALiBi paper, the length extrapolation is that the model is trained on sequence length $T_{train}$ and validate on $T_{valid}$ and $T_{valid}$  is larger than $T_{train}$. Therefore, for the length extrapolation setting, usually, we do not further train the model beyond the training length or longer context data.
>
>
> **RoPE Extrapolation Ability**
>
> The RoPE extrapolation is relatively poor, suggesting that the perplexity will quickly increase when  $T_{valid}$ is larger than  $T_{train}.$ And this is proved in previous works, such as  YaRN.
>
>
> **Our Long-Context Evaluation**
>
> We understand that the reviewer would like to know more about the performance on long-context so we additionally employ $\Delta P$ to further evaluate the model. According to LongPPL [1], such a method has a strong correlation with long-context performance.
>
> In this discussion, we explore how to effectively use perplexity as a metric, incorporating concepts of information gain and entropy. Let $L(\cdot)$ represent the process for calculating loss, and $ M(x) $ denote the logit output generated by the model after processing an input sequence $ x $. For evaluating model performance, we define $ P(M(x), K) $ as follows:
> * Process the entire sequence $ x $ using $ M(x)$.
> *  Compute the perplexity on the last $ K $ tokens of the sequence.
>
> To interpret information gain, we consider the training sequence length $ T_{\text{train}}$. Given an input $ x $, we calculate the change in loss/perplexity, $ \Delta P$, as: $\Delta P = P(M(x[-T_{\text{test}}:]), T_{\text{train}}) - P(M(x), T_{\text{test}})$
>
> The term $\Delta P$ provides insights into the model's information gain relative to local and global context, allowing us to quantify entropy in terms of model uncertainty reduction. We interpret $ \Delta L $ as follows:
>
>
>
>
> | **Method**                               | **RoPE** | **ALiBi** | **Kerple** | **DAPE-Kerple** | **$DAPE_{1x3}$-Kerple** |
> |------------------------------------------|----------|-----------|------------|-----------------|-----------------------------|
> | $ P(M(x_{512}), T_{test}=256)$              | 19.74    | 20.04     | 19.83      | 19.25           | 18.95                        |
> | $ P(M(x_{1024}), T_{test}=256)$             | 261.39   | 19.74     | 19.19      | 18.28           | 17.92                        |
> | $ P(M(x_{1024}[-T_{train}:]), T_{test}=256)$| 19.51    | 19.79     | 19.58      | 19.03           | 18.74                        |
> | $\Delta P_{1024}$                        | -241.88  | 0.05      | 0.39       | 0.75            | **0.82**                         |
> | $ P(M(x_{2048}), T_{test}=256)$             | 411.23   | 20.17     | 20.48      | 17.20           | 16.79                        |
> | $ P(M(x_{2048}[-T_{train}:]), T_{test}=256)$| 18.74    | 19.03     | 19.84      | 18.28           | 18.01                        |
> | $\Delta P_{2048}$                        | -392.49  | -1.14     | -0.64      | 1.08            |  **1.22**                         |
> | $ P(M(x_{4096}), T_{test}=256)$             | 635.80   | 20.50     | 28.33      | 17.58           | 17.05                        |
> | $ P(M(x_{4096}[-T_{train}:]), T_{test}=256)$| 19.11    | 19.35     | 19.07      | 18.59           | 18.19                        |
> | $\Delta P_{4096}$                        | -616.69  | -1.15     | -9.26      | 1.01            |  **1.14**                         |
> | $ P(M(x_{8192}), T_{test}=256)$             | 762.86   | 21.30     | 40.94      | 17.85           | 17.20                        |
> | $ P(M(x_{8192}[-T_{train}:]), T_{test}=256)$| 19.78    | 20.02     | 19.85      | 19.38           | 18.98                        |
> | $\Delta P_{8192}$                        | -743.08  | -1.28     | -21.09     | 1.53            |  **1.78**                         |

---

> ### Author Response · Authors · 2024-11-13
> **Response to Reviewer 7aAt (Part 3/5)**
>
> * When $ \Delta P = 0 $: The model’s information gain from the full sequence is negligible, indicating an entropy level comparable to local attention (e.g., models like ALiBi when the evaluation length is 1024). This suggests the model does not leverage context beyond a limited range.
>
> * When $\Delta P < 0 $: Processing the entire sequence increases entropy, resulting in worse performance than focusing only on the last $ T_{\text{train}} $ tokens. This implies negative information gain and limited extrapolation capability (e.g. such as RoPE), as the model may overfit to recent tokens without capturing broader context effectively.
>
> * When $\Delta P > 0$: The model benefits from the information within $ x[:T_{\text{train}}] $, achieving a reduction in entropy that reflects positive information gain. This suggests the model leverages contextual information beyond the training sequence, indicating extrapolation capability.
>
> By examining $ \Delta P$, we can evaluate the model’s ability to reduce entropy and gain information from extended sequences, providing a measure of its extrapolative power.
>
>
> **Q7: There is no discussion of computational efficiency metrics such as FLOPS, which would be valuable for assessing the proposed method's practicality.**
>
> A7: We have discussed the computation cost in Section 4.2 and real time cost in Section 4.9.
>
> **Computation Cost**
> When T becomes larger, the FLOPS cost of $DAPE_{1x3}$ is $\mathcal{O}(B \cdot (h \cdot d \cdot T^2 + 3 \cdot h \cdot D_{\text{DAPE}} \cdot T^2))$, where $B$, $h$, $d$, $T$ and $D_{\text{DAPE}}$ are the batch size, attention head number, attention hidden dimension, sequence length and DAPE hidden dimension.
>
> **Time Cost in milliseconds**
> | **Method**                        | **350M Total** | **Ratio** | **2.7B Total** | **Ratio** | **6.7B Total** | **Ratio** |
> |------------------------------------|----------------|-----------|----------------|-----------|----------------|-----------|
> | RoPE            | 210.01         | 0.8306    | 472.63         | 1.0472    | 635.57         | 0.8564    |
> | T5's bias   | 355.16         | 1.4046    | 537.62         | 1.1912    | 808.85         | 1.0899    |
> | ALiBi         | 172.60         | 0.6826    | 325.95         | 0.7222    | 596.77         | 0.8041    |
> | Kerple           | 189.91         | 0.7511    | 370.32         | 0.8205    | 661.82         | 0.8918    |
> | FIRE           | 248.13         | 0.9813    | 432.63         | 0.9586    | 797.68         | 1.0748    |
> | $\text{DAPE}$-Kerple | 224.22         | 0.8868    | 422.48         | 0.9361    | 717.46         | 0.9667    |
> | $\text{DAPE}_{1\times3}$-Kerple   | 252.84         | 1.0000    | 451.29         | 1.0000    | 742.10         | 1.0000    |
>
>
> **As the model size increases, the additional computational cost ratio gradually decreases**. As
> shown in the above Table, when the model size is 350M, the time cost for Kerple is 189.91 ms, while
> DAPE-Kerple takes 224.22 ms, and DAPE1×3-Kerple requires 252.84 ms. Compared to DAPE1×3-
> Kerple, the time cost ratios for Kerple and DAPE-Kerple are 0.7511 and 0.8868, respectively. As
> the model size increases from 350M to 2.7B and 6.7B, the time cost ratio for Kerple rises from
> 0.7511 to 0.8205 and 0.8918, respectively. Similarly, the time cost ratio for DAPE-Kerple increases
> from 0.8868 to 0.9361 and 0.9677. Therefore, as the model size increases, the time cost ratio also
> increases, indicating that the additional computational cost decreases progressively.

---

> ### Author Response · Authors · 2024-11-13
> **Response to Reviewer 7aAt  (Part 4/5)**
>
> **Q8: The benchmarks employed in the study are limited, reducing the generalizability of the findings.**
>
> A8: **Our experiment setting follows the DAPE (NeurIPS 2024). Besides the experiment restuls on Arxiv and Books dataset, we also have the experiments on 14 downstream tasks which is evaluated via accuracy matrics.**
>
> | **Level** | **Task**                              | **Baseline**|  **Baseline**|  **Baseline**| **Baseline** | **Baseline** | **DAPE (Kernel Size 1)**| **DAPE (Kernel Size 1)** | **DAPE (Kernel Size 1)** | **DAPE (Kernel Size 3)** | **DAPE (Kernel Size 3)** | **DAPE (Kernel Size 3)**|
> |-----------|---------------------------------------|----------|--------------|-----------|------------|----------|-------------------------------|------------|----------|-------------------------------|------------|----------|
> |  |                               | **RoPE** | **Relative** | **ALiBi** | **Kerple** | **FIRE** | **ALiBi** | **Kerple** | **FIRE** | **ALiBi** | **Kerple** | **FIRE** |
> | **R**     | **Even Pairs**                        | 99.98    | 96.60        | 73.52     | 57.50      | 73.86    | 99.99                          | 99.58      | **100**      | 99.99                          | **100**       | **100**      |
> |           | **Modular Arithmetic Simple**         | 21.35    | 20.84        | 20.02     | 21.79      | 21.09    | 23.58                          | **24.47**      | 24.46    | 21.48                          | 23.90      | 23.43    |
> |           | **Parity Check** †††                 | 50.05    | 50.09        | 50.09     | 50.07      | 50.97    | 50.30                          | 50.07      | 50.04    | 50.13                          | **52.51**      | 50.11    |
> |           | **Cycle Navigation** †††             | 27.63    | 26.95        | 24.64     | 29.47      | 28.41    | 22.99                          | **34.53**      | 27.54    | 24.43                          | 24.32      | 24.34    |
> | **DCF**   | **Stack Manipulation**                | 61.49    | 64.73        | 66.42     | 66.93      | 69.33    | 68.18                          | **72.04**      | 70.90    | 58.90                          | 68.18      | 60.90    |
> |           | **Reverse String**                    | 65.23    | 65.59        | 71.09     | 71.54      | 65.89    | 73.37                          | 70.74      | 76.40    | 56.61                          | **81.84**      | 70.11    |
> |           | **Modular Arithmetic Brackets**      | 31.25    | 31.74        | 30.56     | 24.79      | 30.92    | 31.34                          | **32.37**      | 31.50    | 29.46                          | 26.13      | 27.00    |
> |           | **Solve Equation**                    | 21.85    | 22.93        | 19.92     | 21.15      | 22.06    | 20.03                          | 22.49      | 22.42    | 20.26                          | **23.95**      | 23.62    |
> | **CS**    | **Duplicate String**                  | 64.97    | 67.66        | 65.13     | 66.72      | 69.03    | 70.84                          | **72.95**      | 72.71    | 52.96                          | 57.03      | 66.01    |
> |           | **Missing Duplicate**                 | 63.37    | 72.34        | 74.21     | 79.06      | 79.27    | 83.41                          | 87.57      | 89.17    | 59.33                          | **99.65**      | 74.83    |
> |           | **Odds First**                        | 61.00    | 61.57        | 59.88     | 62.59      | 63.28    | 63.78                          | **67.08**      | 66.34    | 57.35                          | 56.87      | 56.57    |
> |           | **Binary Addition**                   | 55.59    | 56.96        | 54.72     | 56.35      | 55.70    | 59.71                          | **60.88**      | 56.62    | 57.49                          | 55.32      | 57.86    |
> |           | **Compute Sqrt**                      | 51.88    | 51.63        | 50.63     | 51.11      | 50.80    | 51.64                          | 51.33      | **52.46**    | 52.08                          | 51.76      | 51.93    |
> |           | **Bucket Sort** †††                   | 98.12    | 99.31        | 98.45     | 99.38      | **99.57**    | 99.38                          | 98.81      | 99.37    | 96.61                          | 99.06      | 98.56    |
>
>
> **Different tasks have different optimal kernel sizes** For example, on Missing Duplicate task, the  $DAPE_{1x3}$-Kerple improves the 87.57 of DAPE-Kerple to 99.65. However, on the Stack Manipulation task, the  $DAPE_{1x3}$-Kerple decreases the 72.04 of DAPE-Kerple to 68.18. the larger kernel size does not always lead to better performance. Overall, larger kernel size provides a potential way to improve the Transformer length extrapolation performance, and we usually could find a suitable kernel size (ranging from 1×1 to larger
> kernel sizes) to achieve better performance than without further processing attention score.

---

> ### Author Response · Authors · 2024-11-13
> **Response to Reviewer 7aAt (Part 5/5)**
>
> **Q9:Could the authors elaborate on "Proposition 1: Transformers incorporating convolution operations can perform associative recall tasks without the need for positional encoding"? The rationale behind this proposition is unclear and requires further explanation.**
>
> A9 Associative recall is one of the key capabilities of transformer models in handling language tasks. Previous research demonstrates that transformers achieve associative recall through positional embeddings. Intuitively, transformers can perform copy tasks because information about previous tokens is passed to subsequent tokens. For example, in the sequence {a, b, c, d, e, f, a}, the transformer can output "b" as the next token after the second "a" because the initial occurrence of "a" has been "copied" to "b." This allows the second "a" to attend to "b" and predict it as the next token. The mechanism enabling this copying is positional embedding. Most positional embeddings decay with respect to relative position, meaning attention is concentrated on neighboring tokens. This implicit associative recall mechanism has been theoretically verified and is learned during training [2]. However, a similar mechanism can be realized more directly and efficiently using convolution, eliminating the need for implicit learning through positional encodings (Proposition 1). The proof sketch is provided in lines 265-275.
>
> Reference:
>
> [1] Fang, L., Wang, Y., Liu, Z., Zhang, C., Jegelka, S., Gao, J., ... & Wang, Y. (2024). What is Wrong with Perplexity for Long-context Language Modeling?. arXiv preprint arXiv:2410.23771.
>
> [2] Alberto Bietti, Vivien Cabannes, Diane Bouchacourt, Herve Jegou, and Leon Bottou. Birth of a transformer: A memory viewpoint. Advances in Neural Information Processing Systems, 36, 2024.
>
> If there are any questions, please let us know. And if you think that we have addressed your concerns, could you please consider raising the score? Thank you very much for your support.

---

> ### Comment · Reviewer_7aAt · 2024-11-22
>
> Thanks for the hard working of authors during rebuttal. Some of my concerns are addressed and I'd like to raise the score slightly. I highly recommend authors to further improve the manuscript to include all the new experiments and revisions mentioned above.

---

> > ### Author Response · Authors · 2024-11-22
> > **Response to Reviewer 7aAt**
> >
> > Dear Reviewer 7aAt,
> >
> > Thank you very much for raising the score.
> >
> > **According to your precious suggestion, we have improved the manuscript to include all the new experiments and revisions mentioned above.**
> >
> > If possible, could we know whether there are any remaining concerns? Or, whether Reviewer 7aAt would like to discuss more about our perspective of the length extrapolation problem: query and key limits the length extrapolation performance? Or, is there anything we could do to further improve the score?
> >
> > Again, thank you very much for your precious attention.

---

### Official Review · Reviewer_UyHR · 2024-11-03

**Soundness:** 2
**Presentation:** 2
**Contribution:** 2
**Rating:** 5
**Confidence:** 4

**Summary:**

This paper introduces an incremental change over the prior work DAPE (Zheng et al., 2024), by extending the MLP used in the attention to 1x3 convolution. This small change achieves improvement over multiple experiment settings.

**Strengths:**

- The paper is clearly written.
- Extending the MLP in DAPE’s attention model to 1x3 convolution achieves improvement over multiple experiments.

**Weaknesses:**

- The major concern is this paper only introduces an incremental change over DAPE, I.e. extending the MLP in attention model to 1x3 convolution. In addition, compared to the gap between DAPE and other baselines, the gap between this paper and DAPE is relatively small.

- This paper could be written in a more straightforward way, by directly showing the difference between it and DAPE, and highlighting why it is crucial. Readers may have confusion about the contribution of this paper and DAPE.

- Line 126: It is hard to buy the insight: *Transformer’s length extrapolation ability is limited by the expressiveness of the naive query-key dot product.* This conclusion is drawn by showing DAPE without position encoding still achieves improvement. But there exists another explanation as follows. Transformer’s length extrapolation ability is limited due to the lack of accurate position encoding. MLP in DAPE implicitly learns the spatial information from the dot product of query and key, thus improving the performance. And extending MLP to 1x3 convolution can further improve encoding the spatial information.

- Discussion about an important reference is missing. “On Translation Invariance in CNNs: Convolutional Layers can Exploit Absolute Spatial Location” (in CVPR 2020), by Osman Semih Kayhan and Jan C. van Gemert. It found that the boundary effects operate even far from the image boundary, allowing the network to exploit absolute spatial location all over the image. This may help explain why convolution introduces more gains.

**Questions:**

Please see weaknesses (especially the third one).

---

> ### Author Response · Authors · 2024-11-13
> **Response to Reviewer UyHR (Part 1/2)**
>
> Dear Reviewer UyHR,
>
> Thank you very much for your comment, we will address your concerns below.
>
> **Q1: The major concern is this paper only introduces an incremental change over DAPE, I.e. extending the MLP in attention model to 1x3 convolution. In addition, compared to the gap between DAPE and other baselines, the gap between this paper and DAPE is relatively small.**
>
> A1: **Our target is eliminating the effect of training length as much as possible**, and ***Transformer performance has an upper bound (with the same training tokens) which is almost reached by our work**.*
>
> | **Length & Batch Size**                           | **128** | **256** | **512** | **1024** | **2048** | **4096** | **8192** |
> |---------------------------------------------------|---------|---------|---------|----------|----------|----------|----------|
> | **RoPE (Length 4096 & Batch 1)**                  | 38.37   | 33.21   | 27.34   | 25.50    | 23.55    | 24.58    | 152.54   |
> | **AliBi (Length 128 & Batch 32)**                 | 32.27   | 29.22   | 27.06   | 27.41    | 26.29    | 26.81    | 27.70    |
> | **DAPE$_{1x3}$-Kerple (Length 128 & Batch 32)**   | **31.07**   | 27.81   | 24.38   | 23.58    | 22.41    | 23.20    | 23.52    |
> | **DAPE$_{1x3}$-Kerple (Length 256 & Batch 16)**   | 31.18   | **27.61**   | 23.66   | 22.59    | 21.25    | 22.25    | 22.48    |
> | **DAPE$_{1x3}$-Kerple (Length 512 & Batch 8)**    | 31.73   | 27.85   | **23.48**   | **22.17**    | 20.68    | 21.15    | 20.98    |
> | **DAPE$_{1x3}$-Kerple (Length 1024 & Batch 4)**   | 32.74   | 28.53   | 23.73   | 22.18    | **20.61**    | **20.97**    | **20.60**    |
> | **DAPE$_{1x3}$-Kerple (Length 2048 & Batch 2)**   | 34.15   | 29.55   | 24.33   | 22.66    | 20.88    | 21.08    | 20.79    |
>
>
> **With the same training tokens, compared to $DAPE_{1x3}$ with longer training lengths, $DAPE_{1x3}$ with shorter training lengths can achieve comparable performance, indicating that $DAPE_{1x3}$ enhances the model's understanding of text structure**. On the arXiv dataset, $DAPE_{1x3}$-Kerple with training lengths of 512 demonstrates performance close to that of training with a length of 4096 when the evaluation length is 4096. Moreover, the performance curves for training lengths of 1024, and 2048 are almost identical. This trend is also observed with the Books3 dataset. These results indicate that $DAPE_{1x3}$-Kerple effectively helps the model comprehend text structure, enabling it to extend to longer lengths.
>
>
> | Method   |512       | 1024       | 2048       | 4096   | 8192 |
> |-----|-----------|-----------|-----------|------------|------------|
> | NoPE   | 4.68  | 31.79  | 1867.46  | 4666.60   | 5334.85  |
> | DAPE-NoPE   | 4.63  | 12.72  | 751.78  | 2033.33   | 2618.13   |
> | $DAPE_{1x3}-NoPE$   | 4.47  | 6.31  | 56.93  | 196.80  | 259.81   |
> | RoPE  | 4.57  | 43.62  | 144.05  | 278.87   | 297.06   |
> | DAPE-RoPE   | 4.53  | 73.31  | 174.48 | 316.84   | 306.78   |
> | $DAPE_{1x3}-RoPE$   | 4.48  | 13.10  | 29.41  | 53.97   | 68.32   |
>
> * For NoPE: reduce the perplexity **from 2618.13 to 259.81.**
> * For RoPE: reduce the perplexity **from 306.78 to 68.32.**
>
>
> **Q2: This paper could be written in a more straightforward way, by directly showing the difference between it and DAPE, and highlighting why it is crucial. Readers may be confusion about the contribution of this paper and DAPE.**
>
> A2: We have highlighted the difference between DAPE and this work at the beginning of Section 3 Method. We directly copy it below.
>
> **The two key differences between DAPE and this work are**:
> * **1) Insight:** DAPE attributes length extrapolation performance gains to adaptive position encoding and *DAPE believes that the Bias Matrix is Necessary*, while this work finds DAPE could still improve performance without position encoding so that **we take a broader view, explaining that the Transformer's length extrapolation ability is limited by the expressiveness of the naive query-key dot product, which can be enhanced using image processing techniques;**
> * **2) Performance:** DAPE is designed for additive RPE and may underperform with non-additive RPE (e.g., RoPE), whereas this work suggests that increasing kernel size (e.g., with  $DAPE_{1x3}$) may improve RoPE's performance**.

---

> ### Author Response · Authors · 2024-11-13
> **Response to Reviewer UyHR (Part 2/2)**
>
> **Q3: Line 126: It is hard to buy the insight: Transformer’s length extrapolation ability is limited by the expressiveness of the naive query-key dot product. This conclusion is drawn by showing DAPE without position encoding still achieves improvement. But there exists another explanation as follows. Transformer’s length extrapolation ability is limited due to the lack of accurate position encoding. MLP in DAPE implicitly learns the spatial information from the dot product of query and key, thus improving the performance. And extending MLP to 1x3 convolution can further improve encoding the spatial information.**
>
> A3: It seems that our explanation and the above explanation do not conflict, but we would like to highlight that **MLP (kernel size 1x1) cannot learn spatial information from scratch because there is NO Zero Padding.**
>
>
> **The two explanations do not conflict (both aiming for better attention score): limited expressiveness of query and key dot product actually leads to the inaccurate attention score, and the lack of accurate position encoding (which is caused by limited query and key dot product for NoPE [1]) also can lead to inaccurate attention score.**
>
> We suggest that actually the length extrapolation is limited by the query and key dot product expressiveness. And the above explanation suggests that the MLP actually learns the spatial information from the dot product of query and key. Let us explain step by step.
> * **MLP (kernel size 1x1) cannot learn spatial information from scratch because there is NO Zero Padding.**
> * **The convolutional neural network can introduce spatial information because of zero padding [2], while MLP (kernel size 1x1) cannot generate spatial information from scratch because MLP does not have zero padding.** For MLP (convolutional neural network with kernel size 1x1), it cannot generate spatial information from scratch because there is NO zero padding.
> * Then, why MLP (e.g. DAPE-NoPE) can still enhance the performance, as MLP cannot generate position information from scratch?
>    * **For the decoder-only transformer, the naive query and key dot product (without MLP) could already learn and contain position information (spatial information), and this is already proved in NoPE paper [1].**
>    * Therefore, the MLP (e.g. DAPE-NoPE) can enhance the existing spatial information (improve the expressiveness of query and key dot product) but not generate spatial information from scratch because the original query and key dot product already contain such spatial information and there is NO zero padding for MLP.
> * Finally, the underlying reason for DAPE-NoPE improving performance is clear. As the MLP cannot generate spatial information from scratch and it can still improve the performance of DAPE-NoPE, the reason is the query and key dot products (without MLP) have limited expressiveness so that the corresponding attention score expreviness is limited so that the learned spatial information is limited, and MLP enhances such expressiveness so that spatial information is enhanced.
> * For larger kernel size, it could bring two benefits: 1) higher expressiveness; 2) more spatial information because of zero padding.
> * With the explanation of the limited expressiveness of query and key dot product (leading to inaccurate attention score), we could explain both MLP (kernel size 1x1) and large kernel size why they work: improve the expressiveness of attention score so that attention score becomes better.
> * **Therefore, we suggest that we should process the Attention Score as Feature Maps. And it seems that our explanation (enhance expressiveness, aiming at a better attention score) and the above explanation (better position encoding, which also aims at a better attention score) do not conflict. We have revised the manuscript according to the above discussion.**
>
>
> **A4: Discussion about an important reference is missing. “On Translation Invariance in CNNs: Convolutional Layers can Exploit Absolute Spatial Location” (in CVPR 2020), by Osman Semih Kayhan and Jan C. van Gemert. It found that the boundary effects operate even far from the image boundary, allowing the network to exploit absolute spatial location all over the image. This may help explain why convolution introduces more gains.**
>
> A4: Thank you very much for your notice. We have added it to the reference and discussed it in our paper.
>
> If there are any questions, please let us know. And if you think that we have addressed your concerns, could you please consider raising the score? Thank you very much for your support.
>
>
> Reference:
>
> [1] Kazemnejad, A., Padhi, I., Natesan Ramamurthy, K., Das, P., & Reddy, S. (2024). The impact of positional encoding on length generalization in transformers. Advances in Neural Information Processing Systems, 36.
>
> [2] Islam, M. A., Jia, S., & Bruce, N. D. (2020). How much position information do convolutional neural networks encode?. arXiv preprint arXiv:2001.08248.

---

> ### Author Response · Authors · 2024-11-24
> **Further Evidence for the Performance**
>
> Dear Reviewer UyHR,
>
> We further provide explanation for the concerns:
> * **The difference between DAPE and this work:** We have highlighted the core difference in the **Update Summary and the Core Contribution**.
> * **The performance increase between this work and DAPE**: we discuss it in detail in the following to prove that **The Performance Gap between this work and DAPE is actually Significant**.
>
> **The Significantly Perplexity Drop, Compared to original DAPE**
> *Table: Perplexity Performance with Different Kernel Sizes (on Arxiv Datasets, Training Length 512, Evaluation from Length 512 to 8192)*
> | Method   |512       | 1024       | 2048       | 4096   | 8192 |
> |-----|-----------|-----------|-----------|------------|------------|
> | NoPE   | 4.68  | 31.79  | 1867.46  | 4666.60   | 5334.85  |
> | DAPE-NoPE   | 4.63  | 12.72  | 751.78  | 2033.33   | 2618.13   |
> | $DAPE_{1x3}-NoPE$   | 4.47  | 6.31  | 56.93  | 196.80  | 259.81   |
> | RoPE  | 4.57  | 43.62  | 144.05  | 278.87   | 297.06   |
> | DAPE-RoPE   | 4.53  | 73.31  | 174.48 | 316.84   | 306.78   |
> | $DAPE_{1x3}-RoPE$   | 4.48  | 13.10  | 29.41  | 53.97   | 68.32   |
>
> * For NoPE: reduce the perplexity **from 2618.13 to 259.81, which reduces 90.07% perplexity.**
> * For RoPE: reduce the perplexity **from 306.78 to 68.32, which reduces 77.72% perplexity.**
>
>
> **The Significant Loss Reduction, proved by other popular works, such as Differential Transformer**
>
> **The Differential Transformer is a popular work and received an average score of 8 in the ICLR 2025 submission, and we copy its reported loss directly for comparison.** Here, we compare the loss reduction within training length to prove that this work actually achieves significant loss reduction.
>
> *Table: Loss Performance with Different Kernel Sizes (on Books dataset with training length 512)*
> | Method   |Loss       | Loss Reduction |Loss Reduction Ratio |
> |-----|-----------|-----------|-----------|
> | Differential Transformer Baseline   | 3.086  | ---  | ---  |
> | Differential Transformer   | 3.062  | 0.024  | 0.77%|
> | Kerple (This Work)   | 2.987  |  ---  | ---  |
> | DAPE-Kerple (This Work)  | 2.957  | 0.030  | 0.99%|
> | $DAPE_{1x3}-Kerple$ (This Work)   | 2.942  | 0.015  | 0.51%|
>
> **For the performance within training length, even reducing 0.01 loss is still very difficult and challenging. The $DAPE_{1x3}-Kerple$ reduce the 0.045 loss compared to Kerple, while Differential Transformer reduce 0.024 loss with reduction ratio 0.77%. Even compared with DAPE-Kerple, $DAPE_{1x3}-Kerple$ still reduce 0.015 loss and the reduction ratio 0.51%. This has proved that the performance gap reduction is actually significant**

---

> ### Author Response · Authors · 2024-11-25
> **Kind Reminder of Discussion Period**
>
> Dear Reviewer UyHR,
>
> Hope this finds you well.
>
> As the discussion period will be closed in one day, if possible, could we know whether there are any left concerns? And if possible, could you consider increasing the score after reading our above response?
>
> Again, thank you very much for your attention, and thank you very much for your precious comments.

---

> ### Author Response · Authors · 2024-11-27
>
> Dear Reviewer UyHR,
>
> We sincerely appreciate your thorough review and the insightful suggestions you provided for our paper. We have carefully reviewed and addressed each of your comments in detail within our rebuttal.
>
> As the Author-Review Discussion period draws to a close, we want to ensure that all your concerns have been addressed. Should there be any remaining questions or unresolved issues, we would be more than happy to provide further clarification or implement any necessary revisions.
>
> Thank you once again for your precious feedback and thoughtful engagement.

---

> > ### Comment · Reviewer_UyHR · 2024-11-27
> > **Thank you for the response**
> >
> > Thank you to the authors for the detailed reply and for addressing my initial concerns. I understand the paper's primary contributions to be:
> >
> > 1. Improved performance through the use of 1x3 convolution (DAPE_1x3-NoPE)
> > 2. Explanation of the MLP mechanism within the previous DAPE iteration (DAPE-NoPE)
> > 3. Extension of the explanation in (2) to the improvement in (1)
> >
> > To better assess the relative significance of these contributions, could the authors provide a weighted distribution of 100 points across these three contributions (e.g., 30/30/40)?

---

> ### Author Response · Authors · 2024-11-27
> **Response to Reviewer UyHR**
>
> Dear Reviewer UyHR,
>
> Thank you very much for your response, which really helps a lot! We will answer your question below.
>
> **We first clarify the contribution**
> * **Improved performance through the use of 1x3 convolution (DAPE_1x3-NoPE):** reduced the perplexity from 2618.13 to 259.81, which reduces 90.07% perplexity.
> * **Explanation of the MLP mechanism within the previous DAPE iteration (DAPE-NoPE):** DAPE-NoPE works because of improves the expressiveness of query and key dot product.
> * **Extension of the explanation in (2) to the improvement in (1)**: we find that we can process attention score as a feature map to improve query and key dot product expressiveness.
>
> **The weights in details**:
> * **Improved performance through the use of 1x3 convolution (DAPE_1x3-NoPE):** 10 points
> * **Explanation of the MLP mechanism within the previous DAPE iteration (DAPE-NoPE):** 50 points
> * **Extension of the explanation in (2) to the improvement in (1)**: 40 points
>
> **The explanation of the given weights**
> * **The improved performance is just the beginning of this important research new direction**
>    * In this work, we directly use a larger kernel and already achieved significant improvement.
>    * In the future, other research could utilize more and better feature map processing methods to further improve the performance, as the most direct solution (larger kernel size) could already significantly improve the performance
> * **Explanation of the MLP mechanism within the previous DAPE iteration (DAPE-NoPE) suggests an important part that is missed by previous works: the query and key dot product have limited expressiveness, which limits the transformer performance (especially length extrapolation)**
>    * Before this explanation, we mainly solved long-context/length extrapolation from two directions/perspectives:
>       * **Previous Research Direction 1 for Length Extrapoaltion**: Better Position Encoding (proposed by Vaswani, A. (2017) ). With such perspectives, our community develops different position encodings, including RoPE, ALiBi, Kerple, FIRE.
>       * **Previous Research Direction 2 for Length Extrapoaltion**: Position Interpolation (proposed by Chen, S. (2023) ). With such perspectives, there are extensive following papers, including YaRN, CLEX and so on.
>    * **In this work, we propose the Third Research Direction**: the length extrapolation is difficult because the query and key dot product have limited expressiveness, and direct convolution operation on attention score could significantly improve the length extrapolation performance, whatever the position encoding is.
>
> * **Extension of the explanation in (2) to the improvement in (1)**
>    * Based on the explanation in (2), the DAPE_1x3-NoPE achieves better performance than DAPE-NoPE is obvious: a larger kernel brings better expressiveness after processing query and key dot product.
>    * We also give it a high score than (1) because:
>        * **It is the first time that we successfully proved that attention score could be processed as a feature map to improve performance**
>       * **It could inspire us to think the essence of attention score**
>          *   Besides regarding the attention score as a feature map, what else could the attention score be regarded as?
>          * Besides the convolution operation, any other operations that can also improve the expressiveness of attention score?
>          * What is the essence of attention score? If the attention score is not just Feature Map, then could we use a better way to construct the attention score?
>          * ... ...
>    * We also give it a lower score than (2) because:
>         * **The (3) is based on (2), and the explanation of (2) is more essential**.
>         * Based on (2), there may be various opinions of attention score, including but limited to feature map.
>         * **Therefore, (3) (process as feature map) is just a potential way to improve the limited expressiveness (which is suggested by  (2)).**
>
>
> To summarize,
> * **This work's improved performance is just the beginning of this research direction (query and key dot product have limited expressiveness) so that only 10 points.**
> * **Explanation of the MLP mechanism within the previous DAPE iteration (DAPE-NoPE) reveals an important NEW Research Direction: the query and key dot product have limited expressiveness so we have to improve it, whatever the method is, so that 50 points.**
> * **Based on (2), the Extension of the explanation in (2) to the improvement in (1) first suggests that we may improve the expressiveness by processing attention score as a feature map, and experiments support such opinion. As the (3) is based on (2), so that (3) has 40 points.**
>
> If there is anything that Reviewer UyHR would like to discuss (including but not limited to this work but also our vision of the future research direction), please let us know.
>
> **And if possible, could you consider increasing the score?**

---

> > ### Comment · Reviewer_UyHR · 2024-11-28
> >
> > I understand that MLP can improve the semantic correlation between query and key. However, how do you draw the conclusion that this improvement is correlated to the length extrapolation? Could you privide statistic evidence to support this correlation?

---

> > > ### Author Response · Authors · 2024-11-28
> > > **Response to Reviewer UyHR**
> > >
> > > Dear Reviewer UyHR,
> > >
> > > Thank you very much for your response, we will address your concerns below.
> > >
> > > **Q1: how do you draw the conclusion that this improvement is correlated to the length extrapolation?**
> > >
> > > A1: **The improvement of attention score (query and key dot product) expressiveness improves the Overall Performance (within or beyond training length), including length extrapolation (beyond training length) performance**
> > >
> > > Table: Perplexity Performance with Different Kernel Sizes (on Arxiv Datasets, Training Length 512, Evaluation from Length 512 to 8192)
> > > | Method   |512       | 1024       | 2048       | 4096   | 8192 |
> > > |-----|-----------|-----------|-----------|------------|------------|
> > > | NoPE   | 4.68  | 31.79  | 1867.46  | 4666.60   | 5334.85  |
> > > | DAPE-NoPE   | 4.63  | 12.72  | 751.78  | 2033.33   | 2618.13   |
> > > | $DAPE_{1x3}-NoPE$   | 4.47  | 6.31  | 56.93  | 196.80  | 259.81   |
> > >
> > > **The improvement of expressiveness improves the performance within the training length 512 and beyond the training length 512.** According to the above Table, we can find that
> > > * The DAPE-NoPE (kernel 1x1) has higher expressiveness than NoPE
> > >    * **Within training length 512**, the DAPE-NoPE (4.63 perplexity),  achieves better performance (lower perplexity) than NoPE (4.68 perplexity)
> > >    * **Beyond training length 512**, The DAPE-NoPE (2618.13 perplexity) achieves better performance  (lower perplexity) and NoPE (5334.85 perplexity).
> > > * The $DAPE_{1x3}-NoPE$ has higher expressiveness than NoPE:
> > >    * **Within training length 512**， The $DAPE_{1x3}-NoPE$ (4.47 perplexity)  achieves better performance  (lower perplexity) than DAPE-NoPE (4.63 perplexity)
> > >    * **Beyond training length 512**, The $DAPE_{1x3}-NoPE$ (259.81 perplexity) achieves better performance  (lower perplexity)   than DAPE-NoPE (2618.13 perplexity).
> > >    * **Therefore, actually, the improvement of expressiveness leads to better Overall Performance within and beyond training length, and the length extrapolation performance improvement is more significant with the evaluation length increase.**
> > >
> > > **Q2: Could you privide statistic evidence to support this correlation?**
> > >
> > > A2: **As we explain above, the improvement of expressiveness leads to better performance (within and beyond training length). The following is the potential theorem explanation.**
> > > * **Step 1 (The formulation has better expressiveness, according to universal approximation [1] or statistical view).**
> > >    * The formulation is $QK^T+f(QK^T)$ so that the f(QK^T) could be degraded to zero if not necessary. Therefore, the formulation $QK^T+f(QK^T)$ has better expressiveness than original $QK^T$
> > > * **Step 2 (Higher expressiveness leads to better performance, including the performance within the training length and the performance beyond the training length).**
> > >    * We both agree that MLP could bring more expressiveness, and better expressiveness could usually bring better performance. Therefore, the expressiveness is improved so that performance is better, whether within or beyond training length.
> > > * Step 3: **Therefore, based on Step 1 and Step 2, the MLP (better expressiveness) could bring better performance for length extrapolation performance.**
> > >
> > > If there is any other question, please let us know.
> > >
> > > **And if our response addresses the concerns, could you consider increasing the score? Thank you very much for your precious support**
> > >
> > > Reference:
> > >
> > > [1] Hornik, K., Stinchcombe, M., & White, H. (1989). Multilayer feedforward networks are universal approximators. Neural networks, 2(5), 359-366.

---

> > > > ### Author Response · Authors · 2024-11-29
> > > >
> > > > Dear Reviewer UyHR,
> > > >
> > > > If possible, could we know whether you have any other questions？
> > > >
> > > > **As the most concerns (including the initial concerns) are already addressed,  could you please consider raising your rating of the paper  ?**

---

> ### Author Response · Authors · 2024-12-02
> **Further Explanation and Request for Response**
>
> Dear Reviewer UyHR,
>
> **As the discussion period will be closed in 1 day. If possible, could we know whether you have any further questions?**
>
> **For the Question about how we you draw the conclusion that this improvement is correlated to the length extrapolation, we further provide experiment results**
>
> | Method   |512       | 1024       | 2048       | 4096   | 8192 |
> |-----|-----------|-----------|-----------|------------|------------|
> | NoPE   | 4.68  | 31.79  | 1867.46  | 4666.60   | 5334.85  |
> | DAPE-NoPE   | 4.63  | 12.72  | 751.78  | 2033.33   | 2618.13   |
> | $DAPE_{1x3}-NoPE$   | **4.47**  | **6.31**  | **56.93**  | **196.80**  | **259.81**   |
> | ALiBi              | 4.61        | 4.59        | 5.00        | 4.86        | 4.59        |
> | DAPE-ALiBi | 4.52        | 4.23        | 4.24        | 4.04        | 3.83        |
> | $DAPE_{1x3}-ALiBi$ | **4.46**        | **4.17**        | **4.17**        | **3.97**        | **3.76**        |
> | FIRE               | 4.57        | 5.18        | 64.06       | 304.06      | 497.43      |
> | DAPE-FIRE | 4.49        | 4.21        | 4.28        | 4.41        | 5.14        |
> | $DAPE_{1x3}-FIRE$ | **4.43**        | **4.17**        | **4.24**        | **4.29**        | **4.66**        |
> | Kerple   | 4.57  | 4.37  | 5.09  | 6.80   | 9.08  |
> | DAPE-Kerple   | 4.49  | 4.20  | 4.17  | 3.95   | 3.70   |
> | $DAPE_{1x3}-Kerple$   | **4.44**  | **4.14**  | **4.09**  | **3.87**  | **3.58**   |
>
> **The improvement of expressiveness improves the performance within the training length 512 and beyond the training length 512.** According to the above Table, we can find that
> * The DAPE has higher expressiveness than baseline
>    * **Within training length 512**, the DAPE achieves better performance than baseline, whatever the baseline is NoPE, ALiBi, FIRE, or Kerple.
>    * **Beyond training length 512**, The DAPE achieves better performance than the baseline, whatever the baseline is NoPE, ALiBi, FIRE, or Kerple.
> * The $DAPE_{1x3}$ has higher expressiveness than DAPE:
>    * **Within training length 512**, the $DAPE_{1x3}$  achieves better performance than DAPE, whatever the baseline is NoPE, ALiBi, FIRE, or Kerple.
>    * **Beyond training length 512**, The $DAPE_{1x3}$  achieves better performance than DAPE, whatever the baseline is NoPE, ALiBi, FIRE, or Kerple.
> * **Therefore, actually, the improvement of expressiveness leads to better Overall Performance within and beyond training length, and the length extrapolation performance improvement is more significant with the evaluation length increase.**
>
> As the discussion period will be closed in 1 day. If possible, could we know whether you have any further questions?
>
> **As we both agreed that the initial concerns (before rebuttal) are addressed, and we also present the vision/explanation of the length extrapolation problem in the following discussion, could you please consider raising your rating of the paper? We really appreciate your support of our work.**

---

> ### Comment · Reviewer_UyHR · 2024-12-03
>
> Thank you to the authors for the detailed reply. From the results, I observe that using MLP or conv 1x3 achieves better performance, and the performance gain increases with longer training length. While this is an interesting observation, I believe the paper could benefit from a more in-depth analysis of the reasons behind this trend.
>
> For example, when using DAPE, why does the performance fluctuate for FIRE and ALiBi but keep improving for Kerple when training length increases? Analyzing the learned attention maps of these methods at different training lengths might reveal how DAPE affects the model's learning dynamics. Additionally, investigating the impact of different training hyperparameters or conducting ablation studies could provide further insights.
>
> I appreciate the authors' efforts in addressing my previous concerns. However, at this time, I believe a deeper investigation into the reasons behind these trends is needed to fully justify a higher rating.

---

> ### Author Response · Authors · 2024-12-03
> **Response to Reviewer UyHR (1/3)**
>
> Dear Reviewer UyHR,
>
> Thank you very much for your response, We answer your question below.
>
> **Q1: performance fluctuate for FIRE and ALiBi but keep improving for Kerple when training length increases**
>
> A1: **There is a misunderstanding: The FIRE performance keeps improving, and only ALiBi performance needs longer training length**
>
> We present the performance of ALiBi, Kerple and FIRE below.
>
> *Table: Perplexity Performance with on Arxiv Dataset (Training Length 128, Evaluation from Length 128 to 8192)*
> | Model          | 128   | 256   | 512   | 1024   | 2048    | 4096    | 8192    |
> |----------------|-------|-------|-------|--------|---------|---------|---------|
> | Alibi          | 8.33  | 7.16  | 6.12  | 6.32   | 6.56    | 6.42    | 6.05    |
> | DAPE Alibi     | 8.24  | 7.02  | 5.48  | **5.43**   | **5.64**    | **5.55**    | **5.18**    |
> | $DAPE_{1x3}$ Alibi   | **8.17**  | **6.94**  | **5.42**  | 6.69   | 9.53    | 10.52   | 9.55    |
> | Kerple         | 8.30  | 7.11  | 5.85  | 6.92   | 9.17    | 11.49   | 12.60   |
> | DAPE Kerple    | 8.21  | 6.99  | 5.39  | 5.20   | 5.34    | 5.27    | 4.97    |
> | $DAPE_{1x3}$ Kerple  | 8.15  | 6.92  | 5.29  | 5.06   | 5.12    | 4.95    | 4.61    |
> | FIRE           | 8.29  | 7.14  | 6.75  | 22.67  | 170.87  | 759.26  | 1616.82 |
> | DAPE FIRE      | 8.21  | 7.02  | 5.70  | 7.95   | 68.46   | 514.13  | 1290.97 |
> | $DAPE_{1x3}$ FIRE    | 8.13  | 6.92  | 5.42  | 5.94   | 13.51   | 134.27  | 816.04  |
>
> *Table: Perplexity Performance with on Arxiv Dataset (Training Length 512, Evaluation from Length 512 to 8192)*
> | Model          | 512   | 1024   | 2048    | 4096    | 8192    |
> |----------------|-------|-------|-------|--------|---------|
> | Alibi          | 4.61  | 4.59  | 5.00  | 4.86   | 4.59    |
> | DAPE Alibi     | 4.52  | 4.23  | 4.24  | 4.04   | 3.83    |
> | $DAPE_{1x3}$ Alibi   | **4.46**  | **4.17**  | **4.17**  | **3.97**   | **3.76**    |
> | Kerple         | 4.58  | 4.37  | 5.10  | 6.81   | 9.08    |
> | DAPE Kerple    | 4.50  | 4.21  | 4.17  | 3.96   | 3.71    |
> | $DAPE_{1x3}$ Kerple  | 4.44  | 4.15  | 4.10  | 3.87   | 3.58    |
> | FIRE           | 4.57  | 5.18  | 64.06 | 304.06 | 497.43  |
> | DAPE FIRE      | 4.49  | 4.21  | 4.28  | 4.41   | 5.14    |
> | $DAPE_{1x3}$ FIRE    | 4.43  | 4.17  | 4.24  | 4.29   | 4.66    |
>
> * **The FIRE performance keeps impriving, but not Reviewer's understanding: performance fluctuate**
> * **Even with a shorter length,  $DAPE_{1x3}$ Alib, the performance could still be  improved within length 512.**
> * **ALiBI is NOT a good method because it cheats perplexity (ALiBI will quickly become local attention when length increases). Hence, pay more attention to Kerple and FIRE**

---

> ### Author Response · Authors · 2024-12-03
> **Response to Reviewer UyHR (2/3)**
>
> In this discussion, we explore how to effectively use perplexity as a metric, incorporating concepts of information gain and entropy. Let $P(\cdot)$ represent the process for calculating perplexity, and $ M(x) $ denote the logit output generated by the model after processing an input sequence $ x $. For evaluating model performance, we define $ P(M(x), K) $ as follows:
> * Process the entire sequence $ x $ using $ M(x)$.
> *  Compute the perplexity on the last $ K $ tokens of the sequence.
>
> To interpret information gain, we consider the training sequence length $ T_{\text{train}}$. Given an input $ x $, we calculate the change in loss/perplexity, $ \Delta P$, as: $\Delta P = P(M(x[-T_{\text{train}}:]), T_{\text{test}}) - P(M(x), T_{\text{test}})$
>
>
>
> The term $\Delta P$ provides insights into the model's information gain relative to local and global context, allowing us to quantify entropy in terms of model uncertainty reduction. We interpret $ \Delta P $ with $T_{train}=512$ as follows:
>
> * When $ \Delta P = 0 $: The model’s information gain from the full sequence is negligible, indicating an entropy level comparable to local attention (e.g., models like ALiBi when the evaluation length is 1024). This suggests the model does not leverage context beyond a limited range.
>
> * When $\Delta P < 0 $: Processing the entire sequence increases entropy, resulting in worse performance than focusing only on the last $ T_{\text{train}} $ tokens. This implies negative information gain and limited extrapolation capability (e.g. such as RoPE), as the model may overfit to recent tokens without capturing broader context effectively.
>
> * When $\Delta P > 0$: The model benefits from the information within $ x[:T_{\text{train}}] $, achieving a reduction in entropy that reflects positive information gain. This suggests the model leverages contextual information beyond the training sequence, indicating extrapolation capability.
>
> | **Method**                               | **RoPE** | **ALiBi** | **Kerple** | **DAPE-Kerple** | **$DAPE_{1x3}$-Kerple** |
> |------------------------------------------|----------|-----------|------------|-----------------|-----------------------------|
> | $ P(M(x_{512}), T_{test}=256)$              | 19.74    | 20.04     | 19.83      | 19.25           | 18.95                        |
> | $ P(M(x_{1024}), T_{test}=256)$             | 261.39   | 19.74     | 19.19      | 18.28           | 17.92                        |
> | $ P(M(x_{1024}[-T_{train}:]), T_{test}=256)$| 19.51    | 19.79     | 19.58      | 19.03           | 18.74                        |
> | $\Delta P_{1024}$                        | -241.88  | 0.05      | 0.39       | 0.75            | **0.82**                         |
> | $ P(M(x_{2048}), T_{test}=256)$             | 411.23   | 20.17     | 20.48      | 17.20           | 16.79                        |
> | $ P(M(x_{2048}[-T_{train}:]), T_{test}=256)$| 18.74    | 19.03     | 19.84      | 18.28           | 18.01                        |
> | $\Delta P_{2048}$                        | -392.49  | -1.14     | -0.64      | 1.08            |  **1.22**                         |
> | $ P(M(x_{4096}), T_{test}=256)$             | 635.80   | 20.50     | 28.33      | 17.58           | 17.05                        |
> | $ P(M(x_{4096}[-T_{train}:]), T_{test}=256)$| 19.11    | 19.35     | 19.07      | 18.59           | 18.19                        |
> | $\Delta P_{4096}$                        | -616.69  | -1.15     | -9.26      | 1.01            |  **1.14**                         |
> | $ P(M(x_{8192}), T_{test}=256)$             | 762.86   | 21.30     | 40.94      | 17.85           | 17.20                        |
> | $ P(M(x_{8192}[-T_{train}:]), T_{test}=256)$| 19.78    | 20.02     | 19.85      | 19.38           | 18.98                        |
> | $\Delta P_{8192}$                        | -743.08  | -1.28     | -21.09     | 1.53            |  **1.78**                         |

---

> ### Author Response · Authors · 2024-12-03
> **Response to Reviewer UyHR (3/3)**
>
> **Q2: The visualization of learnable position encodings**
>
> A2: **We have clearly presented the learned position encodings in Appendix K, with lengths from 512 to 8192**. According to the visualization, we could find that learned position encoding has both a local-pattern head (long-term decay) and an anti-local head (pay attention to long-distance information)
>
> **Q3: investigating the impact of different training hyperparameters or conducting ablation studies**
>
> A3: **In this work, we have conducted extensive experiment results with different training parameters and ablation studies, including different lengths, different model sizes, different position encoding,different $D_{dape}$ dimension sizes,  methods,and different datasets.**
>
> **Different Length**: we discuss the different length performances in Section 4.1, compared with the baseline
>
> **Different Model Size**: we discuss the model size in Section 4.3: The Effect of Model Size
>
> **Different Position Encoding**: we discuss the different position encoding in Section 4.4, the impact of $DAPE_{1x3}$.
>
> **Ablation: the performance under information leakage**: we discuss how the model performance under information  leakage in Section 4.5, The Performance  with Information Leakage
>
> **Different $D_{dape}$ dimension sizes**: we discuss the the $D_{dape}$ dimension sizes in Section 4.6, comapre DAPE and $DAPE_{1x3}$ with approximate cost.
>
> **Different Kernel Size**: we discuss the performance of different kernel sizes in Section 4.7, the performance of different kernel size.
>
> **Different Datasets**: we use the Arxiv and Books datasets, as long as the 14 downstream datasets.
>
> **Based on the above Clear and Strong evidence, We Request the Reviewer UyHR to reconsider the score, as the mentioned questions are almost ALL clearly present in the paper and the above discussion.**

---

> ### Author Response · Authors · 2024-12-03
> **Further Experiment is On the Way**
>
> Dear Reviewer UyHR,
>
> Moreover, we are conducting the following experiments for further analysis.
>
> **The experiment of ALiBi (compare DAPE-ALiBi and $DAPE_{1x3}$-ALiBi) with larger model size**
>
> **The visualization of $DAPE_{1x3}$-ALiBi**
>
> **If there are any further comments on the ongoing experiments (for example, if you need more), please let us know**

---

> ### Author Response · Authors · 2024-12-03
> **ALiBi-Related Visualization**
>
> Dear Reviewer UyHR,
>
> **We have finished the visualization of ALiBi.**
>
> **Comapred with Kerple, the ALiBI position encoding is linear relation with evaluation length and the largest absulate value is significantly larger**
>
> ALiBi: the bias matrix is $b(i,j) = -r|i-j|$, with the scaler $r>0$ as a hyper-parameter, which is usually set as $\frac{1}{2^h}$ where h is the head number index. **When the evaluation length is 8192, the ALiBi could achieve -6000 position encoding value, while Kerple only achieves -50 position encoding value.** Therefore: the ALiBi absolute position encoding value will become very large when the evaluation length becomes large, and **the ALiBI position encoding value is UnLearnable**. **This makes the ALiBi unstable when the evaluation length is large compared to the training length.**

---

> ### Author Response · Authors · 2024-12-04
> **ALiBI under Different Hyperparameters**
>
> Dear Reviewer UyHR,
>
> According to your suggestion, we have tried ALiBI with different hyperparameters. **The $DAPE_{1x3}-ALiBi$ may need more training cost (longer training iterations, more training tokens, larger model size, and so on) to present its ability, caused by 1) ALiBi position encoding is not learnable and totally based on period knowledge; 2) larger kernel size may take more effort to present its ability[1,2].**
>
> *Table: Perplexity Performance with on Arxiv Dataset*
> | Model          | Training Iteration | Model Size | Training Length|128   | 256   | 512   | 1024   | 2048    | 4096    | 8192|
> |----------------|-------|-------|-------|--------|---------|---------|---------|---------|---------|---------|
> | Alibi          |50K| 125M|128|8.33  | 7.16  | 6.12  | 6.32   | 6.56    | 6.42    | 6.05    |
> | DAPE Alibi     | 50K|125M|128|8.24  | 7.02  | 5.48  | **5.43**   | **5.64**    | **5.55**    | **5.18**    |
> | $DAPE_{1x3}$ Alibi   | 50K|125M|128|**8.17**  | **6.94**  | **5.42**  | 6.69   | 9.53    | 10.52   | 9.55    |
> | DAPE Alibi     | **200K**|125M|128|7.52  | 6.35  | 5.09  | 5.11   | **5.27**    | **5.07**    |-|
> | $DAPE_{1x3}$ Alibi   | **200K**|125M|128|**7.47**  | **6.29**  | **4.99**  | **4.90**   | 9.57    | 13.05    |-|
> | DAPE Alibi     | 50K|**350M**|128|7.71  | 6.55  | 5.09  | 5.05   | **5.24**    | **5.16**    |-|
> | $DAPE_{1x3}$ Alibi   | 50K|**350M**|128|**7.65**  | **6.48**  | **4.99**  | **4.93**   | **5.25**    | 5.73    |OOM|
> | DAPE Alibi     |50K|125M|**512**| - | -|4.52  | 4.23  | 4.24  | 4.04   | 3.83    |
> | $DAPE_{1x3}$ Alibi   | 50K|125M|**512**| - | -|**4.46**  | **4.17**  | **4.17**  | **3.97**   | **3.76**    |
>
>
>
> Analze the above Table:
> * With 50K iterations, 125M model and 128 training length
>     * **Within evaluation length 512**,  $DAPE_{1x3}-ALiBi$ achieves better or comparable performance than DAPE-ALiBI.
> * With **200K iterations**, 125M model and 128 training length
>     * **Within evaluation length 1024**,  $DAPE_{1x3}-ALiBi$ achieves better or comparable performance than DAPE-ALiBI.
> * With 50K iterations, **350M** model and 128 training length
>     * **Within evaluation length 2048**,  $DAPE_{1x3}-ALiBi$ achieves better or comparable performance than DAPE-ALiBI.
> * With 50K iterations, 125M model and **512 training length**
>     * **Within evaluation length 8192**,  $DAPE_{1x3}-ALiBi$ achieves better or comparable performance than DAPE-ALiBI.
>
> **Therefore, according to the above experiment results, we could have confidence to say that:**
> * $DAPE_{1x3}-ALiBi$ will present its ability with longer training (longer training iterations, more training tokens, larger model size , and so on) to present its ability, compared to DAPE-ALiBI.
> * This is caused by:
>    * ALiBI position encoding is not learnable and is designed by hand. Hence, we have to pay more cost to help the model adapt to ALiBI
>    * The larger kernel size may need longer training to present its ability [1,2].
>
> **Reviewer UyHR believes a deeper investigation into the reasons behind these trends is needed to fully justify a higher rating, and suggest the visualization of learnable positions encoding and analyzing the impact of different training hyperparameters. We have finish both and will definitely add them to the final version.**
>
> **As we have finished the required position encoding visualization and the impact of different training hyperparameters, we have further investigated and conducted an in-depth analysis of the reasons behind this trend. Therefore, could Reviewer UyHR consider raising the score?**
>
> If you have any further questions, **you can click the Edit button to update your official review and give comments**. And we will give the response immediately.
>
> Reference:
> [1] Chen, H., Chu, X., Ren, Y., Zhao, X., & Huang, K. (2024). PeLK: Parameter-efficient Large Kernel ConvNets with Peripheral Convolution. In Proceedings of the IEEE/CVF Conference on Computer Vision and Pattern Recognition (pp. 5557-5567).
>
> [2] Chen, Y., Liu, J., Zhang, X., Qi, X., & Jia, J. (2023). Largekernel3d: Scaling up kernels in 3d sparse cnns. In Proceedings of the IEEE/CVF Conference on Computer Vision and Pattern Recognition (pp. 13488-13498).

---

### Official Review · Reviewer_i54z · 2024-11-04

**Soundness:** 2
**Presentation:** 2
**Contribution:** 2
**Rating:** 6
**Confidence:** 3

**Summary:**

The authors propose to improve the extrapolation abilities of Transformer models beyond their training sequence length by building upon the previously introduced method of data-adaptive positional encodings (DAPE). The authors find that replacing DAPE’s standard MLP through a convolutional MLP further improves performance.

**Strengths:**

**Originality & Significance:**
- The authors slightly expand on the insights of the original DAPE paper and provide new results on the original and their improved variant

**Quality:**
- Experiments conducted across two datasets in comparison with multiple popular ‘positional embedding’ methods, including NoPE, RoPE, CoPE, ALiBi, Kerple and FiRE
- Insights into how the computational complexity is affected are provided, as well as results for three model sizes

**Clarity:**
- The paper is mostly easy to read;
- Graphs and tables are clearly labeled and easy to interpret

**Weaknesses:**

_TL;DR: While I appreciate the work the authors have put into the manuscript and their experiments, the main ‘methodological’ novelty facilitating the approach has already been presented in the original DAPE paper. The authors’ addition of using a convolution instead of an MLP (i.e. replacing a 1x1 conv with a 1x3 conv, combined with inconsistent improvements) combined with the manuscript in its current state is in my opinion not enough to pass the bar for ICLR;_

-	Minor ‘methodological’ addition to existing DAPE, with results varying from ‘improvement’ to ‘decrease in performance’ – see questions.
-	Insufficient (no) discussion of limitations, although inconsistencies can be easily seen from the presented results – see questions.
-	Interpretation on an in-sight level of results obtained with different method (FIRE, ALiBi, etc.) could be significantly extended
-	Minor: Quality of Manuscript in terms of wording/preciseness of statements

**Questions:**

**Main concerns, questions & potential improvements:**
- Most results (in fact, almost all) are reported with for ‘Kerple’, which seems to work well (e.g. Figure 2, Figure 3, and Figure 3) in combination with DAPEv2;
However, when looking at the ‘broader’ applicability in Figure 5, it quickly becomes clear that results across the board are much more inconsistent!
-> e.g. ALiBi: ALiBi performs well on its own for training seq-len 128, is improved by DAPE-ALiBi – but significantly worse for DAPEv2;
The manuscript however states that DAPE-1x3 ‘consistently improves performance’, which is incorrect and should be discussed (including insights)
- Appendix E / Section 4.8 shows results for DAPE with kernel-size 1 and 3 – I assume ‘1’ is the classic DAPE, and ‘3’ the v2?
If so, again – results vary a lot in terms of which one is better for which task and combined with which ‘pe-method’, and I don’t see this discussed in the manuscript appropriately.
- General: A wider discussion of the limitations would significantly help any reader/user, and I’d suggest the authors consider being upfront about these and provide the reader with helpful guidance (Similarly when using DAPEv2 with FIRE, while there is some improvement, it still ‘diverges’ quickly)

- I’d like the authors to include actual insights based on their experiences and the background knowledge of working with these different approaches (FIRE, ALiBi, Kerple, etc.) – e.g. is one generally preferable? If not, what are the situations you would recommend combining DAPEv2 with any particular one of these?
- In Figure 6, although the model can cheat, I’d be curious why the authors think that the DAPEv2-ALiBi becomes significantly less stable (than both non-cheating and original-non-cheating)

Additional comments:
- I’d suggest the authors replace some of the references through the seminal works in their introduction in terms of how Transformers have made an impact (e.g. noting CV but not citing ViT/DeiT isn’t good research practice, as these authors should be acknowledged)
- I’d like to suggest the authors to check and potentially slightly rework the manuscript in terms of preciseness of their wording; While I am aware this might be due to language barrier, there are multiple instances where statements are misleading/confusing/too general, e.g.
  - Abstract: ‘[…] contributing to interactions among distinct tokens, in contrast to earlier feed-forward NNs’ -> This is not really true/correct, as any FFN can establish interactions between elements of data – e.g. a CNN establishes the same over a local window in a sequence, etc.;
  - L 49: ‘rendering the outputs non-sensical’ -> In the context of NLP, the output will still be a valid word and hence ‘sensical’, the architecture simply loses its ability to learn relationships over a sequence and reverts back to sets/bag-of-words; Also note: The authors discuss “Transformers” in general, and there actually are multiple use cases where Transformers are used on set-based problems
  - …


---
## Update post-rebuttal:
Some of my concerns have been addressed, and I am therefore increasing my score slightly from 5 to 6 -- but it still remains a borderline case to me.

---

> ### Author Response · Authors · 2024-11-13
> **Response to Reviewer i54z (Part 1/4)**
>
> Dear Reviewer i54z,
>
> Thank you very much for your comments, we will address your concerns below.
>
> **A1: Minor ‘methodological’ addition to existing DAPE, with results varying from ‘improvement’ to ‘decrease in performance’ – see questions.**
>
> Q1: We discuss the $DAPE_{1x3}$ improvement in **The improvement compared to DAPE** in **Response to ALL Reviewers**.
>
> **Original DAPE explanation is NARRAW.** The two key differences between DAPE and this work are:
>
> 1) **Insight**: DAPE attributes length extrapolation performance gains to adaptive position encoding and DAPE believes that the Bias Matrix Is Necessary, while this work finds DAPE could still improve performance without position encoding so that we take a broader view, explaining that the Transformer's length extrapolation ability is limited by the expressiveness of the naive query-key dot product, which can be enhanced using image processing techniques;
> 2) **Performance**: DAPE is designed for additive RPE and may underperform with non-additive RPE (e.g., RoPE), whereas this work suggests that increasing kernel size (e.g., with $DAPE_{1x3}) may improve RoPE's performance.
>
>
> We have discussed the relation between kernel size and performance in Section 4.7: Different experiment settings may have different optimal kernel sizes. **We do not claim that larger kernel size always brings better performance but suggest that different settings have different optimal kernel sizes**.
>
> *Table: Performance with Different Kernel Sizes (Training Length 128, Evaluation from Length 128 to 8192)*
>
> | Dataset | Method                                  | 128   | 256   | 512   | 1024  | 2048  | 4096  | 8192  |
> |---------|-----------------------------------------|-------|-------|-------|-------|-------|-------|-------|
> | Arxiv   | Kerple                                  | 8.30  | 7.10  | 5.85  | 6.91  | 9.17  | 11.48 | 12.59 |
> |         | DAPE-Kerple (Kernel Size 1x1)           | 8.21  | 6.98  | 5.38  | 5.20  | 5.33  | 5.26  | 4.97  |
> |         | $\textrm{DAPE}_{1\times3}$-Kerple (Kernel Size 1x3) | 8.15  | 6.92  | 5.29  | 5.05  | 5.11  | 4.95  | 4.60  |
> |         | $\textrm{DAPE}_{1\times5}$-Kerple (Kernel Size 1x5) | 8.13  | 6.91  | 5.27  | 5.04  | 5.10  | 4.91  | 4.57  |
> |         | $\textrm{DAPE}_{1\times7}$-Kerple (Kernel Size 1x7) | **8.12** | **6.89** | **5.26** | **5.02** | **5.09** | **4.91** | **4.57** |
> | Books3  | Kerple                                  | 32.10 | 29.09 | 28.10 | 35.75 | 44.68 | 56.39 | 66.23 |
> |         | DAPE-Kerple (Kernel Size 1x1)           | 31.49 | 28.27 | 24.93 | 24.31 | 23.34 | 24.38 | 25.01 |
> |         | $\textrm{DAPE}_{1\times3}$-Kerple (Kernel Size 1x3) | 31.07 | 27.81 | 24.38 | 23.57 | 22.40 | 23.19 | **23.52** |
> |         | $\textrm{DAPE}_{1\times5}$-Kerple (Kernel Size 1x5) | 31.02 | 27.79 | 24.36 | 23.57 | 22.41 | 23.32 | 23.71 |
> |         | $\textrm{DAPE}_{1\times7}$-Kerple (Kernel Size 1x7) | **30.98** | **27.76** | **24.31** | **23.47** | **22.30** | **23.00** | 23.57 |
>
> *Table: Performance with Different Kernel Sizes (Training Length 512, Evaluation from Length 512 to 8192)*
>
> | Dataset | Method                                  | 512   | 1024  | 2048  | 4096  | 8192  |
> |---------|-----------------------------------------|-------|-------|-------|-------|-------|
> | Arxiv   | Kerple                                  | 4.57  | 4.37  | 5.09  | 6.80  | 9.08  |
> |         | DAPE-Kerple (Kernel Size 1x1)           | 4.49  | 4.20  | 4.17  | 3.95  | 3.70  |
> |         | $\textrm{DAPE}_{1\times3}$-Kerple (Kernel Size 1x3) | 4.44  | 4.14  | 4.09  | 3.87  | 3.58  |
> |         | $\textrm{DAPE}_{1\times5}$-Kerple (Kernel Size 1x5) | 4.44  | 4.14  | 4.10  | 3.85  | 3.59  |
> |         | $\textrm{DAPE}_{1\times7}$-Kerple (Kernel Size 1x7) | **4.43** | **4.13** | **4.08** | **3.85** | **3.57** |
> | Books3  | Kerple                                  | 19.83 | 19.19 | 20.48 | 28.33 | 40.94 |
> |         | DAPE-Kerple (Kernel Size 1x1)           | 19.25 | 18.28 | 17.20 | 17.58 | 17.85 |
> |         | $\textrm{DAPE}_{1\times3}$-Kerple (Kernel Size 1x3) | 18.95 | 17.92 | 16.79 | 17.05 | 17.20 |
> |         | $\textrm{DAPE}_{1\times5}$-Kerple (Kernel Size 1x5) | 18.89 | 17.87 | 16.76 | 17.09 | **17.10** |
> |         | $\textrm{DAPE}_{1\times7}$-Kerple (Kernel Size 1x7) | **18.86** | **17.82** | **16.70** | **17.01** | 17.16 |
>
>
> **Different experiment settings may have different optimal kernel sizes**. For the Arxiv dataset, larger kernel sizes consistently achieve
> better performance, evaluating with training lengths of 128 or 512. However, for the Books3 dataset,
> $DAPE_{1×3}$ performs best when the training length is 128 and evaluated at 8192, whereas DAPE1×5
> performs best at the same evaluation level when the training length is 512. Although larger kernel sizes contribute to stronger expressiveness from intuition, we
> conjecture that the performance degradation for overly large kernel sizes results from optimization
> challenges.

---

> ### Author Response · Authors · 2024-11-13
> **Response to Reviewer i54z (Part 2/4)**
>
> **A2: Most results (in fact, almost all) are reported with for ‘Kerple’, which seems to work well (e.g. Figure 2, Figure 3, and Figure 3) in combination with DAPEv2; However, when looking at the ‘broader’ applicability in Figure 5, it quickly becomes clear that results across the board are much more inconsistent**
>
> Q2: There may be a misunderstanding here. **All baseline (ALiBi, Kerple or FIRE) performance can be improved with enough training length (such as at least training length 512)**
>
> **We will explain it step by step:** 1) We have discussed in Section 4.7: Different experiment settings may have different optimal kernel sizes and larger kernel size does not always bring better performance but provides the potential; 2) The Explanation of $DAPE_{1x3}$-ALiBi performance; 3) The revised presentation of Section 4.4.
>
> **We have already highlighted several times in our paper (Section 4.7 and Section 4.8) that Different experiment settings may have different optimal kernel sizes**.
>
> As the Table shown in Q1 (Paper Section 4.7), larger kernel size doest not always brings better performance. Also, we also hightlight in Section 4.8 that Different tasks have different optimal kernel sizes and The large kernel size performance improvement is related to the baseline bias matrix. Therefore, the choice of kernel size is related to bias matrix and different experiment setting, and different setting has different optimal kernel size.
>
>
> **The Explanation of $DAPE_{1x3}$-ALiBi performance**
>
> Different experiment setting has different optimal kernel size. **The $DAPE_{1x3}-ALiBi$ needs a larger training sequence length so that $DAPE_{1x3}-ALiBi$ achieves better performance than $DAPE_{1x1}-ALiBi$ when the training length is 512** but worse performance when the training length is 128.
>
> **The revised presentation of Section 4.4.**
>
> We add the following sequence to Section 4.4: The $DAPE_{1x3}-ALiBi$ may needs longer training length so that the performance is better than $DAPE-ALiBi$
>
> **Q3: Results vary a lot in terms of which one is better for which task and combined with which ‘pe-method’, and I don’t see this discussed in the manuscript appropriately.**
>
> A3: **For language modeling tasks, with enough longer training length (such as at least 512), the larger kernel size usually brings better performance.** **Also, we have discussed the impact of bias matrix (which is pe-method) on other tasks in Section 4.8 and also the impact of kernel size in Section 4.7 (discussed in Q1)**. We directly copy the discussion of Section 4.8 here.
>
> **Different tasks have different optimal kernel sizes, as shown in Appendix G and Appendix F.**
> For example, on MISSING DUPLICATE task, the DAPE1×3-Kerple improves the 87.57 of DAPEKerple to 99.65. However, on the STACK MANIPULATIONtask, the DAPE1×3-Kerple decreases
> the 72.04 of DAPE-Kerple to 68.18. Also, as shown in Appendix D, the larger kernel size does
> not always lead to better performance. Overall, larger kernel size provides a potential way to improve the Transformer length extrapolation performance, and we usually could find a suitable kernel
> size (ranging from 1×1 to larger kernel sizes) to achieve better performance than without further
> processing attention score.
>
> **The large kernel size performance improvement is related to the baseline bias matrix(which is pe-method)**. As
> shown in Appendix E, the best performance is usually achieved by further processing attention
> scores via kernel size 1 or 3. Moreover, on 11 permutation-variant tasks, the DAPE1×3-Kerple
> achieves better performance on 8 of 11 tasks compared to Kerple. And the DAPE1×3-FIRE achieves
> better performance on 6 of 11 tasks compared to FIRE. This suggests that the large kernel size
> performance improvement is related to the baseline bias matrix.
>
>
>
>
> **Q4: General: A wider discussion of the limitations would significantly help any reader/user, and I’d suggest the authors consider being upfront about these and provide the reader with helpful guidance (Similarly when using DAPEv2 with FIRE, while there is some improvement, it still ‘diverges’ quickly)**
>
> A4: The improvement is related to the baseline pe-method (such as ALiBi, Kerple, or FIRE), and longer training length could partialy solve the diverges problem.
>
> * **Longer Training Length (usually at least 512)**. The performane diverges quickly when the training length is small (such as 128) but works well when the training length is larger (such as 512).
> * **Baseline PE Method is important for the performance and Kerple is a good default choice**. Choose a suitable bias matrix (such as Kerple or FIRE) for training.
> * The effect of $D_{DAPE}$. The hidden dimension of DAPE could be small, and 10 is enough (as shown in Figure 7).

---

> ### Author Response · Authors · 2024-11-13
> **Response to Reviewer i54z (Part 3/4)**
>
> **Q5: I’d like the authors to include actual insights based on their experiences and the background knowledge of working with these different approaches (FIRE, ALiBi, Kerple, etc.) – e.g. is one generally preferable? If not, what are the situations you would recommend combining DAPEv2 with any particular one of these**
>
> A5: **The Kerple is a good choice for almost all settings, the FIRE may need longer training length/tokens to present its ability, and do not use ALiBi unless necessary .**
>
> * It is easy to train Kerple, as Kerple usually has few trainable parameters compared to FIRE. If you do not know which one to use, directly use Kerple.
> * FIRE may have better performance, but may need longer training length (diverges at 128 but works well at 512, with DAPE).  FIRE $b(i,j) = f_{\theta}\left(\frac{\psi(i-j)}{\psi(\max\{L, i\})}\right)$ so that we may need longer training length or more training tokens to well-train the neural network $f_{\theta}$.
> * Do not use ALiBi unless necessary. The ALiBi will quickly become local attention as the sequence length increases.
>
>
> **Q6: In Figure 6, although the model can cheat, I’d be curious why the authors think that the DAPEv2-ALiBi becomes significantly less stable (than both non-cheating and original-non-cheating)**
>
> A6: **The Figure 6 is presented with training length 128. The $DAPE_{1x3}-ALiBi may needs longer training length to make it stable (such as longer length with 512). Moreover, compared to Kerple and FIRE, the baseline ALiBi actually almost became local attention to keep the low perplexity so that it actually could not get information from the previous sequence, while other position encodings (such as RoPE or Kerple) do not abandon long-distance information and they cannot handle them well so that the perplexity is higher.**
>
> In this discussion, we explore how to effectively use perplexity as a metric, incorporating concepts of information gain and entropy. Let $P(\cdot)$ represent the process for calculating perplexity, and $ M(x) $ denote the logit output generated by the model after processing an input sequence $ x $. For evaluating model performance, we define $ P(M(x), K) $ as follows:
> * Process the entire sequence $ x $ using $ M(x)$.
> *  Compute the perplexity on the last $ K $ tokens of the sequence.
>
> To interpret information gain, we consider the training sequence length $ T_{\text{train}}$. Given an input $ x $, we calculate the change in loss/perplexity, $ \Delta P$, as: $\Delta P = P(M(x[-T_{\text{train}}:]), T_{\text{test}}) - P(M(x), T_{\text{test}})$
>
>
>
> The term $\Delta P$ provides insights into the model's information gain relative to local and global context, allowing us to quantify entropy in terms of model uncertainty reduction. We interpret $ \Delta P $ with $T_{train}=512$ as follows:
>
>
> | **Method**                               | **RoPE** | **ALiBi** | **Kerple** | **DAPE-Kerple** | **$DAPE_{1x3}$-Kerple** |
> |------------------------------------------|----------|-----------|------------|-----------------|-----------------------------|
> | $ P(M(x_{512}), T_{test}=256)$              | 19.74    | 20.04     | 19.83      | 19.25           | 18.95                        |
> | $ P(M(x_{1024}), T_{test}=256)$             | 261.39   | 19.74     | 19.19      | 18.28           | 17.92                        |
> | $ P(M(x_{1024}[-T_{train}:]), T_{test}=256)$| 19.51    | 19.79     | 19.58      | 19.03           | 18.74                        |
> | $\Delta P_{1024}$                        | -241.88  | 0.05      | 0.39       | 0.75            | **0.82**                         |
> | $ P(M(x_{2048}), T_{test}=256)$             | 411.23   | 20.17     | 20.48      | 17.20           | 16.79                        |
> | $ P(M(x_{2048}[-T_{train}:]), T_{test}=256)$| 18.74    | 19.03     | 19.84      | 18.28           | 18.01                        |
> | $\Delta P_{2048}$                        | -392.49  | -1.14     | -0.64      | 1.08            |  **1.22**                         |
> | $ P(M(x_{4096}), T_{test}=256)$             | 635.80   | 20.50     | 28.33      | 17.58           | 17.05                        |
> | $ P(M(x_{4096}[-T_{train}:]), T_{test}=256)$| 19.11    | 19.35     | 19.07      | 18.59           | 18.19                        |
> | $\Delta P_{4096}$                        | -616.69  | -1.15     | -9.26      | 1.01            |  **1.14**                         |
> | $ P(M(x_{8192}), T_{test}=256)$             | 762.86   | 21.30     | 40.94      | 17.85           | 17.20                        |
> | $ P(M(x_{8192}[-T_{train}:]), T_{test}=256)$| 19.78    | 20.02     | 19.85      | 19.38           | 18.98                        |
> | $\Delta P_{8192}$                        | -743.08  | -1.28     | -21.09     | 1.53            |  **1.78**                         |

---

> ### Author Response · Authors · 2024-11-13
> **Response to Reviewer i54z (Part 4/4)**
>
> * When $ \Delta P = 0 $: The model’s information gain from the full sequence is negligible, indicating an entropy level comparable to local attention (e.g., models like ALiBi when the evaluation length is 1024). This suggests the model does not leverage context beyond a limited range.
>
> * When $\Delta P < 0 $: Processing the entire sequence increases entropy, resulting in worse performance than focusing only on the last $ T_{\text{train}} $ tokens. This implies negative information gain and limited extrapolation capability (e.g. such as RoPE), as the model may overfit to recent tokens without capturing broader context effectively.
>
> * When $\Delta P > 0$: The model benefits from the information within $ x[:T_{\text{train}}] $, achieving a reduction in entropy that reflects positive information gain. This suggests the model leverages contextual information beyond the training sequence, indicating extrapolation capability.
>
> By examining $ \Delta P$, we can evaluate the model’s ability to reduce entropy and gain information from extended sequences, providing a measure of its extrapolative power. **Apparently, the ALiBI abandons long-distance information so do not use ALiBI unless necessary**.
>
>
>
> **Q7: I’d suggest the authors replace some of the references through the seminal works in their introduction in terms of how Transformers have made an impact (e.g. noting CV but not citing ViT/DeiT isn’t good research practice, as these authors should be acknowledged)**
>
> A7: Thank you very much for your comments. We have added the suggested works, including CV-related, ViT/DeiT, swin transformer and so on.
>
> **Q8: I’d like to suggest the authors to check and potentially slightly rework the manuscript in terms of preciseness of their wording; While I am aware this might be due to language barrier, there are multiple instances where statements are misleading/confusing/too general, e.g.**
>
> A8: Thank you very much for your suggestions, we have revised the corresponding sentences and we are carefully checking other presentations.
>
> * Abstract (FFN):
>    * Original: The attention mechanism is a fundamental component of the Transformer model,
> contributing to interactions among distinct tokens, in contrast to earlier feedforward neural networks
>    * Current: The attention mechanism is a fundamental component of the Transformer model,
> contributing to interactions among distinct tokens
>
> * L 49: ‘rendering the outputs non-sensical’
>    * Original: Without these encodings,
> token generation would lack the necessary contextual order, rendering the outputs nonsensical.
>    * Current: Without these encodings,
> token generation would lack the necessary contextual order.
>
> If there are any questions, please let us know. And if you think that we have addressed your concerns, could you please consider raising the score? Thank you very much for your support.

---

> ### Comment · Reviewer_i54z · 2024-11-25
> **Thanks to the authors**
>
> I'd like to sincerely thank the authors for the amount of work they have put into the rebuttal!
> Some of my concerns have been addressed, and I am therefore increasing my score slightly from 5 to 6 -- but it still remains a borderline case to me.
> I'd highly encourage the authors to also add the insights presented in this rebuttal into the paper (and/or appendix), and to continue refining the wording for improved clarity (as mentioned across the reviews).

---

> ### Author Response · Authors · 2024-11-25
> **Response to Reviewer i54z**
>
> Dear Reviewer i54z,
>
> **Thank you very much for your support. We will definitely add the insights presented in this rebuttal into the paper (and/or appendix), and also to continue refining the wording for improved clarity**
>
> Moreover, please let us know if there is any concerns left or anything that you would like to discuss. If you are interested, the following are the potential topics:
> * Any left concerns or anything that is unclear?
> * Our vision of the length extrapolation/long-context problem.
> * Our understanding of the attention score
> * Or anything else that you would like to discuss.
>
> **Finally, we thanks Reviewer i54z again for your precious support. And please let us know whether there is any thing that you would like to discuss, including but not limited to this paper, our vision of the length extrapolation problem or anything else.**

---

### Author Response · Authors · 2024-11-13
**Response to All Reviewers**

Dear all reviewers:

We sincerely appreciate the reviewers for their time and effort in the review. We first address some common questions, followed by detailed responses to each reviewer separately. We hope our responses clarify existing doubts. We will really appreciate it if reviewers could kindly reconsider the decision, provided that the main comments are well addressed.

**Q1: The key difference between DAPE and this work. (Reviewer i54z, Reviewer UyHR, Reviewer 7aAt)**

A1: We have highlighted the difference between DAPE and this work at the beginning of Section 3 Method. We directly copy it below.

**The two key differences between DAPE and this work are**:
* **1) Insight:** DAPE attributes length extrapolation performance gains to adaptive position encoding and DAPE believes that the Bias Matrix Is Necessary, while this work finds DAPE could still improve performance without position encoding so that **we take a broader view, explaining that the Transformer's length extrapolation ability is limited by the expressiveness of the naive query-key dot product, which can be enhanced using image processing techniques;**
* **2) Performance:** DAPE is designed for additive RPE and may underperform with non-additive RPE (e.g., RoPE), whereas this work suggests that increasing kernel size (e.g., with $DAPE_{1x3}) may improve RoPE's performance**.

**Q2: The improvement compared to DAPE(Reviewer  i54z, Reviewer UyHR)**


A2: **The improvement of $DAPE_{1x3}$ is significant, compared to the original DAPE**. The following are the perplexity results on the Arxiv dataset with training length 512 and model size 125M.

| Method   |512       | 1024       | 2048       | 4096   | 8192 |
|-----|-----------|-----------|-----------|------------|------------|
| NoPE   | 4.68  | 31.79  | 1867.46  | 4666.60   | 5334.85  |
| DAPE-NoPE   | 4.63  | 12.72  | 751.78  | 2033.33   | 2618.13   |
| $DAPE_{1x3}-NoPE$   | 4.47  | 6.31  | 56.93  | 196.80  | 259.81   |
| RoPE  | 4.57  | 43.62  | 144.05  | 278.87   | 297.06   |
| DAPE-RoPE   | 4.53  | 73.31  | 174.48 | 316.84   | 306.78   |
| $DAPE_{1x3}-RoPE$   | 4.48  | 13.10  | 29.41  | 53.97   | 68.32   |

* For NoPE: reduce the perplexity **from 2618.13 to 259.81.**
* For RoPE: reduce the perplexity **from 306.78 to 68.32.**

**Q3: The major contribution of the work, compared to previous work related to length extrapolation  (Reviewer i54z, Reviewer UyHR, Reviewer 7aAt)**

A3: The major contribution of the work is the interpretation of the length extrapolation problem.

**The Importance of Perspective of long-context problem and the Corresponding Research Directions**:
* **Transformer [1] suggests that position encoding is important** so that we have better position encoding methods such as RoPE, ALiBi, Kerple, FIRE, CoPEand so on.
* **Position Interpolation [2] suggests Transformer performs badly because of unseen position ID** so that it down-scales the input position indices so that we have better methods including YaRN, CLEX, and so on.
* Now, **we suggest that the Length Extrapolation is caused by the limited expressiveness of the query and key dot product. Though the current method is straightforward, our community could develop better methods in the future based on this perspective.**

In this work, we first point out that the length extrapolation problem is caused by the limited expressiveness of the query and key dot product. Therefore, from now on, We Do Not have to employ complex methods for length extrapolation/long-context but directly enhance the attention score via image processing techniques.

Before this work, There is NO Paper interpreting long-context/length extrapolation on such perceptive. Following the direction, there are many potential works in the future.
* How about utilizing more powerful image processing methods to further process attention scores?
* If the attention score is just a feature map, then why not first resize it to a lower resolution to process it to reduce the cost?
* Besides regarding the attention score as feature maps, what else could the attention be regarded?

**This work is NOT a simple extension related to DAPE, but contributes an important perspective of how we think about the length extrapolation/long context problem.**  Based on such interpretation, better attention score processing methods will be developed in the future, and long-context solutions WILL  NOT be limited to position encoding design or position interpolation any more.

Reference:

[1] Vaswani, A. (2017). Attention is all you need. Advances in Neural Information Processing Systems.

[2] Chen, S., Wong, S., Chen, L., & Tian, Y. (2023). Extending context window of large language models via positional interpolation. arXiv preprint arXiv:2306.15595.

---

### Author Response · Authors · 2024-11-22
**Update Summary and the Core Contribution**

Dear AC and Reviewers,

Thank you very much for your arrangement and comments. We have updated the paper and highlighted the changes in red color.
* We add the results of new results, including $\Delta P$.
* We further discuss the related works, including ViT/DeiT, and On Translation Invariance in CNNs.
* Various writing improvements.

Moreover, we would like to highlight the core contribution of the work: the methodology is NOT the most important part of the contribution, but **the core contribution is: We Provide a Totally New Perspective of Long-Context/Length Extrapolation Problem**. **The experiment parts are all used to prove such perspective**.

Next, we will discuss how such a perspective could have an impact on the following works and the community.
* **Before this work, we mainly solved long-context/length extrapolation from two directions/perspectives:**
   * **Research Direction 1: Better Position Encoding (proposed by Vaswani, A. (2017) ).** With such perspectives, our community develop different position encodings, including RoPE, ALiBi, Kerple, FIRE and so on.
   * **Research Direction 2: Position Interpolation (proposed by Chen, S. (2023) ).** With such perspectives, there are extensive following papers, including YaRN, CLEX and so on.
* **In this work, we propose the Third Research Direction: the length extrapolation is difficult because the query and key dot product have limited expressiveness, and direct convolution operation on attention score could significantly improve the length extrapolation performance, whatever the position encoding is  (e.g. reduce NoPE perplexity from 2618.13 to 259.81)**.

Following this direction, we could have extensive potential works in the future. Here, we leave several questions that we are also interested to share:
   * Besides regarding the attention score as a feature map, what else could the attention score be regarded as?
   * Besides the convolution operation, any other operations that can also improve the expressiveness of attention score?
   * What is the essence of attention score? If the attention score is Just Feature Map, then could we use a better way to construct the attention score?
   * ... ...

**It is the work that first suggests and proves the length extrapolation/long-context is limited by naive query and key dot product.**

**Besides developing better position encoding or better position interpolation method, now we have the third direction: improve the expressiveness of query and key dot product (attention score expressiveness)**.

Again, thank you all for your attention to this work.

Best regards,

Paper 4210 authors

---

### Author Response · Authors · 2024-12-04
**Discussion Period Summary**

**After the rebuttal, we addressed the Reviewer's concerns, the Reviewers became more positive about the paper and  there is no more clearly presented concerns after the rebuttal:**
* **The Reviewer i54z increased the score to 6, and there are no more concerns mentioned.**
* **The Reviewer UyHR claims that we have addressed the initial concerns and thanks for addressing the previous concerns**. And the last question is also well-addressed.
* **The Reviewer 7aAt also increased the score, and there are no more concerns mentioned.**

**The Core Contributions** of this work: besides position encoding and position interpolation for length extrapolation, we propose the third research direction: improve the expressiveness of query and key dot product for attention score.
* **Before this work, we mainly solved long-context/length extrapolation from two directions/perspectives:**
   * **Research Direction 1: Better Position Encoding (proposed by Vaswani, A. (2017) ).** With such perspectives, our community develops different position encodings, including RoPE, ALiBi, Kerple, FIR,E and so on.
   * **Research Direction 2: Position Interpolation (proposed by Chen, S. (2023) ).** With such perspectives, there are extensive following papers, including YaRN, CLE,X and so on.
* **In this work, we propose the Third Research Direction: the length extrapolation is difficult because the query and key dot product have limited expressiveness, and direct convolution operation on attention score could significantly improve the length extrapolation performance, whatever the position encoding is.

**The Significant Performance Improvement**
*Table: Perplexity Performance with Different Kernel Sizes (on Arxiv Datasets, Training Length 512, Evaluation from Length 512 to 8192)*
| Method   |512       | 1024       | 2048       | 4096   | 8192 |
|-----|-----------|-----------|-----------|------------|------------|
| NoPE   | 4.68  | 31.79  | 1867.46  | 4666.60   | 5334.85  |
| DAPE-NoPE   | 4.63  | 12.72  | 751.78  | 2033.33   | 2618.13   |
| $DAPE_{1x3}-NoPE$   | 4.47  | 6.31  | 56.93  | 196.80  | 259.81   |
| RoPE  | 4.57  | 43.62  | 144.05  | 278.87   | 297.06   |
| DAPE-RoPE   | 4.53  | 73.31  | 174.48 | 316.84   | 306.78   |
| $DAPE_{1x3}-RoPE$   | 4.48  | 13.10  | 29.41  | 53.97   | 68.32   |

* For NoPE: reduce the perplexity **from 2618.13 to 259.81, which reduces 90.07% perplexity.**
* For RoPE: reduce the perplexity **from 306.78 to 68.32, which reduces 77.72% perplexity.**

The Differential Transformer is a popular work and received an average score of 8 in the ICLR 2025 submission, and we copy its reported loss directly for comparison.
Here, we compare the loss reduction within training length to prove that this work actually achieves significant loss reduction.
*Table: Loss Performance with Different Kernel Sizes (on Books dataset with training length 512)*
| Method   |Loss       | Loss Reduction |Loss Reduction Ratio |
|-----|-----------|-----------|-----------|
| Differential Transformer Baseline   | 3.086  | ---  | ---  |
| Differential Transformer   | 3.062  | 0.024  | 0.77%|
| Kerple (This Work)   | 2.987  |  ---  | ---  |
| DAPE-Kerple (This Work)  | 2.957  | 0.030  | 0.99%|
| $DAPE_{1x3}-Kerple$ (This Work)   | 2.942  | 0.015  | 0.51%|

For the performance within the training length, even reducing 0.01 loss is still very difficult and challenging. The $DAPE_{1x3}-Kerple$ reduces the 0.045 loss compared to Kerple, while Differential Transformer reduces 0.024 loss with a reduction ratio of 0.77%. Even compared with DAPE-Kerple, $DAPE_{1x3}-Kerple$ still reduces 0.015 loss and the reduction ratio is 0.51%. This has proved that the performance gap reduction is actually significant

**The work not only brings significant performance improvement but also helps the community find a New Research Direction and better understand the essence of the long-context/length extrapolation problem. The community could follow such research direction for further analysis, including but not limited to better attention score construction, better feature map processing methods for length extrapolation,and so on.**

Best regards,

Paper 4210 authors

---

### Meta-Review · Area_Chair_swsz · 2024-12-23

**Metareview:**

The proposed method replaces the MLP layer of DAPE with a 1x3 convolutional layer. While the author rebuttal successfully addressed most clarification requests from the reviewers, a critical concern regarding the limited technical novelty (raised by Reviewers i54z and UyHR) remains unresolved. Overall, the contributions of the paper fall short of the ICLR standard, and the AC, therefore, recommends rejecting the paper.

Additionally, the paper must differentiate itself from a significantly related study on correlation-structure-based attention:
[A] Kim et al., Learning Correlation Structures for Vision Transformers, CVPR 2024.

**Additional Comments On Reviewer Discussion:**

The reviewers requested several clarifications and, more critically, raised concerns about the technical novelty of the proposed method. While the clarifications were largely addressed in the author rebuttal, the concern about the limited technical novelty remains unresolved. Specifically, the proposed method simply modifies the MLP of DAPE by replacing it with a 1x3 convolutional layer, which is seen as an incremental change.

---

### Decision · Program_Chairs · 2025-01-22

Reject